# Current and projected regional economic impacts of heatwaves in Europe

David García-León [1✉], Ana Casanueva [2,3], Gabriele Standardi[4,5], Annkatrin Burgstall [2], Andreas D. Flouris [6] & Lars Nybo [7]

Extreme heat undermines the working capacity of individuals, resulting in lower productivity, and thus economic output. Here we analyse the present and future economic damages due to reduced labour productivity caused by extreme heat in Europe. For the analysis of current impacts, we focused on heatwaves occurring in four recent anomalously hot years (2003, 2010, 2015, and 2018) and compared our findings to the historical period 1981–2010. In the selected years, the total estimated damages attributed to heatwaves amounted to 0.3–0.5% of European gross domestic product (GDP). However, the identified losses were largely heterogeneous across space, consistently showing GDP impacts beyond 1% in more vulnerable regions. Future projections indicate that by 2060 impacts might increase in Europe by a factor of almost five compared to the historical period 1981–2010 if no further mitigation or adaptation actions are taken, suggesting the presence of more pronounced effects in the regions where these damages are already acute.

[1] European Commission, Joint Research Centre, Edificio Expo, Inca Garcilaso 3, 41092 Seville, Spain. [2] Federal Office of Meteorology and Climatology MeteoSwiss, 8058 Zurich, Switzerland. [3] Meteorology Group, Department of Applied Mathematics and Computer Sciences, University of Cantabria, 39005 Santander, Spain. [4] Euro-Mediterranean Center on Climate Change (CMCC) and Ca' Foscari University of Venice. Edificio Porta dell'Innovazione, 2nd floor, Via della Libertà 12, 30175 Venice, Italy. [5] RFF-CMCC European Institute on Economics and the Environment (EIEE). Edificio Porta dell'Innovazione, 2nd floor, Via della Libertà 12, 30175 Venice, Italy. [6] FAME Laboratory, Department of Exercise Science, University of Thessaly, 42100 Trikala, Greece. [7] Department of Nutrition and Exercise Sciences, University of Copenhagen (NEXS), 2100 Copenhagen, Denmark. ✉email: david.garcia-leon@ec.europa.eu

Environmental factors have a clear influence on how humans behave and perform. Excessive heat has been shown to be an important negative externality with an effect on the productivity of workers[1–5]. Excessively hot environments are precursors of biophysical and cognitive impacts, causing physiological strain to workers[6], lowering the number of hours of work supplied[7], affecting the capacity of assimilating information[8] and interfering with decision-making[9], ultimately undermining human capital accumulation and, therefore, economic growth. In a context of rising temperatures, quantifying the economic impact of these externalities with spatially resolved socioeconomic data and models is key to combat their effect, acting as the necessary input for the design of evidence-based adaptation plans and occupational health policies[10].

The number of days exceeding the 90th percentile threshold (baseline period, 1970–2000) have doubled between 1960 and 2017 across the European land area[11], largely attributed to human-induced climate change[12–14]. According to Stott et al.[15] and IPCC[16], it is likely that the human influence has more than doubled the risk of some past heatwaves, such as the 2003 European heatwave. Along with the proliferation of these extreme weather events, climate change projections show that they might become more frequent and to last longer across all Europe during the 21st century[16–18]. Therefore, extreme temperatures pose profound threats to future occupational health and labour productivity while exacerbating existing health problems in populations and introducing new health threats, such as heat exhaustion and heat stroke[19].

The use of bottom-up interdisciplinary approaches have gained importance in the assessment of climate risks[20–22]. Previous studies have already analysed the economic implications of heat-related labour productivity losses at different spatial and temporal scales. However, these studies have mainly focused on the effects of average temperatures rather than extreme heat. Orlov et al.[23] analyse the effects of past heatwaves in Europe but do no characterise the extent and duration of extreme heat episodes. Knittel et al.[24] and Orlov et al.[25] study the projected climate change effects of labour productivity in Germany and globally respectively, but they only account for the projected average temperature conditions at the workplace.

In this study, we comprehensively analyse the present and future economic damages due to reduced labour productivity caused by extreme heat in Europe. We do so by carefully identifying past and projected heatwaves and by adopting an unprecedented level of spatial, temporal and sectoral detail. Based on hourly climate reanalysis data (ERA5-Land) and using the Wet Bulb Globe Temperature (WBGT) as our reference heat stress index[5], we integrated sector-specific estimates of heat-induced productivity losses during heatwaves in 274 European regions into a regionalised general equilibrium economic model[26]. This allowed us to quantify the economy-wide effects of excessive heat while disentangling the associated direct and indirect economic impacts as well as the mechanisms of impact propagation. More details of our analytic approach and studied area are documented in the Methods. We then applied this model to a high emission scenario represented by two climate model simulations forced by greenhouse gases emissions following the Representative Concentration Pathway 8.5 (RCP8.5, thereafter) over the years 2035–2064. This mid-21st century period offers a good balance between foresight and uncertainty. On the one hand, it is less affected by uncertainties associated with the internal natural climate variability, which dominate for near-term projections, thus allowing the emergence of signals. On the other hand, this time period is subject to less uncertainty associated to mitigation pathways than late-21st century periods, as the latter uncertainty increases constantly over time[27,28]. Finally, we discuss the implications of the results and identify potential avenues for future research.

## Results

**Heat spreads unevenly across Europe.** Extreme hot spells in Europe varied greatly in frequency, duration, extension, and severity in the years analysed (2003, 2010, 2015, and 2018). Considering the 274 regions contained in our area of study (see the Methods for further details) and adopting the TX90p criterion, an average of $N = 1180$ (s.d.: ±230.2) regional heatwave events were identified per year. The temporal distribution pattern of the emergence of heatwaves was multimodal, with peaks arising as a result of high-amplitude heat episodes (Fig. 1a, left panel). The case of 2010 is particularly illustrative, with a large heatwave sweeping the continent at the end of June (Fig. 1a). The duration of the median heatwave ranged from 5 to 6 days, with 2003 and 2018 being the years showing events with higher mean duration (8.20 and 8.24 days, respectively; Fig. 1a, right panel). Years 2003 and 2018 featured the events with the longest duration (Fig. 1a), with a small fraction of heatwaves surpassing two weeks. Most heatwaves were concentrated during the summer months (June, July and August; JJA henceforth), but extended before and after this time frame, particularly in 2003 and 2018 (Fig. 1a, b). However, heatwaves initiated during summer were on average two times longer (8.5 versus 4.3 days) and more severe than non-summer heatwaves.

The total European area affected by heatwaves varies according to the time of the year analysed (Fig. 1b). The average heatwave affected 27–38% of the European territory, a situation that becomes aggravated during summer months, with an average spatial extension of 49% of the total number of regions studied and a maximum coverage of more than 95% during large-scale episodes. Years 2003 and 2018 showed events with higher spatial coverage, with a summer average of around 55% of the total European area. From a regional perspective, heatwaves in 2003 concentrated in central Europe, affecting mainly regions of France, Germany, and Italy (Supplementary Table 1). The summer of 2010 showed less heat exposure in Western Europe, affecting primarily Eastern Europe and Russia[29]. Southern Europe and the Baltic countries experienced more frequent heatwaves in 2015. In contrast, 2018 was exceptionally hot in regions where heatwaves are typically less frequent (see also WMO[30]). During that year, Northern Europe, in particular Scandinavian countries, experienced sustained positive temperature anomalies, which added up to around 2 calendar months.

It is important here to distinguish between the duration and the severity of a heatwave. Both elements are relevant for the determination of productivity losses, but the latter is essential, as the physiological effects of heat on workers usually emerge above the WBGT threshold of 26 °C[31]. To illustrate heatwave severity, we adapted the concept of heating degree-days of ref. [32], generally used to calculate total energy demand. A Wet Bulb Degree-Day (WBDD) is here defined as any additional Wet-Bulb degree over 26 °C experienced by a worker under heatwave days, considering only working hours. Total outdoor WBDD per year and region are shown in Fig. 1c, reflecting that southern regions tend to always suffer from high cumulative heat stress, even when heat patterns become more intensified in other latitudes (see also Supplementary Fig. 1). Our analysis of regional heatwaves shows that these events are largely heterogeneous in terms of spatial and temporal characteristics (see, for example, Northern Italy in 2003 or Croatia in 2003 and 2015 in Fig. 1c). This underpins the importance of using local and timely data as well as high-resolution economic tools when it comes to analyse the impacts of heatwaves and other related extreme weather events.

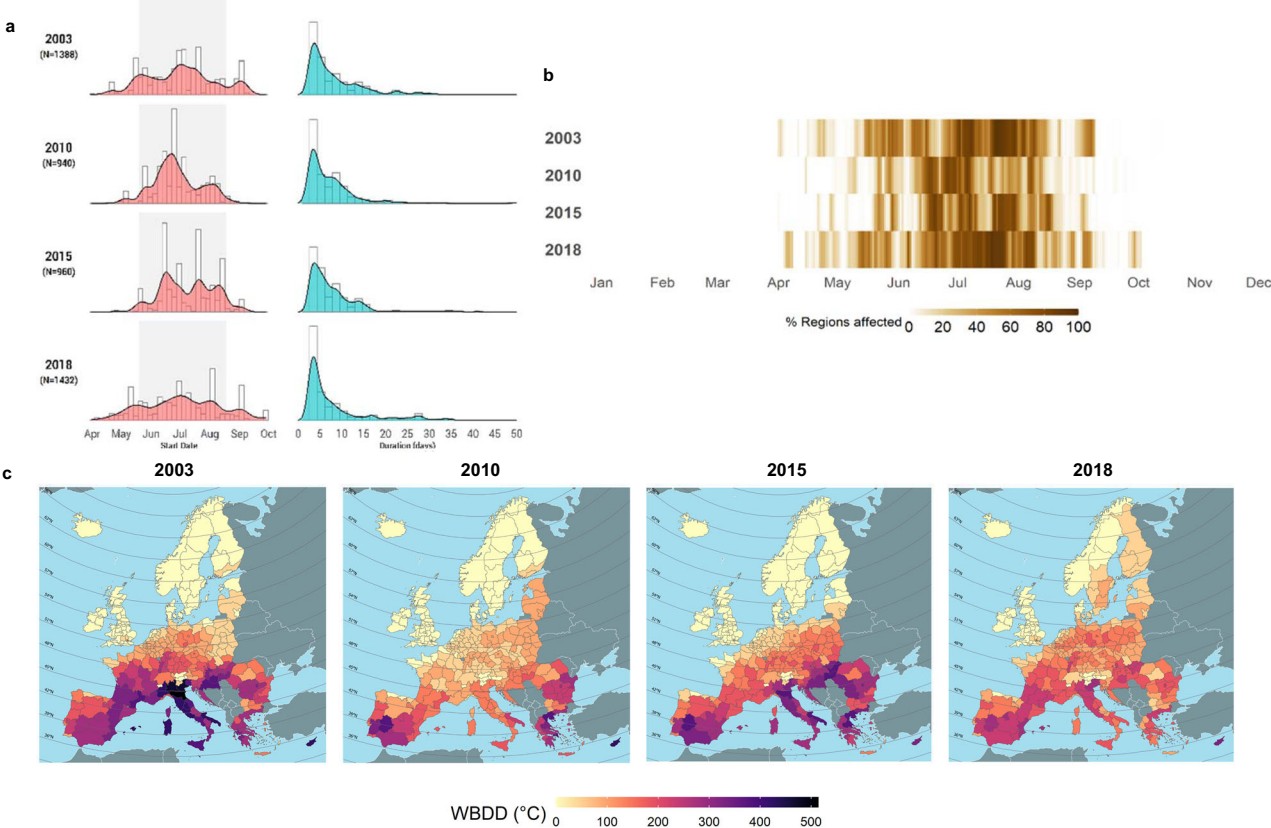

**Fig. 1 Characteristics of detected heatwaves. a** Start date (left) and duration (in days, right) of the identified events at the regional level in the four years analysed, where N denotes the number of events identified each year. Grey-shaded areas in the left panel correspond to summer months (JJA). **b** Total area (as percentage of total number of regions) affected by detected heatwaves (TX90p criterion) in the four years studied. **c** Annual cumulative observed heat during heatwaves in outdoor environments*, as obtained from the sun version of WBGT. *A Wet Bulb Degree Day (WBDD) index was constructed based on the concept of Heating Degree Days (Spinoni et al., 2015) using a temperature threshold of 26 ºC.

**More exposed regions spearhead economic losses**. Economic losses triggered by heatwaves show great spatial heterogeneity, both between and within countries (Fig. 2a). During the analysed years, heatwaves undermined the economic growth of Europe in the range of 0.3%–0.5% of European GDP, 1.5 to 2.5 times more than during an average year (0.2% GDP losses experienced on average over the period 1981–2010 due to extreme heat). Regional damages were dependent on the extension, duration, and magnitude of the considered events as well as on the regional economic structure. We observe a north-south gradient in economic losses, consistent with average warmer temperatures and higher proportion of outdoor production in the southern part of Europe (Fig. 2b). Even when heat patterns were intensified to the north of the continent, as was the case in 2018, with observed impacts in Northern Europe well above their historical maxima, the economies of southern regions were always shown more affected relative to the size of their economies. Regional disparities in estimated damages are high, with several regions experiencing GDP losses beyond 1%, some over 1.5%, and a few over 2% (Fig. 3).

The analysis of economic losses by year reflects that most affected regions are those with either more heat prone environments or a more exposed economic structure, or a combination of the two. In general, observed greater damages are associated with a group of regions that we call 'fully exposed' regions (upper-right quadrant of Supplementary Fig. 2), that is, regions showing high average heat exposure and featuring a relatively large fraction of outdoor sectors. This group of regions shows on average twice the heat exposure and one and a half times the economic exposure of the remaining regions.

Our results suggest that, in present times, direct impacts of heat on labour productivity take place mostly in outdoor sectors. However, these losses propagate to the entire economy. This propagation takes place mainly through the mechanism of intermediate goods used in the production processes, for example, in services relying on agricultural and industrial products or transport services as inputs. Given the complementarity between primary and intermediate inputs, indirect effects spread substantially through the service sector. In contrast, trade mechanisms, i.e., trade between regions, act as a buffer to mitigate this negative effect by substituting intermediate goods from less affected regions. These two mechanisms are embedded into our economic model. The fact that we identify economic losses in most indoor sectors, suggest that the intermediate goods mechanism outweighs the mitigating effect of trade. In our analysis we predominantly identified direct impacts of heat on labour in outdoor sectors. This becomes clear by looking at the differences between the distributions of the sun and shade versions of WBGT (Supplementary Fig. 3), used for outdoor and indoor sectors, respectively. Under current climate conditions and, irrespectively from the exposure-response function used, heatwaves tend to show a strong impact on ambient exposed work, while indoor work remains hardly affected. The lack of solar radiation and the typically lower metabolic intensity of indoor jobs helps to protect further this group of workers.

Our results are qualitatively consistent with other studies dealing with labour impacts of excessive heat in Europe[23]. However, this work differs from the previous literature in several aspects. Among them, we study the impacts of extreme heat,

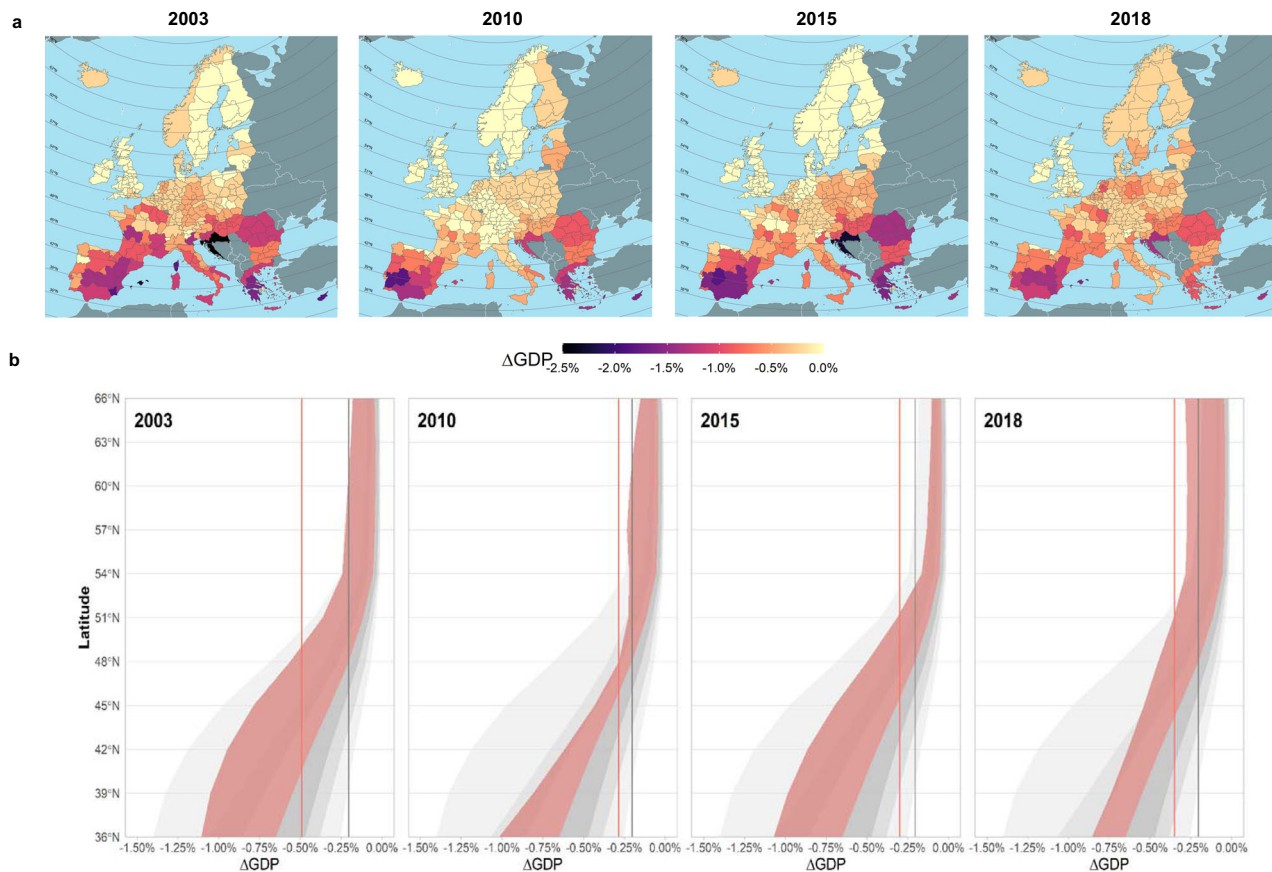

**Fig. 2 Regional economic impacts of heatwaves: spatial pattern. a** Regional-level cost of heatwaves (as a share of regional GDP) in the four years analysed. **b** Regional impacts of heatwaves at different latitudes. Vertical lines show the average, cross-regional, annual GDP impacts of heatwaves (solid red line) and the corresponding effect over the historical period 1981–2010 (solid grey line), as obtained from simulating all the years over the historical period (see Methods). Grey-shaded areas describe the distribution percentiles of damages (1st, 10th, 25th, 50th, 75th, and 99th percentiles) over 1981–2010. Red-shaded areas denote positive anomalies in economic damages compared to the historical median effect.

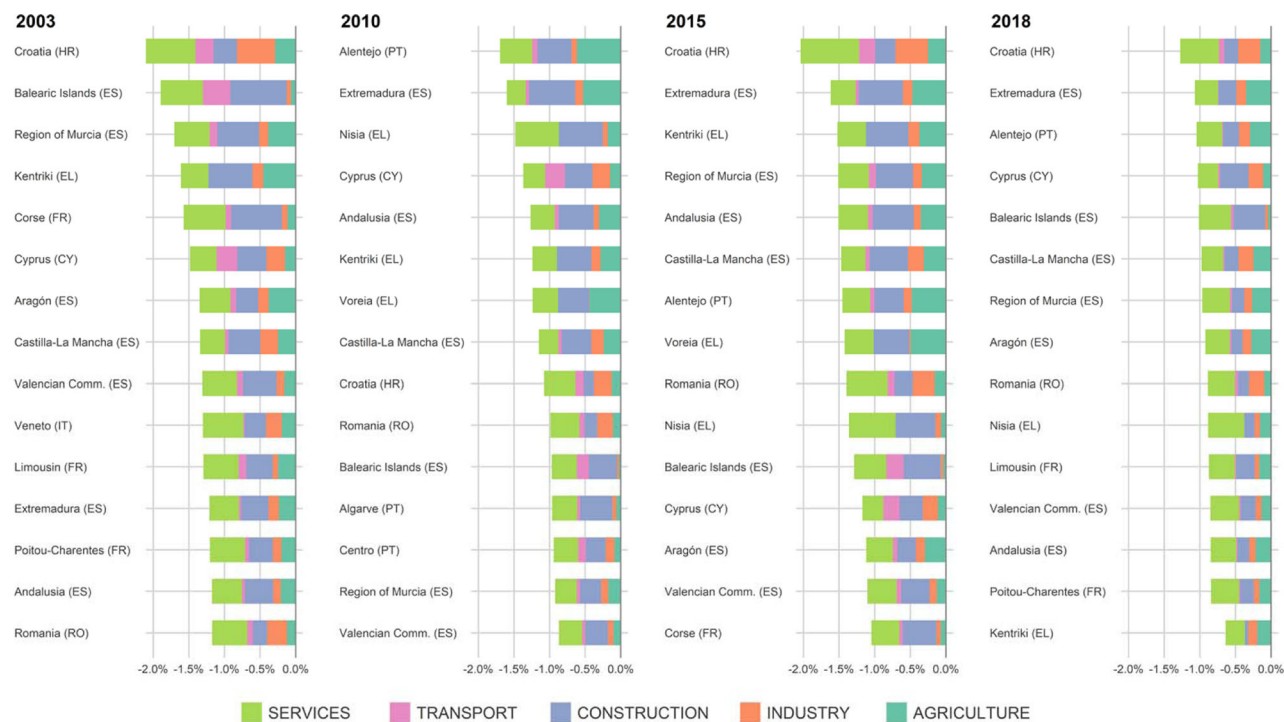

**Fig. 3 Regional economic impacts of heatwaves: sectoral composition.** Sectoral breakdown of damages in most affected regions each analysed year. It is shown the contribution of each sector (in percentage points) to the total percentage variation of the respective regional Gross Value Added (GVA).

understood as periods when a region's temperatures are abnormally high (rather than measuring the effect of summer average temperatures), consider all the productive economic sectors, and adopt a higher spatio-temporal resolution level. We also tested the sensitivity of our findings to the choice of different heat-exposure functions (see Methods, Heat exposure functions). Resulting differences responded to how different heat exposure functions were constructed and were proportional across the three considered approaches, producing on average 11% less damages in the case of NIOSH and 30% in the case of Hothaps compared to ISO standards (Supplementary Fig. 4).

There are a number of reasons that lead us to think of the reported costs here only as a lower bound of the actual labour-related economic damages caused by heatwaves. First, our estimates do not include the costs caused by heat-derived occupational injuries[33,34] and their related public health costs. Second, the calibration of the heat exposure functions is based on a few empirical studies, where production subsectors are treated homogeneously, thus not capturing well their workload diversity. On the opposite side, this study does not consider the likely existence of adaptation and heat-insulation measures adopted by companies and households, such as air conditioning, extensively implemented in Europe[35]. However, since the implementation of heat-insulation measures is still quite low in outdoor sectors and air conditioning availability only affects indoor sectors, these adaptation effects do likely not have a significant impact on our current estimates.

**Projected costs under unmitigated scenarios.** Our approach to future climate scenarios focuses on a 30-year period spanned by the years 2035–2064, for which the number and severity of regional heatwaves were calculated. Climate model projections are subject to large uncertainties regarding the model, initial conditions or emission scenario considered[36,37]. We sought to account for these uncertainties by analysing two different climate models, which are representative simulations of different warming conditions over Europe (see Methods) selected from a larger multi-model ensemble[5], and by focusing on a 30-year period by mid-century, less affected by the chosen emission scenario. In the two considered models, we observed an increase in heat severity for both outdoor and indoor workers[5]. In contrast to what is observed during current heatwaves, our projections of WBGT suggest that indoor workers will become more directly affected by heat, especially in regions from southern and central Europe, provided current working conditions and insulation methods prevail (Supplementary Fig. 3, see the projected shift in the WBGT distribution in red).

Heatwave-induced GDP losses in Europe are projected to grow steadily over the next 40 years (Fig. 4a). This finding is robust, irrespectively of the climate model considered. The identified trend is homogenous until 2050, where the signal of the two considered climate models starts to diverge. Average costs will pass from the current average of 0.21% over 1981–2010 to an expected annual average of 0.77% in 2035–2045 (s.d.: ±0.16%), to around 0.96% (s.d.: ±0.26%) in 2045–2055 and will go beyond 1.14% (s.d.: ±0.25%) in the 2060's, which denotes a steady increase in economic damages accompanied by an increase in their variability. As a matter of comparison, Orlov et al.[25] found that by 2050 Europe will experience economic losses in the range of 0.5%. We attribute this difference with respect to our findings

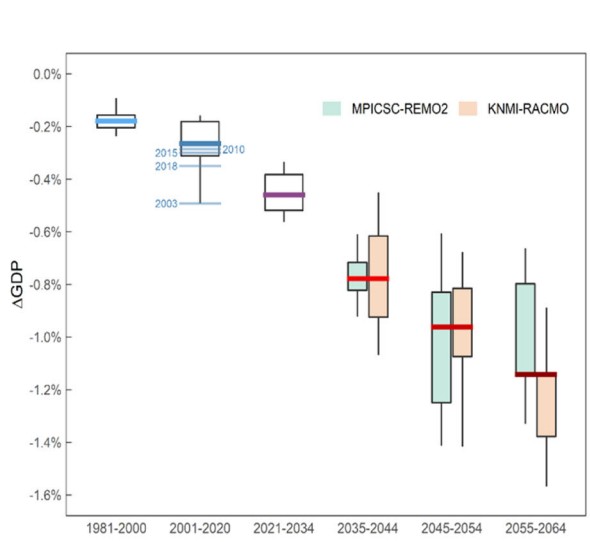

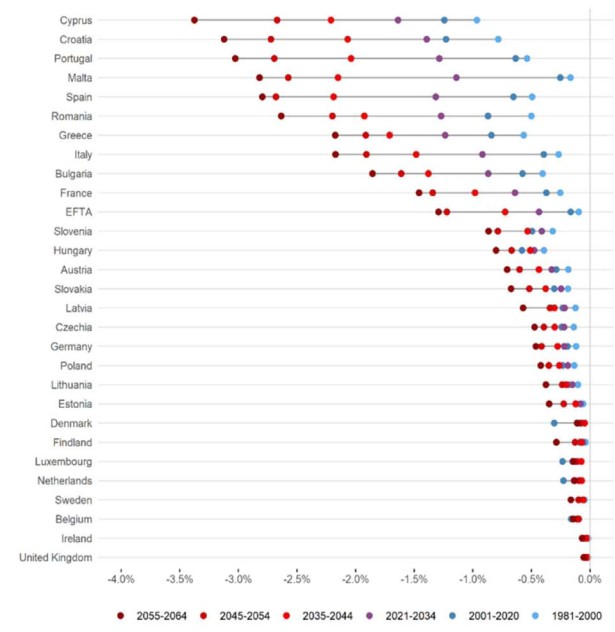

**Fig. 4 Climate change projected impacts. a** Estimates indicate that heatwave-induced total aggregated damages will grow steadily at the European level during the next four decades, peaking in the 2055–64 decade with annually expected GDP losses below −1.1%. This represents approximately a five-fold increase in losses, compared to what has been observed in the period 1981-2010. Each boxplot shows the interannual distribution of total European, annually estimated impacts over different time periods. In-depth analysed years (2003, 2010, 2015, and 2018) are highlighted. Boxes cover the interquartile range (IQR, 25th–75th percentiles) of the damage distribution and whiskers show the values contained within ±1.5·IQR. Thick solid lines denote the estimated median (multi-model median in the climate change analysis) GDP impact over each time period. Observations for the period 2001–2020 correspond to the annual simulations carried out over the period 2001–2010 together with the analysis of years 2015 and 2018. To simulate the economic model over the period 2021–2034, regional-level time series of labour productivity shocks were obtained by linearly interpolating average results over the historical period (1981–2010) and projected multi-model averages over the decade 2035–2044. **b** Holding fixed the current economic sectoral composition, heat damages will grow in all areas, especially in southern countries, more vulnerable to heatwaves due to their high heat and economic exposure (refer also to Supplementary Fig. 2). Each red-coloured dot represents the 10-year, multi-model average projected GDP impact for each country.

mainly to the heat-exposure function used (Hothaps in ref. [25]; ISO in this study), as ISO is more sensitive to lower temperatures (Supplementary Fig. 3; Supplementary Fig. 4) as well as to differences in the parametrisation of the economic model and the experimental design, as the cited work is based on a dynamic framework. Carefully exploring different climate and socio-economic (RCP-SSP) combinations is vital to assess the uncertainty posed by future projected scenarios to the present results (the reader is referred to the Supplementary Discussion for an overview of the projected outcomes under the scenario RCP8.5-SSP5).

Aggregating our regional estimates at the country level, we found that southern European countries will clearly be the most economically harmed by excessive heat in the future (Fig. 4b). Aside from Cyprus, the country with current and future highest relative losses, we identified that Portugal, Spain, and Croatia will gradually move from a range of losses of 2% in 2040 to around 3% in 2060. As it happens in present times, large heterogeneities in impacts are also expected within countries in this area, due to different regional heat and economic exposure (Supplementary Fig. 5). Balkan countries, Italy, and Greece will also experience considerable increases in their expected damages, with average annual losses of more than 2% in 2055–2064 attributed to extreme heat. Central and northern European countries will experience minor but significant negative impacts ($p$-value < 0.001), as thermal stress will increase across all latitudes. GDP impacts in those regions will be more modest but still meaningful, with Germany being projected to experience a negative impact on GDP of 0.5% by 2050, similar to what is reported in Knittel et al.[24], who also include the productivity effects of non-EU imports in their estimates. In contrast, UK, Iceland and Scandinavian countries will only suffer very mild losses, ranging from 0% to 0.2% of GDP.

The climate change signal provided by these two and other climate models for the next four decades is very similar, irrespectively of the emission scenario considered (see ref. [5] for the full ensemble of simulations; scenarios depict more distinct signals from 2050 onwards). Hence, heatwave-related productivity losses are expected to increase considerably in Europe by the mid-21st century even if stringent mitigation pathways were adopted. There exist, however, big margins for adaptation and acclimatisation, even though adapting to low-probability extreme events could challenge our potential adaptability[38].

## Discussion

At the European level, total annual losses attributable to heatwaves amounted to 0.3–0.5% of European GDP in the analysed years (2003, 2010, 2015, and 2018) while the average GDP loss over the period 1981–2010 was estimated to be close to 0.2%. Nevertheless, southern regions were consistently shown more vulnerable to these damages due to their high geographical and economic exposure to heat, experiencing GDP losses above 1% and, occasionally, above 2%. Under current climate conditions, outdoor workers seem disproportionately more affected by extreme heat, while most indoor work remains insulated. The analysis of the distribution of our heat stress measures (WBGT$_{sun}$ and WBGT$_{shade}$) suggests the presence of generalised and widespread outdoor productivity impacts but very mild indoor productivity damages only in southernmost regions. However, economic damages in outdoor activities spread to the remaining sectors due to the tight interconnectedness of the economic system. For example, sectors dependent on agricultural inputs, such as food manufacturing, tourism, and travel-related services, were shown largely affected. Meanwhile, bilateral trade between regions and input factor substitutability between sectors acted as smoothing factors to total damages. Looking ahead, the projected

costs of heatwaves are expected to increase steadily in the next decades if no further measures to adapt to warmer temperatures are implemented in work environments. Economic losses are projected to increase by almost a factor of five by 2060 compared to the historical damages experienced over the period 1981–2010 and will affect more the areas where heatwave-induced productivity damages are already pronounced.

We recognise the existence of additional factors that can play a role in a comprehensive assessment of the costs associated with excessive heat in work environments. First, heat-induced occupational injuries temporally shrink the effective labour force, harming total production while increasing public health costs. Occupational injuries data availability at the regional level, currently very limited, must be guaranteed for an effective integration of this dimension into our analysis. Second, the incremental heat effect in urban areas (see[39] and references therein), i.e., the urban heat island effect, and its potential large effects on, for example, construction workers, should be explored with the WBGT or alternative heat stress indicators including a human heat balance model, such as the Mean Radiant Temperature (MRT) and the Universal Thermal Climate Index (UTCI)[40], always adopting enough spatial resolution and/or parameterised processes to capture heat signals at the city level. Third, the presence of adaptation measures, such as air conditioning, may lessen heat-derived economic damages[41]. Its unlikely implementation in outdoor environments and the current absence of direct heat effects on indoor activities do likely not compromise our current estimates but can potentially affect our climate change results. Fourth, we used the existing state-of-the-art functions to translate heat stress into labour productivity impacts. Emerging field studies are beginning to provide evidence on the existence of heat stress impacts on productivity at temperatures below 26 °C WBGT[2,42,43], also featuring notable disparities within economic sub-sectors. Nevertheless, the results from these field studies have yet to be translated into more sophisticated, sector-specific transfer functions. Fifth, a comprehensive assessment based on the combination of future climate and socio-economic pathways must be carried out in order to obtain a complete overview of the distribution of future heatwave impacts and their associated uncertainties. However, for this kind of assessment, hourly-level and spatially downscaled projections of heat stress measures should first be available under different climate forcings and existing regional SSP narratives need to be able to accommodate changes in the regional economic structure[44]. These five gaps should be addressed by future research.

Beyond measuring the economic damages triggered by extreme heat, the proposed methodology can also be used as a tool for the assessment of future occupational health and the formulation of local-level adaptation policies. Finally, this study reinforces the need for spatially resolved, bottom-up approaches as a requisite to capture local socio-economic and climatic idiosyncrasies, crucial to analyse the potential economic consequences of climate change.

## Methods

**Area of study**. 274 regions representing all the EU-27 countries, United Kingdom and EFTA countries (Iceland, Norway, and Switzerland) were considered. Where available (EU-27 and UK), we adopted the second-level Nomenclature of Territorial Units for Statistics (NUTS 2) classification as the administrative basic unit of reference. Before feeding the model with the respective labour productivity shocks, a subsequent spatial aggregation procedure was required for some regions, since the resolution of the economic model was not the same for all countries because of the difficulty to estimate mutually consistent regional-level Social Accounting Matrices (SAMs) required to calibrate the economic model (refer to Supplementary Table 2 for a description of the spatial resolution used in the economic model). Typically, higher spatial disaggregation levels (NUTS 2) were available for more economically relevant countries (France, Italy, and Spain). NUTS 0 (country) or

NUTS 1 (sub-country) population-weighted spatial aggregation was applied to obtain the values in the remaining regions.

**Heatwaves characterisation.** There is no consistent and methodical approach to defining heatwaves (sometimes also referred to as warm spells)[45,46]. For example, the Intergovernmental Panel on Climate Change defines a heatwave as 'a period of abnormally hot weather'. Furthermore, several criteria have been used to characterise heatwaves based on mean, maximum, minimum temperature, humidity, or a combination of those[47]. In this study, we selected the TX90p criterion, i.e., a heatwave occurs when the 90th percentile of maximum temperatures is exceeded for at least 3 consecutive days. This criterion is based on the anomaly of maximum temperature and includes information about the entire annual cycle, which eases the identification of productivity impacts above a certain threshold of temperature. Using the period 1981–2010 as the reference to calculate temperature percentiles, we identified the number of heatwaves taking place in four years considered anomalously hot in Europe[29,48–50], namely, 2003, 2010, 2015, and 2018, and accounted for the duration, severity and spatial scope of these events.

**Heat stress index.** The Wet Bulb Globe Temperature (WBGT) was used as a heat stress index. WBGT is the most widely used index to determine heat stress in an occupational context (e.g., [23,51–53]), it can be easily obtained from standard meteorological variables and is recommend by the International Standard Organization as occupational heat stress index[54]. WBGT was calculated using the R package *HeatStress*[55] for both outdoor, i.e., WBGT in the sun (WBGT$_{sun}$), and indoor environments, i.e., WBGT in the shade (WBGT$_{shade}$). WBGT$_{sun}$[56] takes into account air temperature, dew point temperature, wind speed, and solar radiation, whereas WBGT$_{shade}$ is a simplified version based on air temperature and dew point temperature only[57], assuming a wind speed of 1 m/s (slow walk). For detailed information on the WBGT calculations and their formulations, the reader is referred to Lemke and Kjellstrom[58]. Hourly WBGT values were used to calculate the respective productivity losses due to heat exposure during working time. The use of hourly WBGT is essential to capture intra-daily heat variability, since the heat stress level encompasses the actual time devoted to work, avoiding the presence of potential biases resulting from the use of 24 h, day- or night-time temperature (e.g. [5], illustrate the clear underestimation of heat stress based on daily mean WBGT).

Measures to mitigate excessive heat include rescheduling tasks, increasing the number of breaks, or switching activities from outdoor to indoor environments. Although it is true that in some occupational settings the productivity losses triggered by excessive heat or the working time loss related to more frequent breaks may be reduced by rescheduling certain tasks[59], some of these countermeasures are already implemented in normal or warm days (and, therefore, workers cannot further change their behaviour during a heatwave) while other tasks need to be completed at specific hours of the day or location. Also, benefits from switching activities from sun to shadow are sometimes outweighed by aggravated effects of direct sunlight exposure[60]. Thus, the inherent flexibility of the WBGT, which allows to account for heat stress for either indoors or outdoors with a single index, entail an important advantage.

**Climate data.** For the analysis of past heatwave events, we used ERA5-Land reanalysis data[61], which provides hourly estimates of a large number of atmospheric climate variables on a high-resolution grid (0.1° × 0.1°; native resolution is 9 km). Daily maximum temperature from 1981 to present was used to account for past heatwaves. Heatwaves were identified at the regional level using the TX90p criterion, i.e., when the 90th percentile of the distribution of regional maximum temperatures spanned by data from the period 1981–2010 was exceeded for at least 3 consecutive days. Additionally, hourly values of near surface air temperature, dew point temperature, solar radiation, and wind speed data for the four selected years were retrieved to obtain hourly values of WBGT (see section above).

In the climate change exercise, we studied a period by mid-21st century (2035–2064), which is a compromise between using a not too distant future for immediate action and far enough so that the climate change signal emerges from the internal natural climate variability[62]. We assessed the evolution of heat stress costs based on two different climate model projections stemming from regional climate model (RCM) simulations from EURO-CORDEX[63,64] and gridded observational data (WFDEI, WATCH Forcing Data methodology applied to ERA-Interim data[65]), where the latter was necessary to establish the correction of systematic biases in the climate model data. For this purpose, we applied the empirical quantile mapping technique (QM[66]), using the implementation from Rajczak et al.[67]. QM was calibrated between the daily observed and modelled distributions of the input variables of the WBGT in the period 1981–2010 prior to the index calculation, resulting in bias-corrected projections of daily WBGT on a 0.5° grid (approximately 50 km). More methodological details and a comprehensive evaluation of the bias-corrected WBGT (in particular, the multivariate structure) is presented in Casanueva et al.[68]. Heatwaves over this period were estimated taking the historical period 1981–2010 as reference[69].

We focused on projections based on the emission scenario RCP8.5, as differences between the various scenarios were found to be small within the considered time horizon. In particular, we considered two specific simulations

which were chosen among a large ensemble of RCM simulations[5], namely, MPICSC-REMO2 driven by MPIESM (RCP8.5, EUR-44) and KNMI-RACMO driven by HADGEM (RCP8.5, EUR-11). They are representative simulations of the lower and the upper 25% of the distribution of climate change signals of summer mean temperature averaged across Europe in the period 2070–2099[5,70].

Hourly future values of WBGT were approximated from daily mean and maximum WBGT values based on the 4 + 4 + 4 approach[5,51]. Note that daily mean WBGT values were calculated from daily mean values of the considered input variables, whereas daily maximum WBGT values were estimated with daily maximum air temperature and solar radiation, and daily mean dew point temperature and wind speed. The 4 + 4 + 4 method assumes the daily maximum WBGT value during the hottest part of the day (12–16 h), the daily mean WBGT during 2 h in the early morning (8–10 h) and 2 h in the early evening (18–20 h), and the average of daily mean and daily maximum WBGT values during the remaining 4 h (10–12 h and 16–18 h).

**Heat exposure functions.** Heat stress experienced in different environments and metabolic loads were transformed into productivity losses using the ISO, NIOSH, and Hothaps heat exposure metrics[31]. The sun and shade versions of the WBGT were considered for outdoor and indoor activities, respectively. Economic sectors were classified into three different categories depending on their workload intensity (Supplementary Table 3). Low, moderate, and high workload groups were identified. Hourly work ability (workability) levels were computed and expressed in the range of 0–100%, where 100% represents no productivity damages and 0% denotes inability to work. Workability hourly values for the ISO and NIOSH approaches were obtained as

$$\text{workability}_{(\text{ISO,NIOSH}),h} = \max\left\{0; \min\left[1; \left(\frac{\text{WBGT}_{\text{lim,rest}} - \text{WBGT}_h}{\text{WBGT}_{\text{lim,rest}} - \text{WBGT}_{\text{lim}}}\right)\right]\right\} \quad (1)$$

where $h$ denotes the hour of day, $\text{WBGT}_{\text{lim,ISO}} = 34.9 - M/46$ and $\text{WBGT}_{\text{lim,NIOSH}} = 56.7 - 11.5 \log_{10} M$. $\text{WBGT}_{\text{lim,rest}}$ results from applying Eq. 1 to the resting metabolic rate $M_{\text{rest}} = 117 \text{W}$[54]. Selected values of $M$ for low, medium, and high workloads were $M = \{200, 300, 400\} \text{W}$, respectively.

Contrary to the workability estimates based on $\text{WBGT}_{\text{lim}}$, showing a lower limit of 0% (Eq. 1), the Hothaps approach[71] proposes a lower limit of 10%, i.e., working is possible for 6 min. within each hour even under extreme heat. Hothaps workability values were approximated with the following two-parameter logistic function

$$\text{workability}_{\text{Hothaps}} = 0.1 + 0.9/[1 + (\text{WBGT}/\alpha_1)^{\alpha_2}] \quad (2)$$

where the set of parameters $(\alpha_1, \alpha_2)$ were equal to (34.64, 22.72) for low, (32.93, 17.81) for moderate and (30.94, 16.64) for high workload, respectively. However, the Hothaps functions are subject to great parameter uncertainty due to being based on a few empirical studies. Therefore, we adopted the ISO standards as our benchmark functions and used the NIOSH and Hothaps functions to test for the sensitivity of our estimates.

A constant and homogenous working day was assumed to take place from 9 h to 17 h. Hourly workability values were averaged at the daily level for all days in a heatwave.

**Population data.** An effort was made to situate economic activity. Gridded data on heat-induced worker productivity losses were matched with population masks to obtain population-weighted impacts on worker productivity. We used the gridded UN WPP-adjusted population count data v4.11, provided by the Socioeconomic Data and Applications Center (SEDAC, Columbia University; CIESIN Center for International Earth Science Information Network[72]). Since only data for years 2000, 2005, 2010, 2015, 2020 were available, year-specific population masks were calculated by linear interpolation. The spatial mismatch between the resolution of population (0.25°) and climate data (0.1° for the reanalysis and 0.5° for the bias-corrected climate model data) were handled by interpolating the latter towards the population grid using the nearest neighbour. In the climate change analysis, we retrieved population data (0.25° resolution) for the years 2030, 2040, 2050, 2060, and 2070 under the scenario SSP5[73,74], denoting a situation of high energy and resource intensity based on fossil fuel development, consistent with the emission scenario RCP8.5. The same procedure of temporal interpolation of the population data and spatial interpolation of the climate data towards the population grid was applied.

**Accounting for seasonal patterns in economic activity.** Seasonal and calendar effects matter if an accurate economic measurement is sought. For example, heatwaves tend to concentrate during summer months while economic output generated by outdoor activities, such as agricultural work, is also predominant during that time of the year. We controlled for seasonality by weighting all the economic shocks by the number of heatwave days per quarter. We used economic data from the Quarterly National Accounts provided by Eurostat[75] to approximate the seasonal pattern of regional-sectoral activities.

**Regional computable general equilibrium (CGE) model**. The sub-national CGE model used is based on the GTAP model[76] and was calibrated on the GTAP 8 database and parameters for the reference year 2007[77]. The choice of the calibration year might bias to some extent the outcome of our simulations, as the regional database already incorporates the effect of heatwaves in the economy during the calibration year. We expect, however, this bias to be low in our case, as 2007 was in general a lower-than-average year in terms of heat load.

The GTAP database is a series of Social Accounting Matrices (SAMs) for 129 countries or groups of countries and 57 sectors covering the global economy. The maximum level of spatial detail in GTAP is the country level. For this reason, we complemented the GTAP database with regional economic information retrieved from Eurostat and countries' National Statistical Offices. We also followed a methodology based on Simple Location Quotients (SLQs) and gravity equations to get mutually consistent regional SAMs and bilateral trade flows between the EU sub-country regions[26].

The model follows the neoclassical paradigm, where investments are saving-driven and primary factors (labour and capital) are fully employed. More specifically, labour and capital can move between different economic sectors but not outside the region in which they are located. Production is represented by a Leontief technology between factors of production and value added, which is in turn a Constant Elasticity of Substitution (CES) function between the primary factors. When a shock hits the economic system, agents (households and firms) adjust their economic decisions (consumption, production, primary factors allocation) based on relative price changes.

One relevant feature of our sub-national version of the GTAP model concerns the specification of the trade relationships between regions. In CGE modelling, including the GTAP framework, the Armington assumption is typically used to model the trade structure. Armington elasticities imply an imperfect substitution between domestic and foreign products, which prevents an unrealistic sectoral specialisation after a shock being absorbed by the model. We developed the trade structure of the GTAP model regionally to disentangle the international and intra-national trade flows. Unlike the standard GTAP country-level specification, we include the domestic sub-national demand and the intra-national imports from other regions. We used two types of functions to model our trade structure. The CES function links the sub-national domestic demand and the aggregate imports of the sub-national region and uses an elasticity of substitution $\sigma_{ARM}$ between the two variables. The CRESH (Constant Ratios of Elasticity of Substitution, Homothetic[78]) function breaks the aggregate imports according to the source region, which can be a region within or outside the country. In this case, the elasticity is the bi-dimensional $\sigma_{IMP}$, which allows us to identify the source and the destination region and to differentiate between intra- and international trade. Compared to the standard GTAP model, we increase $\sigma_{IMP}$ by 20% if the region is trading with another region within the country. This calibration is based on the so-called *border effect*[79], which states that, *ceteris paribus*, trade between two locations is reduced by 20-50% if these two locations are separated by national borders[80]. We adopt a conservative choice on the value of Armington elasticities for sub-national units because we focus on the short-term economic consequences of heatwaves, when trade frictions can be high. This value is meant to provide only a reference to make trade more fluid within a country than between countries. Further research and a sensitivity analysis on this value would be certainly valuable but is out the scope of the paper.

Regional Value Added (*VA*) for region *r* and sector *s* is represented in the CGE model with a constant return to scale function of capital and labour as follows:

$$VA_{rs} = \left( \chi_{rs} K_{rs}^{\frac{\sigma_s - 1}{\sigma_s}} \psi_{rs} L_{rs}^{\frac{\sigma_s - 1}{\sigma_s}} \right)^{\frac{\sigma_s}{\sigma_s - 1}} \quad (3)$$

where the total value added generated by region *r* in sector *s* in a certain year depends on the combined use of capital (*K*) and labour (*L*), each of which show a specific degree of region-sector factor productivity, $\chi$ and $\psi$, respectively. The parameter $\sigma_S$ denotes the elasticity of substitution between primary factors.

**Coupling between biophysical impacts of heat and economic model**. Since the economic model shows a yearly temporal resolution, we transformed all the productivity shocks to their annual equivalents before simulating the model. During a given year, the productivity of labour in region *r* and sector *s* was assumed to be reduced by a percentage $\tau_{rs}$, that is,

$$\psi'_{rs} = (1 - \tau_{rs})\psi_{rs} \quad (4)$$

where the parameter $\tau$ represents the annual-equivalent average workability in region r and sector s. We calculated $\tau$ for each region-sector combination at each of the analysed years using MS Excel.

**Historical GDP losses**. For the sake of comparison, the distribution of historical economic losses in response to heatwaves was calculated for the period 1981–2010. Using historical maximum temperature records, regional heatwaves were identified along the whole reference period and their severity was documented. Regional-level, sun and shade WBGT values were calculated after retrieving the full time series of WBGT components from the ERA5-Land hourly data catalogue over the period 1981–2010. Since CIESIN population count data was not available for the

period 1981–1999, we used population data from year 2000 as weighting factor. The resulting WBGT values were used to obtain the respective regional-sectoral annual productivity losses and shocks, which were used to hit the economic model as described in the section above to derive the underlying regional economic losses.

**Reporting summary**. Further information on research design is available in the Nature Research Reporting Summary linked to this article.

## Data availability
The historical climate data that support the findings of this study are publicly available at the Copernicus Climate Data Store (https://doi.org/10.24381/cds.bd0915c6). Future climate projections are available for download via the Earth System Grid Federation (ESGF, https://esgf.llnl.gov/) under the project name "CORDEX" at any of the ESGF nodes, such as for example, https://esgf-node.ipsl.upmc.fr/search/cordex-ipsl/. The Social Accounting Matrices (GTAP) used to calibrate the economic model were used under license for the current study, and so are not publicly available. Data are however available from the authors upon reasonable request and with permission of the Center for Global Trade Analysis, Department of Agricultural Economics, Purdue University. UN WPP-Adjusted Population Count, v4.11 are available for download at https://sedac.ciesin.columbia.edu/data/set/gpw-v4-population-count-adjusted-to-2015-unwpp-country-totals-rev11. Quarterly sectoral accounts used in the seasonal adjustment of productivity shocks were obtained from Eurostat (https://ec.europa.eu/eurostat/databrowser/view/namq_10_gdp/default/table?lang=en).

## Code availability
The two versions of the WBGT used in this study were implemented in R using the R package HeatStress (https://github.com/anacv/HeatStress. https://doi.org/10.5281/zenodo.3264929), under license GPL-3.

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

## Acknowledgements

D.G.L. acknowledges financial support from the European Commission (H2020-MSCA-IF-2015) under REA grant agreement no. 705408. A.B., A.C., A.F., and L.N. received funding from the European Union's Horizon 2020 research and innovation program under the grant agreement no. 668786.

## Author contributions

D.G.L. conceived the study. D.G.L., A.C., A.F., L.N., and G.S. designed the experiments. D.G.L., A.B., A.C., and G.S. collected the data and ran the model simulations. D.G.L. wrote the manuscript with contributions from all co-authors.

## Competing interests

The authors declare no competing interests.
