## [Peer Review File · Nature Communications]

Current and projected regional economic impacts of heatwaves in EuropeREVIEWER COMMENTS

Reviewer #1 (Remarks to the Author):

Thank for an opportunity to review an article. This study assesses the impact of heat waves on workers' productivity and the economy in Europe. While the research topic is important, I feel the novelty of this study is limited and it just replicates existing studies with minor modifications. For example, Orlov et al. (2019) have already conducted a very similar study for the past heat waves. Knittel et al. (2020) also conducted a future projection focusing on the propagation effects. I believe that replication is an important part of scientific activities, but there should be some significant progresses compared to the existing literature to be published as an original article. Higher spatial and sectoral resolution may be one of the strong points of this study. The manuscript could be restructured focusing more on this point and could be submitted elsewhere.

[References]

Orlov, A., Sillmann, J., Aaheim, A. et al. Economic Losses of Heat-Induced Reductions in Outdoor Worker Productivity: a Case Study of Europe. *Economics of Disasters and Climate Change* 3, 191–211 (2019). <https://doi.org/10.1007/s41885-019-00044-0>

Knittel, N., Jury, M.W., Bednar-Friedl, B. et al. A global analysis of heat-related labour productivity losses under climate change—implications for Germany's foreign trade. *Climatic Change* (2020). <https://doi.org/10.1007/s10584-020-02661-1>

Reviewer #2 (Remarks to the Author):

This study investigates the impact of extreme heat on worker productivity and the associated declines in GDP for different European regions. The study uses regional economic information to derive localized impacts on GDP. Impacts are assessed for four recent heatwaves and for future heat extremes using climate model projections. While the study is generally relevant and interesting, it could take more advantage of the regionalization it includes. Moreover, I see some methodological issues that should be addressed.

Major comments:

1. From Figure 1c it seems that heat exposure in Southern Europe is high every year, independent of the spatial structure of the heatwave. Consequently, the economic loss due to extreme heat should already be included in the actual national GDP each year (because there is no "heat free" reference year). I think it would thus be worth to quantify this "background GDP loss" due to the average heat that is occurring every year in Europe and to compare it to the losses that are estimated for the four extreme years.

2. The study assumes that people work from 9-17 every day. I am no expert on this, but from my personal experience with farmers and construction workers, it seems that 9-17 is not really fitting to their working hours. Instead, they might work earlier in the morning or later in the afternoon/evening to avoid the extreme heat at noon. I understand that it might be difficult to get robust numbers for working hours, but it would already be interesting to see the effects of a break at noon (e.g. between 12-15) and to check if this could be an effective option to mitigate some of the negative heat impacts.

Another question connected to adaptation is how adequate sun WBGT is to estimate heat stress for outdoor workers. While it is true that they cannot avoid being in the sun for certain tasks, several tasks can also be carried out in the shade. In very hot weather they will likely try to work in the shade as much as possible. Using sun WBGT is thus rather giving an upper estimate of the economic heat impacts but is likely not representative for what the workers actually experience.

3. One novelty, particularly highlighted in the study, is the usage of data with high spatial resolution (using the NUTS-2 regions). I think that the study could take more advantage of this. The only comprehensive overview for the impacts on all regions is given in Figure 1 for WBDD. While this purely climatic indicator could even be obtained in higher resolution from ERA5 or

EURO-CORDEX data, the economic analysis only focuses on the most affected regions and lacks general information about the spatial variations of economic impacts. I would thus strongly advise to complement the study with a more comprehensive overview about regional economic impacts.

4. I would recommend to add some variability analysis to the study. The four selected heatwaves in the current climate already show quite substantial variations, and I would expect a similar picture for future climate projections. Additionally, I do not fully understand why averaged values for one decade were chosen for future projections. It should also be possible to perform the analysis for every year and thereby get an estimate of interannual variations (unless the economic modelling is too time consuming to be run for every year). In this way, it would also be possible to obtain a measure of uncertainty for the results both for current and future periods (connected to my point 1).

I am also wondering whether the difference in the methodology for present (using only heatwave days) and future (using JJA averages) periods impacts the results. By applying the same methodology to both time periods, biases connected to this could be excluded.

5. Bias correction for WBGT is performed using WFDEI data as reference. I think it would be more consistent to use ERA5-Land as reference dataset, since the results for current climate conditions are also obtained with this dataset. At the same time, I am not entirely convinced by using quantile mapping in combination with extreme heat data. Quantile mapping usually only adds constant values to extreme data (i.e., to values outside of the ones sampled in the reference period) and it can introduce artificial trends (Maraun 2016, doi:10.1007/s40641-016-0050-x). This could be prevented by using other methods such as quantile delta mapping (Cannon 2015, doi:10.1175/JCLI-D-14-00754.1) or multivariate bias correction (Cannon 2017, doi:10.1007/s00382-017-3580-6).

Other questions:

- Which are the assumptions for future economic development? Is it based on an SSP scenario? Or is the current state of the economy also used for future projections?
- The two models used for the future projections have two different spatial resolutions. Is this on purpose? Otherwise, I would rather choose models with the same resolution to avoid that this discrepancy influences the results.

I also think it would be more adequate to select the climate models based on the total model range in 2035-2064 (since that is the analysed period and the changes seen in RCP8.5 are unlikely to happen, especially towards the end of the 21st century).

Minor comments:

- Line 47: "Increasingly growing" says the same twice. "Increasing" or "growing" should be enough.
- Line 49: Change to "evidence-based"
- Line 51: What exactly is meant by "unusually warm days"?
- Lines 51-54: To me, the two statements about warm days and attribution of climate change do not really fit together and I think it would be better to have these in two different sentences.
- Line 56: I would suggest writing 21st century to be clear
- Line 59: Which health threats? Can they be named?
- Lines 64-68: I would split this sentence in two. Explaining the definition of TX90p splits the sentence pretty strongly, so that it is hard to understand that the second part actually refers to the TX90p criterion.
- Lines 64-71: Are all daily data of the 30 years pooled together? This should be stated somewhere (probably in the methods).
- Line 81: RCP8.5 is not a "business as usual" scenario, but – if at all – should be considered as "worst case" (see e.g. Hausfather and Peters, 2020, doi: 10.1038/d41586-020-00177-3).
- Lines 83-84: I think the explanation why this period is chosen should also be mentioned here and not just in the methods (I was at least wondering here why you chose it).
- Lines 87-88: It is not instantly clear to what "Its high spatial and temporal resolution" refers. I would advise to rephrase it to make it clearer.
- Lines 105-106: Did you test this statement?
- Lines 120-121: What does it mean that people suffered from heat anomalies for around 2 months? What is meant by "heat anomalies" and how did people "suffer"?

- Line 150: What is meant by “relevant way”? Why relevant?
- Line 151: What about primary inputs? Can they also be substituted in the model?
- Lines 154-155: Is it possible to separate the impacts on intermediate inputs and the trade effect to see how strongly trade can mitigate the negative effects on input?
- Line 157: What does “exposed environment” mean?
- Line 158: How high is the correlation? There are a few regions in Figure S2 that are high in both economic and heat exposure, but on average the correlation seems to be not so high (at least from visual inspection of Figure S2). I think it would be good to support this statement by some numbers.
- Lines 167-169: Is it possible to be more specific here? From a short look at the Orlov et al. study it seems that they estimated an impact of about 2-3\$ per capita relative to a GDP of about 2500\$ per capita. This rather suggests a decrease in the order of 0.1%. Does the difference between your study and Orlov et al. come from the fact that you also consider indirect effects? That should be mentioned.
- Line 198: What is meant here by “measures of heat”? WBDD?
- Lines 200-203: This information could be added to supplementary Figure S3 in addition to the WBGT distributions under current climate conditions.
- Line 217: I guess “Central and northern European countries”
- Lines 235-237: While this statement is true, it is not really novel.
- Line 238: This statement is rather trivial due to the seasonal cycle of temperature (in case daily values of the whole year were pooled to calculate TX90p).
- Line 253: Again, RCP8.5 is not “business as usual”.
- Line 263: I guess for investigating urban heat island effects, primarily models with higher spatial resolution are needed. And which specific heat stress indicators does this statement refer to?
- Line 276: Referring to one of my major comments, I think that the spatial resolution can be more exploited in this study, particularly because it is highlighted in several instances.
- Line 297: What is the reason that the resolution of the economic model is not the same? What determines its resolution?
- Line 300: What are the NUTS 0 and NUTS 1 regions? I think it would be good to shortly explain them here.
- Line 308: This ISO norm has been revised in 2017. I think the newest version should be checked and cited here (and in the following instances).
- Line 403: Does nearest neighbour interpolation conserve the total population? I am not entirely convinced that it is the best fit for this purpose. At the same time, indicating 15 arcmin as 0.25° would facilitate to see the resolution difference between both datasets.
- Line 406: Which resolution does this population dataset have?
- Lines 434-436: Does this apply for short- or long-term?
- Line 456: What is “IMP”?
- Line 458: What are ARM2 and IMP2 used for? For intranational trade?
- Line 460: Are the 20-50% reduction the reason why a factor of 1.5 is used in line 458? Moreover, I think it would be good to check if these numbers are still valid (the references are 20 years old). Especially in Europa I would expect that trade between different countries is high.

Figure 1:

- The abbreviation and meaning of NUTS should be explained somewhere in the caption or the main text (or it could be avoided in the main manuscript and only be mentioned in the methods).
- If I understand correctly, the units of WBDD is “°C days”. I think it would be good to add this unit to the colorbar in subplot c.

Figure 2:

- There should be enough space to write the full names of the analysed sectors in subplot b instead of the abbreviations.
- It would be nice to highlight the selected figures in Figure S2.

Figure 3:

- I think it would be feasible to show all countries here.
- I think it would be better to not use country codes but the full country names, so that readers can identify the countries more easily.

Response to reviewers' comments

We highly appreciate all the constructive comments and suggestions from both reviewers, which have contributed substantially to improve the quality of our paper. We have tried our best to revise the manuscript. The following are our point-to-point responses to the reviewers' comments.

▪ To Reviewer #1's comments

1. Thank for an opportunity to review an article. This study assesses the impact of heat waves on workers' productivity and the economy in Europe. While the research topic is important, I feel the novelty of this study is limited and it just replicates existing studies with minor modifications. For example, Orlov et al. (2019) have already conducted a very similar study for the past heat waves. Knittel et al. (2020) also conducted a future projection focusing on the propagation effects. I believe that replication is an important part of scientific activities, but there should be some significant progresses compared to the existing literature to be published as an original article.

RESPONSE:

We appreciate that this reviewer finds the topic of our paper relevant. In agreement with the reviewer, we indeed find merit in the work conducted by Orlov et al. (2019), the recently released Orlov et al. (2020) as well as in Knittel et al. (2020). However, we believe our work provides a substantial advance in the methodology, detail and scope of the analyses delivered. Next, we highlight the main features of our study and the points of departure from the referred works:

i) Orlov et al. (2019) report analyses of heatwaves impacts in Europe in 2003, 2010 and 2015, but they in fact look only at certain months of the year (*"Our analysis focuses on the strongest heat waves that occurred in Europe in August of 2003, July of 2010, and July of 2015"*).

Since a heatwave characterisation step is absent from their analysis, Orlov and colleagues end up measuring the average effect of temperatures on the

productivity of workers in some relatively warm months. Of interest and importance, but not of the same nature as the events analysed in the present work.

In our methodology, a careful characterisation of heatwaves is present for the whole analysed years. Heat stress is accounted for only at instances of time where temperatures were above the 90th percentile of maximum temperatures for at least three consecutive days. As such, we study the impacts of heatwaves, understood as extreme weather events, caring for the additional effect of low probability deviations from the average of the climate distribution.

Other significant advancements compared to Orlov et al. (2019) are:

- We provide in a single paper a comprehensive analysis of present (current heatwaves) and future (climate change projections) impacts of this hazard on the European economy.
- We study heatwave events taking place at any time of the year. As evidenced by Fig.1a and Fig.1b this is quite relevant, since not only summer months experience heatwaves. This gets worse in a climate change context, partly due to changes in the standard deviation of the temperature distribution (Ballester et al. 2010).
- We analyse the year 2018, which showed a peculiar geographical pattern, with large temperature anomalies in Central Europe and Scandinavia.
- As highlighted by the reviewer, the spatio-temporal resolution of the climate data and economic model used are much finer in our case, enabling us to capture climatic heterogeneities as well as regional economic characteristics.
- We use ERA5-Land (climate) hourly data, as opposed to the 4 hour time-snapshots used by Orlov (“we select the measurements at 9 am, 12 noon, 3 pm, and 6 pm”), which lets us account for intra-daily temperature variation and to estimate with precision the heat stress experienced during working hours.
- They focus on the agricultural and construction sector while we consider the whole economy. This feature is especially relevant for the climate change analysis.

- We analyse and compare three different heat exposure-response functions.
- Using quarterly accounts, we attribute economic activity to the time of the year (quarter) this activity takes place, resulting in a more precise impact measurement.
- We provide a regional vulnerability assessment to heatwaves by considering environmental and economic exposure to extreme heat.

ii) Orlov et al. (2020). The same authors have recently released a related analysis, which extends the approach of the former to a global-scale analysis of climate change impacts. In this work, the authors look at the effects of the generalised expected increase in average temperature on worker productivity. Meanwhile, we stick to the analysis of extreme temperature events.

While global climate models (with horizontal spatial resolution of 100-200km, approximately) are able to reproduce the main features of the climate system and are thus frequently used in global-scale analyses (as Orlov et al. 2020), we make use of regional climate models (with horizontal spatial resolution of 12km and 50km, approximately), which add valuable information with respect to the global counterparts due to more detailed spatial patterns and the better representation of local processes (e.g. Maraun et al. 2010). In spite of using the same observational dataset to bias-correct the climate model simulations (WFDEI, at approximately 50km), the added value of the regional climate models remains due to the more accurate representation of the temperature-humidity relationships in the uncorrected data (Casanueva et al. 2019).

Their approach has also some implications in how heat-induced economic shocks are modelled. In this respect, the way economic shocks are conceived (non-foreseeable, non-insurable) will apply better to the notion of a one-off heatwave shock rather than an average (structural) increase in temperature. Hence, our implementation of the shock makes more sense in economic modelling terms within the framework of Computable General Equilibrium Models.

iii) Knittel et al. (2020) investigate climate change impacts of heat-related labour productivity losses under climate change transferred via foreign trade to

Germany. As in Orlov et al. (2019, 2020), the authors look only at the effects of average temperatures. Additionally, the geographical and temporal scope of the analyses and the spatio-temporal resolution of the data used are notably different in Knittel et al. and in our study.

2. Higher spatial and sectoral resolution may be one of the strong points of this study. The manuscript could be restructured focusing more on this point and could be submitted elsewhere.

RESPONSE:

We agree with reviewer #1 in that high spatial, temporal and sectoral resolution is one of the strong points of the study. In particular,

- Higher sectoral resolution means becoming comprehensive by accounting for all the effects in the economy.
- Higher spatial resolution accounts for within country climate and economic heterogeneity (see Figure R1).
- Higher temporal resolution means to account for intra-daily temperature variation.

Figure R1: GDP impacts of heatwaves in 2003 assuming country (left) and regional spatial aggregation scheme (right).

All the above features result in increasing the accuracy with which heatwaves effects are measured. However, as reviewer #1 (and reviewer #2) have correctly pointed out, this aspect was not sufficiently emphasised in the former manuscript.

In the revised version, we have adequately addressed this issue, by for example, applying a spatio-temporal variability analysis to current heatwaves impacts (compared to historical impacts), as illustrated in Fig.2a and analysed in the main text. The title has also been changed, trying to highlight the regional scope of the paper.

The breakdown of sectoral impacts was already present in the previous version of the manuscript (see for instance Fig.2b and its analysis in lines 149-167 of the former version of the manuscript).

▪ To Reviewer #2's comments

This study investigates the impact of extreme heat on worker productivity and the associated declines in GDP for different European regions. The study uses regional economic information to derive localized impacts on GDP. Impacts are assessed for four recent heatwaves and for future heat extremes using climate model projections. While the study is generally relevant and interesting, it could take more advantage of the regionalization it includes. Moreover, I see some methodological issues that should be addressed.

RESPONSE:

Thanks to the reviewer for the positive and constructive comments. We appreciate their thorough and careful revision of our paper. We tried our best to revise our work and addressed all comments as follows:

Major comments:

1. From Figure 1c it seems that heat exposure in Southern Europe is high every year, independent of the spatial structure of the heatwave. Consequently, the economic loss due to extreme heat should already be included in the actual national GDP each year (because there is no “heat free” reference year). I think it would thus be worth to quantify this “background GDP loss” due to the average heat that is occurring every year in Europe and to compare it to the losses that are estimated for the four extreme years.

RESPONSE:

The reviewer makes an exceptional point here. We find very pertinent the suggestion to calculate a benchmark/historical GDP loss. In this way, the departures in losses from normal/average conditions could be better framed and quantified.

To do so, we have characterised the total number of heatwaves (and their duration) experienced by each region over the period 1981-2010 using the TX90p criterion. Given the computational cost of processing the WBGT (indoor and

outdoor versions with ERA5 hourly data) for the whole historical period, we have assumed a linear relationship between TX and WBGT and have obtained the effective TX during heatwaves in years 2003, 2010, 2015 and 2018 and their corresponding WBGT values. A theoretical WBGT has been then constructed by applying a linear fit between recorded TX and measured WBGT (Figure R2 corroborates that a linear fit can yield a good approximation of the relation between TX and WBGT). A historical productivity shock was estimated for each region using the average number of days under a heatwave over 1981-2010 and the previously estimated WBGT. The economic model was simulated using this theoretical shock to obtain a benchmark yearly loss representative from the historical period.

Figure R2: Scatterplot and linear fit between regional average TX (X-axis) and WBGT outdoor (Y-axis) experienced during heatwaves in 2003.

An analysis of GDP loss anomalies with respect to the historical period is now available at Fig.2a (see Figure R3 below) and several parts of the main text. A dedicated section describing the estimation of the GDP benchmark loss has been included in the methods.

Figure R3: New Fig.2a in the revised manuscript. Annual losses are compared to the benchmark results at different levels of latitude, exploiting in this way the rich spatial resolution of our climate and economic data.

2. a. The study assumes that people work from 9-17 every day. I am no expert on this, but from my personal experience with farmers and construction workers, it seems that 9-17 is not really fitting to their working hours. Instead, they might work earlier in the morning or later in the afternoon/evening to avoid the extreme heat at noon. I understand that it might be difficult to get robust numbers for working hours, but it would already be interesting to see the effects of a break at noon (e.g. between 12-15) and to check if this could be an effective option to mitigate some of the negative heat impacts.
- b. Another question connected to adaptation is how adequate sun WBGT is to estimate heat stress for outdoor workers. While it is true that they cannot avoid being in the sun for certain tasks, several tasks can also be carried out in the shade. In very hot weather they will likely try to work in the shade as much as possible. Using sun WBGT is thus rather giving an upper estimate of the economic heat impacts but is likely not representative for what the workers actually experience.

RESPONSE:

2.a: We acknowledge that the suggestions provided by the reviewer can be indeed of great interest as adaptation options can help to mitigate heat stress and counteract health impacts and productivity losses. In the manuscript we also refer

to other adaptation options, such as a generalised implementation of air conditioning in indoor environments as plausible adaptation measures. Other options include, a shift of working hours to cooler periods of the day (Day et al. 2019) or the opportunity to adjust clothing (Morgan and de Dear 2003), but also changes in the ability of people to cope with heat stress (Kjellstrom et al. 2016) or the mechanisation of work. Regarding the example the reviewer is highlighting, Takakura et al. (2017) already provide a precise estimate of the effect of worker breaks associated to climate change.

However, since these measures are not widespread (or already part of normal work procedures - hence not possible to reduce exposure during a heatwave), we consider that it would be inappropriate to incorporate them into our methodology, as they would mask the actual costs attributed to heatwaves.

2.b: We agree that some tasks may be rearranged while others, as emphasised above and recognised by the reviewer, cannot be moved from sun to shade. However, when the overall impact in major industries is evaluated, the benefits from reduced work-loss by rearranging tasks may be sometimes outweighed by workers being more affected by solar radiation than already predicted by the WBGT-model (see, for example, Piil et al. 2020). Productivity losses related to seeking shadow during additional (planned or unplanned) breaks is also a factor that may outweigh the benefits when lost work time or efficiency is evaluated across large outdoor industries. In summary, we believe that the WBGT offers a fair approximation to the actual heat stress experienced by workers under current arrangements. We acknowledge, however, the caveats raised by the reviewer. To reflect these points, we now have added the below paragraph in the methods section.

Changes in the manuscript (Line 349): *“Measures to mitigate excessive heat include rescheduling tasks, increasing the number of breaks or switching activities from outdoor to indoor environments. Although it is true that in some occupational settings the productivity losses triggered by excessive heat or the working time loss related to more frequent breaks may be reduced by rescheduling certain tasks (Morabito et al., 2020), some of these countermeasures are already implemented in normal or warm days (and therefore workers cannot further change their behaviour during a heatwave) while*

other tasks need to be completed at specific hours of the day or location. Also, benefits from switching activities from sun to shadow are sometimes outweighed by aggravated effects of direct sunlight exposure (Piil et al., 2020). Thus, the signal captured by the WBGT represents a fair proxy of the heat stress experienced by the worker in both outdoor and indoor environments”.

3. One novelty, particularly highlighted in the study, is the usage of data with high spatial resolution (using the NUTS-2 regions). I think that the study could take more advantage of this. The only comprehensive overview for the impacts on all regions is given in Figure 1 for WBDD. While this purely climatic indicator could even be obtained in higher resolution from ERA5 or EURO-CORDEX data, the economic analysis only focuses on the most affected regions and lacks general information about the spatial variations of economic impacts. I would thus strongly advice to complement the study with a more comprehensive overview about regional economic impacts.

RESPONSE:

We agree with the point raised by the reviewer. Revised Fig.2a (Figure R3 here) looks to take benefit of the spatial richness of our dataset and economic model.

To convey the spatial heterogeneity of damages, we have come up with a diagram that takes spatial slices across equally distanced ranges of latitude. A North-South gradient in damages is visually evident from the graph, resulting from southern regions being generally warmer and more intensive in vulnerable sectors.

This new graph incorporates the benchmark GDP loss, also split by latitudes. In this way, yearly spatial anomalies can be also analysed. Finally, it also let us compare more efficiently the spatio-temporal variability of heatwaves, as requested in comment #4.a.

4. a. I would recommend to add some variability analysis to the study. The four selected heatwaves in the current climate already show quite substantial variations, and I would expect a similar picture for future climate projections.

b. Additionally, I do not fully understand why averaged values for one decade were chosen for future projections. It should also be possible to perform the analysis for every year and thereby get an estimate of interannual variations (unless the economic modelling is too time consuming to be run for every year). In this way, it would also be possible to obtain a measure of uncertainty for the results both for current and future periods (connected to my point 1).

c. I am also wondering whether the difference in the methodology for present (using only heatwave days) and future (using JJA averages) periods impacts the results. By applying the same methodology to both time periods, biases connected to this could be excluded.

RESPONSE:

4.a: We have taken the reviewer's recommendation. As noted in the previous comment, we have included a spatio-temporal analysis of historical events, as we have been able to synthesise all the analysis in a single graph.

Modifications to the climate change analysis are addressed in 4.b and 4.c.

4.b: The reviewer is right in that year-to-year climatic variations are not negligible, nor is climate model uncertainty. It is then worth analysing the range of outcomes derived from each year-model realisation. In the revised version, we have simulated the climate-economic set of models for each year in the interval 2035-2064. The results are analysed in the revised Fig.3a (Figure R4 in this document). We have preserved the decadal perspective, but we present all the range of outcomes for each climate model, evidencing both the climate model variability and the climatic interannual variability.

Figure R4: New Fig.3a in the revised manuscript. Decadal and model variability are shown as well as current/historical average impacts.

4.c: This comment has been quite enlightening, so we are very grateful to the reviewer. As the reviewer correctly remarks, applying different methodologies for current and future analyses was adding noise (uncertainty/biases) and confusion to the paper. We have revised how future damages were calculated. Using historical (1981-2010) TX from MPICSC-REMO2 and KNMI-RACMO models as reference and applying the TX90p criterion, we have characterised the expected regional heatwaves (for each model) in the time interval 2035-2064. For this time period, the whole time series of hourly WBGT (indoor and outdoor) was already available, so we used these data directly in the analysis (avoiding the regression approach of comment #1). Essentially, we replicate in the climate change scenario, the identification strategy adopted in the historical years analysed.

Climatologically speaking, it would make more sense taking the period 2035-2064 as reference to obtain the reference percentiles and, thus, check for the existence of a heatwave, but as a matter of comparison and to account for the physical 26°C threshold, we found more appropriate to stick to the period 1981-2010 (Dosio et al. 2018).

We believe that, with this modification, our paper is now more consistent, as the identification strategy is homogenous, regardless the time period analysed. As a result, since we are now studying heatwaves impacts in the present time and in

the future, we have proposed a new title for the paper, which now would read as “Current and projected regional economic impacts of heatwaves in Europe”.

5. Bias correction for WBGT is performed using WFDEI data as reference. I think it would be more consistent to use ERA5-Land as reference dataset, since the results for current climate conditions are also obtained with this dataset. At the same time, I am not entirely convinced by using quantile mapping in combination with extreme heat data. Quantile mapping usually only adds constant values to extreme data (i.e., to values outside of the ones sampled in the reference period) and it can introduce artificial trends (Maraun 2016, doi:10.1007/s40641-016-0050-x). This could be prevented by using other methods such as quantile delta mapping (Cannon 2015, doi:10.1175/JCLI-D-14-00754.1) or multivariate bias correction (Cannon 2017, doi:10.1007/s00382-017-3580-6).

RESPONSE:

Thanks for this comment, where a number of interesting issues were mentioned. Regarding the **observational reference**, WFDEI has been and is still commonly used in climate model evaluation and bias correction. By the time we developed the climate projection analysis, WFDEI was the only available dataset for all the required variables and had the advantage of having a similar spatial resolution to the EURO-CORDEX simulations developed on the 0.44° grid (i.e. bias correction does not introduce uncertainties related to the scale mismatch, known as downscaling effects, see Casanueva et al. 2020a).

EURO-CORDEX simulations on higher resolution (0.11°, here KNMI-RACMO driven by HADGEM) were conservatively remapped to the WFDEI grid (0.5°). This way, some details (related to better resolved local processes) from the high resolution cannot be discerned, but some indication is still present after smoothing them onto a coarser resolution (Casanueva et al. 2016). Note also that climate models skilful scale is anyways coarser than grid spacing (Grasso 2000) and using a much higher resolution dataset for bias correction (such as ERA5-Land) might introduce statistical artefacts (Maraun 2013).

For all the above reasons, we considered that using WFDEI for bias correction was a reasonable approach and avoided further uncertainties in the bias correction process (downscaling effect). ERA5-Land was still the preferred observational dataset to quantify present climate heatwaves since, among other reasons, allow to assess years 2015 and 2018 (WFDEI extends until 2012).

We acknowledge that the observational dataset used for bias correction constitutes one source of uncertainty, being usually more critical for precipitation than temperature and larger for projected values than for projected changes (Casanueva et al. 2020a). In this sense, our economic results are presented as changes relative to a reference scenario under current climate conditions, thus the effect partly cancels out. Note that the selected bias correction method builds the correction function based on empirical percentiles from the observed and modelled distributions, therefore observational uncertainty would be small as long as the distributions of the two observations match well (see Figure R5).

Figure R5: Q-Q plot for ERA5-Land vs. WFDEI percentiles of daily maximum temperature in the period 1981-2010 for five grid boxes over exemplary European cities. For this, ERA5-Land data were conservatively remapped onto the WFDEI grid.

Regarding the selection of the **bias correction method**, the present work builds upon previous works which show a thoughtful evaluation of the models multivariate structure after empirical quantile mapping for the specific case of the input variables of the WBGT (Casanueva et al. 2019) and bias-corrected climate projections of a larger and state-of-the-art ensemble of regional climate models (Casanueva et al. 2020b). The use of other methods, as those pointed out by the reviewer, would need extensive evaluations in the context of WBGT. Trend-preservation of the climate change signals constitutes a continuous debate. Modifications of the simulated changes might be advantageous in case of

stationary, intensity-dependent biases (Gobiet et al., 2015, Ivanov et al., 2018) although they might not be justified in case of credible raw signals. Among the different possible extrapolation techniques, constant extrapolation was identified as a more robust approach than other more sophisticated approaches and it is the usual methodology (Déqué 2007, Themeßl et al. 2012, Gutiérrez et al. 2019). Note also that we focus on a period by mid-century, for which the number of new extremes with respect to the present climate (i.e. values estimated by extrapolation) is limited. Another aspect which is often overlooked is that trend-preserving methods, such as quantile delta mapping, rely to a greater extent on the distribution of the observational dataset (Casanueva et al. 2020a), since the simulated signal is transferred to the observations to generate pseudo-future observations, to which the quantile mapping is applied. All in all, we think that the use of empirical quantile mapping is well supported by the previous literature. It might be certainly interesting to assess the sensitivity of the economic projections to the bias correction method in future works.

Other questions:

6. Which are the assumptions for future economic development? Is it based on an SSP scenario? Or is the current state of the economy also used for future projections?

RESPONSE:

Future climate data is weighted, before being aggregated to the regional level, by population projections under the projected socioeconomic scenario (SSP5), consistent with the emission scenario RCP8.5, by which the climate models used are forced.

Meanwhile, the economic model (economic structure) is a fixed picture of today's economy (year 2007). To our knowledge, there does not exist a detailed projection of the sectoral composition of national (not even regional) economies for the next 30-40 years and only projections of future GDP would be available (European Commission, 2018).

In any case, holding the economic structure fixed makes present and future outcomes more comparable, while reinforces our approach of studying the implications of heatwaves in the absence of adaptation measures. Additionally, it minimises economic uncertainty (Orlov et al. 2020) and is more suitable to studying extreme behaviours, not a generalised shift in the behaviour of economic agents.

7. The two models used for the future projections have two different spatial resolutions. Is this on purpose? Otherwise, I would rather choose models with the same resolution to avoid that this discrepancy influences the results.

RESPONSE:

All available regional climate model simulations (total of 39 simulations, stemming from 15 EU-11 and 24 EUR-44, see Table S2 in Casanueva et al. 2020b and Figure R6) were conservatively remapped onto the WFDEI grid prior to bias correction of the WBGT input variables (see comment #5). Therefore, despite the different spatial resolution, they were treated equally in the subsequent calculations and we do not expect this to affect the results. The two selected simulations sample the lower and upper 25% of the distribution of the full ensemble climate change signal of summer mean temperature averaged across Europe in the period 2070-2099 with respect to 1981-2010. The spatial resolution is not a decisive factor in this ranking, while the driving global climate model is the dominant factor (see Figure R6).

Figure R6: Classification of the full ensemble of simulations (available for RCP 8.5) in terms of their simulated warming. Each row corresponds to one global climate model and each cell represents one simulation (i.e. one combination of regional and global climate model), being EUR-44 simulations in bold and EUR-11 in italics. The numbers depict the climate change signal of summer mean temperature averaged across Europe in the period 2070-2099 with respect to 1981-2010 and the colours represent the four quartiles.

- I also think it would be more adequate to select the climate models based on the total model range in 2035-2064 (since that is the analysed period and the changes seen in RCP8.5 are unlikely to happen, especially towards the end of the 21st century).

RESPONSE:

Our aim was to detect two clearly distinct simulations in order to cover substantial spread in the economic scenarios. The ensemble spread due to the different models and scenarios is small in the analysed period and becomes more evident from mid-century onwards (see full-century projections of WBGT for the full model ensemble in Fig.1 in Casanueva et al. 2020b and the temporal evolution of global surface temperature in Fig.12.5 in Collins et al. 2013). For this reason, we performed the selection of models conditioned to larger climate sensitivity, i.e. end of the century and strong emission scenario, and from the largest possible set of models (39 vs 13 simulations for RCP 8.5 and RCP 2.6, respectively, at the time of the analysis). Considering this time horizon by mid-century allows us to disregard both the scenario uncertainty and the discussion about the likelihood of the different pathways.

Minor comments:

9. Line 47: “Increasingly growing” says the same twice. “Increasing” or “growing” should be enough.

RESPONSE:

Thanks for your suggestion. We accepted and changed the sentence to be “rising temperatures”.

10. Line 49: Change to “evidence-based”.

RESPONSE:

We apologise for this typo. Expression amended to “evidence-based”.

11. Line 51: What exactly is meant by “unusually warm days”?

RESPONSE:

By “unusually warm”, we wanted to give a sense of high temperature anomaly, but we agree that this can be confounded by days that should not be warm but turn out to be warm. The difference is subtle, but for the sake of clarity, the phrase has been modified to “exceptionally warm”.

12. Lines 51-54: To me, the two statements about warm days and attribution of climate change do not really fit together and I think it would be better to have these in two different sentences.

RESPONSE:

We agree with the reviewer that the concatenation of these two statements reads poorly. We have amended the beginning of this paragraph as follows:

Changes in the manuscript (Line 54): “*The number of exceptionally warm days increased by up to 10 days per decade between 1960 and 2018 in most of*”

southern Europe and Scandinavia, partly attributed to human-induced climate change (King et al., 2016; Diffenbaugh, 2020; Perkins-Kirkpatrick and Lewis, 2020). This has contributed to the proliferation of heatwaves, which are projected to become more frequent and to last longer across all Europe during the 21st century (Fischer and Schär, 2010; IPCC, 2013; Russo et al., 2014)”.

13. Line 56: I would suggest writing 21st century to be clear.

RESPONSE:

Thanks for the suggestion. We have taken this suggestion onboard.

14. Line 59: Which health threats? Can they be named?

RESPONSE:

In the revised version of the manuscript, we name some health threats associated with excessive heat, as extracted from *The 2019 report of The Lancet Countdown on health and climate change* (Watts et al. 2019).

Changes in the manuscript (Line 60): “Therefore, extreme temperatures pose profound threats to future occupational health and labour productivity while exacerbating existing health problems in populations and introducing new health threats, such as heat exhaustion and heat stroke (Watts et al., 2019)”.

15. Lines 64-68: I would split this sentence in two. Explaining the definition of TX90p splits the sentence pretty strongly, so that it is hard to understand that the second part actually refers to the TX90p criterion.

RESPONSE:

We have rephrased these two sentences based on suggestions by this reviewer.

Changes in the manuscript (Line 70): “In this study, we selected the TX90p criterion, i.e., a heatwave occurs when the 90th percentile of maximum temperatures is exceeded for at least 3 consecutive days. This criterion is based on the anomaly of maximum temperature and includes information about the

entire annual cycle, which eases the identification of productivity impacts above a certain threshold of temperature”.

16. Lines 64-71: Are all daily data of the 30 years pooled together? This should be stated somewhere (probably in the methods).

RESPONSE:

Indeed, percentiles were obtained for each region from the full historical time series. This was explained in the methods, but perhaps was slightly unclear. The affected part has been rephrased as follows to improve clarity:

Changes in the manuscript (Line 367): “Heatwaves were identified at the regional level using the TX90p criterion, i.e. when the 90th percentile of the distribution of regional maximum temperatures spanned by the data from the period 1981-2010 was exceeded for at least 3 consecutive days”.

17. Line 81: RCP8.5 is not a “business as usual” scenario, but – if at all – should be considered as “worst case” (see e.g. Hausfather and Peters, 2020, doi: 10.1038/d41586-020-00177-3).

RESPONSE:

There are supporters and denials of that statement¹, even the IPCC AR5 says that “*Scenarios without additional efforts to constrain emissions lead to pathways ranging between RCP6.0 and RCP8.5*” (IPCC, 2014).

The reviewer is right, though, that given recent developments in climate policy and the energy system, the appropriateness of the term “business as usual” to be associated with RCP8.5 is questionable (Grant et al. 2020).

We have substituted this term by “high emission scenario” throughout the document.

¹ <https://climatenexus.org/climate-change-news/rcp-8-5-business-as-usual-or-a-worst-case-scenario/>

18. Lines 83-84: I think the explanation why this period is chosen should also be mentioned here and not just in the methods (I was at least wondering here why you chose it).

RESPONSE:

An additional justification of the use of this time window has been included in the main text.

Changes in the manuscript (Line 87): *“We then applied this model to a high emission scenario represented by two climate model simulations forced by the Representative Concentration Pathway 8.5 (RCP8.5, thereafter) to estimate the projected heatwave-induced costs over the years 2035–2064, less affected than more future periods by uncertainties of mitigation pathways or climate model inherent variability”.*

19. Lines 87-88: It is not instantly clear to what “Its high spatial and temporal resolution” refers. I would advise to rephrase it to make it clearer.

RESPONSE:

We agree with the reviewer’s remark. We have rephrased this part to improve clarity.

Changes in the manuscript (Line 93): *“The interdisciplinary modelling framework developed here is inspired by the emerging literature of bottom-up assessments of climate risks (Císcar et al., 2018; Conway et al., 2019; García-León et al., 20202021). Thanks to its high level of spatial disaggregation, we were able to (1) better understand the distribution of costs between sectors and regions and the mechanisms of impact propagation and (2) characterise with precision the areas more vulnerable to extreme heat stress as we quantify their present and expected future damages”.*

20. Lines 105-106: Did you test this statement?

RESPONSE:

We certainly did (please see Table R1). In the revised version, we provide some additional comments and figures on the different properties of summer vs non-summer heatwaves. They are also more severe due to the occurrence of higher temperatures during summer.

	Summer (JJA) Heatwaves	Non-summer Heatwaves
2003	9.168	5.016
2010	7.135	3.316
2015	8.075	3.789
2018	9.530	4.904
AVG	8.477	4.256

Table R1: Mean duration of heatwaves (in days) during summer (JJA) and non-summer months in the four years analysed.

21.Lines 120-121: What does it mean that people suffered from heat anomalies for around 2 months? What is meant by “heat anomalies” and how did people “suffer”?

RESPONSE:

The referred expression has been revised.

Changes in the manuscript (Line 128): “During that year, Northern Europe, in particular Scandinavian countries, experienced sustained temperature anomalies, which added up to around 2 calendar months”.

22.Line 150: What is meant by “relevant way”? Why relevant?

RESPONSE:

The referred expression has been rephrased.

Changes in the manuscript (Line 172): “Given the complementarity between primary and intermediate inputs, indirect effects spread substantially through the service sector”.

23. Line 151: What about primary inputs? Can they also be substituted in the model?

RESPONSE:

Primary inputs (or just 'inputs' or 'factors of production') in our model are labour and capital and are not included among intermediate inputs. We refer to 'intermediate inputs' as goods and services used in the production process of a firm. The firm can purchase these items in the domestic economy or import them from abroad.

Primary inputs (labour and capital) are substitutes to a certain extent. The degree of substitution or 'elasticity of substitution' between primary inputs (σ_S) is described in Eq. 3 and is sector-specific. The value of each elasticity is based on the parametrization of the GTAP 8 database (Narayanan et al. 2012).

Since we are analysing the economic consequences of heatwaves (short-term shock), we introduce some market inertias (or frictions) in the assumptions of the economic model. First, labour and capital can move freely across sectors, but they cannot move outside the sub-national unit they belong to. Then, a Leontief technology is assumed between primary and intermediate inputs. This means that primary and intermediate inputs are used in fixed proportions and intermediate inputs cannot be substituted among them.

In Orlov et al. (2019) the degree of market friction is even bigger than ours because labour and capital are immobile also across sectors. However, the model in Orlov et al. (2019) is specified at the country level while our CGE is sub-national, which further limits the spatial mobility of factors of production.

24. Lines 154-155: Is it possible to separate the impacts on intermediate inputs and the trade effect to see how strongly trade can mitigate the negative effects on input?

RESPONSE:

It is very difficult to disentangle the two effects in the CGE model because all markets are linked to each other and must be in equilibrium simultaneously. After a shock, relative prices change endogenously and markets adjust accordingly to restore equilibrium.

An explorative way to isolate the trade effect could be setting to zero all the Armington elasticities, removing all the possibilities to substitute the domestic and the imported goods. However, this poses two problems. First, setting all the Armington elasticities to zero could pose computational problems to solve for the new equilibrium. Second, even if we isolate the trade effect via switching off Armington elasticities, we will probably not be capturing the intermediate effect taking place due to the simultaneous action of different mechanisms that also play a role in determining the final equilibrium outcome. For example, how investments are redistributed across regions, or how labour and capital re-allocate across sectors.

25.Line 157: What does “exposed environment” mean?

RESPONSE:

We meant regions being exposed to more stringent heat, radiation and humidity conditions, factors all resulting in higher average WBGT. We have rephrased this part to:

Changes in the manuscript (Line 160): *“The analysis of economic losses by year reflects that most affected regions are those with either more heat prone environments or a more exposed economic structure, or a combination of the two”.*

26.Line 158: How high is the correlation? There are a few regions in Figure S2 that are high in both economic and heat exposure, but on average the correlation seems to be not so high (at least from visual inspection of Figure S2). I think it would be good to support this statement by some numbers.

RESPONSE:

From Figure S2, we learn that the (regional) correlation between heat exposure and damages is -0.78 (the higher cumulative heat, the larger the economic losses). The correlation between economic exposure and economic damages is -0.42 (again, the more exposed economic activity is, the larger the damages).

The 20 regions experiencing more economic losses show on average twice the heat exposure of the remaining regions. Analogously, this group of regions shows an economic structure that is on average 55% more exposed in economic terms (outdoor sectors). Regions showing high heat and economic exposure (that we call 'fully exposed' regions and that are encapsulated in the upper-right quadrant of Fig. S2) are associated with greater damages, as we claim in the text.

Changes in the manuscript (Line 162): *"In general, observed greater damages are associated with a group of regions that we call 'fully exposed' regions (upper-right quadrant of Fig. S2), that is, regions showing high average heat exposure and featuring a relatively large fraction of outdoor sectors. This group of regions shows on average twice the heat exposure and one and a half times the economic exposure of the remaining regions".*

27. Lines 167-169: Is it possible to be more specific here? From a short look at the Orlov et al. study it seems that they estimated an impact of about 2-3\$ per capita relative to a GDP of about 2500\$ per capita. This rather suggests a decrease in the order of 0.1%. Does the difference between your study and Orlov et al. come from the fact that you also consider indirect effects? That should be mentioned.

RESPONSE:

As stated in the first comment to reviewer #1, our study differs from **Orlov et al. (2019)** (Orlov19, hereafter) in many dimensions, some of which are repeated here:

- We assess heatwaves impacts (after a rigorous characterisation of the hazard) while Orlov19 look at average monthly values.
- We consider the whole year while Orlov19 focuses on summer months.

- Orlov19 only looks at two sectors (agriculture and construction) while we consider the whole economy.
- The economic model is different: GRACE (Orlov19), Bosello and Standardi, 2018 (ours).
- Orlov19 works with magnitudes in nominal values. We instead fix the nominal values of the economy and calculate the percentage deviations from an equilibrium situation, which improves comparability.

Consequently, the results from both approaches are not directly comparable. Still, referring to the comment from the reviewer, Orlov19 find that the social per capita cost (this is how they refer to the cost that accounts for direct and indirect effects) and the share of these costs out of total monthly per capita GDP are the following:

	2003	2010	2015
Agriculture	\$2.7	\$2.1	\$2.5
Construction	\$2.2	\$1.6	\$1.9
Agriculture + Construction	\$4.9	\$3.7	\$4.4
(Monthly) GDP	\$2023	\$2806	\$2684
Social cost share (% monthly GDP) (Orlov19)	0.24%	0.13%	0.16%
Cost of heatwaves (García-León et al.)	0.49%	0.29%	0.39%

Table R2: Social costs (direct and indirect effects) of heatwaves in Europe, as estimated by Orlov et al. (2019) and costs of heatwaves derived by García-León et al.

They also claim that “Direct economic losses were especially high in countries, such as Cyprus, Italy, and Spain”. Therefore, we say in the text that both approaches yield consistent results (losses in 2003 were the highest, then comes 2015, and then 2010) but the results are certainly not comparable in magnitude and this responds to the various differences between their methodology and ours.

With respect to **Knittel et al. (2020)**, they investigate the average impact of global warming in Germany due to labour productivity losses. They find GDP losses to range between 0.41-0.46% in 2050. We find that Germany will experience in the decade 2055-2064 losses of GDP equivalent to ~0.5% (see revised Fig.3b in the manuscript, Figure R9 in this document) in response to heatwaves. Again, ignoring the many dissimilarities between their approach and ours, the numbers are quite comparable in magnitude and thus our statement in the manuscript.

Changes in the manuscript (I) (Line 186): “In spite of the different methodologies adopted, our results are qualitatively consistent with Orlov et al. (2019). However, our approach is spatially richer and more comprehensive, since we implement a systematic heatwave characterisation method and consider all the productive economic sectors”.

Changes in the manuscript (II) (Line 245): “Central and northern European countries will experience minor but significant negative effects, as thermal stress will increase across all latitudes. GDP impacts in those regions will be more modest but still meaningful, with Germany being projected to experience a negative impact of 0.5% by 2050, a figure very similar to what is shown in Knittel et al. (2020)”.

28. Line 198: What is meant here by “measures of heat”? WBDD?

RESPONSE:

We meant that the WBGT was averaged over summer months and over years. However, in the revised version, given that we adopt the same definition of heatwaves as that used in the historical analysis and we simulate all the years in the period 2035-2064, no aggregation is required. The main text (and the methods) have been modified accordingly to reflect the new identification and estimation strategy.

29. Lines 200-203: This information could be added to supplementary Figure S3 in addition to the WBGT distributions under current climate conditions.

RESPONSE:

Figure S3 (Figure R7, below) has been updated with a new (red) shaded area reflecting the projected shift to the right of the tails of both WBGT distributions, which shows how indoor sectors will begin to be more often affected by productivity losses as a result of heat. As indicated in the text, these extreme WBGT indoor values usually correspond to southern regions.

Figure R7: New Fig.S3 in the revised manuscript. The projected shift in the right tail of the distributions of WBGT is shaded in red.

30.Line 217: I guess “Central and northern European countries”.

RESPONSE:

The reviewer is right. Phrase amended to "Central and northern European countries".

31.Lines 235-237: While this statement is true, it is not really novel.

RESPONSE:

The reviewer has a point in that this and the next sentence (comment #32) do not add much substantial value to the spatio-temporal analysis of heatwaves. These properties have been already reviewed by other papers, so it is perhaps a good idea not to highlight them in the conclusion of our paper. We have now removed

this part from the main text, shifting more the focus of to our messages about the economic implications of these events.

32.Line 238: This statement is rather trivial due to the seasonal cycle of temperature (in case daily values of the whole year were pooled to calculate TX90p).

RESPONSE:

Please refer to comment #31.

33.Line 253: Again, RCP8.5 is not “business as usual”.

RESPONSE:

Please see minor comment #17.

34.Line 263: I guess for investigating urban heat island effects, primarily models with higher spatial resolution are needed. And which specific heat stress indicators does this statement refer to?

RESPONSE:

To account for the Urban Heat Island (UHI) effect, not only models and a health-based definition for heatwaves are required, but also high temporal and spatial resolution observations, which are needed to run model simulations and/or perform statistical adaptations. Sufficiently long, highly resolved thermal heat stress data is still rare, since most of the long-observed records belong to the city outskirts, in order to fulfil certain WMO standards.

A couple of high-resolution indicators to explore the UHI (the Mean Radiant Temperature–MRT– and the Universal Thermal Climate Index–UTCI) are now suggested in the text based on recent evidence by Di Napoli et al. (2020).

35.Line 276: Referring to one of my major comments, I think that the spatial resolution can be more exploited in this study, particularly because it is highlighted in several instances.

RESPONSE:

We agree with the reviewer on their point here, since we did not fully exploit the spatial richness of our data/model in the previously submitted manuscript. We have now included a spatial analysis of the economic losses triggered by heatwaves (see comment #1), as we also describe the historical spatial pattern of losses with our GDP benchmark calculations. We have included this new feature in the manuscript as part of our conclusion and have modified the main text accordingly.

36. Line 297: What is the reason that the resolution of the economic model is not the same? What determines its resolution?

RESPONSE:

The spatial resolution of the economic model is heterogeneous due to the difficulty to obtain mutually consistent Social Accounting Matrices (SAMs) for the sub-national regions. SAMs represent the flows of all economic transactions that take place within an economy (regional or national). This object is hard to produce and is very often not available sub-nationally. Therefore, the regional SAMs must be derived from the national ones using different techniques. We use the methodology based on Simple Location Quotients (SLQs) and gravity (Bosello and Standardi, 2015).

A justification of the mismatch in regional spatial resolution has been included in the text.

Changes in the manuscript (Line 322): *“Before feeding the model with the respective labour productivity shocks, a subsequent spatial aggregation procedure was required for some regions, since the resolution of the economic model was not the same for all countries as a result of the difficulty to estimate mutually consistent regional-level Social Accounting Matrices (SAMs) required to calibrate the model (refer to Tab. S2 for a description of the spatial resolution used in the economic model)”*.

37.Line 300: What are the NUTS 0 and NUTS 1 regions? I think it would be good to shortly explain them here.

RESPONSE:

We agree that, for the non-familiarised reader, the NUTS nomenclature can result rather obscure. The text has been amended as follows:

Changes in the manuscript (Line 329): “*NUTS 0 (country) or NUTS 1 (sub-country) population-weighted spatial aggregation was applied to obtain the values in the remaining regions*”.

38.Line 308: This ISO norm has been revised in 2017. I think the newest version should be checked and cited here (and in the following instances).

RESPONSE:

We have taken this onboard (ISO 7243:2017). Thank you for pointing this out.

39.Line 403: Does nearest neighbour interpolation conserve the total population? I am not entirely convinced that it is the best fit for this purpose. At the same time, indicating 15 arcmin as 0.25° would facilitate to see the resolution difference between both datasets.

RESPONSE:

Yes – nearest neighbour interpolation preserves the total population, since we assign an average temperature level to each population cell grid, that is, we merge the two datasets adopting the resolution of the coarser one.

As requested, ‘15 arcmin’ has been substituted by ‘0.25°’.

40.Line 406: Which resolution does this population dataset have?

RESPONSE:

Future population projections also show a spatial resolution of 0.25°. A clarification has been added to the text.

41. Lines 434-436: Does this apply for short- or long-term?

RESPONSE:

In this class of analysis (comparative statics), the model is perturbed from its initial state of equilibrium and the subsequent adjustment to the new equilibrium is instantaneous, that is, there is no dynamic adjustment. The resulting two 'photographs' of the economy are compared to elucidate the effect of the shock. Hence, there is no role for the short-, long-term vision.

42. Line 456: What is "IMP"?

RESPONSE:

Thank you for this comment. The term "IMP" was indeed unclear in the former version of the manuscript. σ_{IMP} is the elasticity of substitution between imports coming from different regions, within and outside the country. This elasticity is bi-dimensional because we use the CRESH function, which allows us to identify the source and destination region of the trade flow. In this way, we can differentiate between intra-national and international trade. We amended the text to better explain this point.

Changes in the manuscript (Line 493): "Unlike the standard GTAP country-level specification, we include the domestic sub-national demand and the intra-national imports from other regions. We used two types of functions to model our trade structure. The CES function links the sub-national domestic demand and the aggregate imports of the sub-national region and uses an elasticity of substitution σ_{ARM} between the two variables. The CRESH (Constant Ratios of Elasticity of Substitution, Homothetic; Hanoch, 1971) function breaks the aggregate imports according to the source region, which can be a region within the country or outside the country. In this case, the elasticity is the bi-dimensional σ_{IMP} , which allows us to identify the source and the destination region and to differentiate between intra- and international trade. Compared to the standard GTAP model we increase σ_{IMP} by 20% if the region is trading with another region within the country".

43. Line 458: What are ARM2 and IMP2 used for? For intranational trade?

RESPONSE:

As stated above, σ_{IMP} represents the elasticity of substitution between imports coming from different regions within and outside the country. σ_{ARM} is the elasticity of substitution between sub-national domestic products and aggregated imports, which include goods from the rest of the country or the rest of the world. The numbers '1' and '2' indicate that the elasticity of substitution refers to the parameterisation of the country-level GTAP model and our regionalised version, respectively. We realised that the description was a bit confusing and have modified the paragraph to clarify further this aspect (please see changes in the manuscript shown in comment #42).

44. Line 460: Are the 20-50% reduction the reason why a factor of 1.5 is used in line 458? Moreover, I think it would be good to check if these numbers are still valid (the references are 20 years old). Especially in Europa I would expect that trade between different countries is high.

RESPONSE:

Thank you for this comment, which allows us to clarify again a not very clear paragraph. It is important to note that it is very difficult to find in the trade literature values of the Armington elasticities for such a high number of sub-national regions and even the Armington elasticities specified at the national level are subject to uncertainty. Therefore, we decided to start from the GTAP parametrisation for country-level elasticities and use a robust result in the trade literature, the so-called border effect (Anderson and Wincoop, 2003).

In our simulations we multiply the GTAP country level elasticity by a factor of 1.2 when the trade flow takes place between two regions of the same country, based on the quoted reference. This value is meant to provide only a reference to make trade within a country more fluid than trade between countries. We agree with the reviewer that further research and a sensitivity analysis on this value would be

certainly valuable, but this is out the scope of the paper and would require much computational effort. Moreover, our analysis focuses on the short-term economic consequences of heatwaves, where the effects of market frictions can be considerable. This is an additional reason for our conservative choice on the value of Armington elasticities for sub-national units.

Figure 1:

45. The abbreviation and meaning of NUTS should be explained somewhere in the caption or the main text (or it could be avoided in the main manuscript and only be mentioned in the methods).

RESPONSE:

Thanks for the suggestion. The use of 'NUTS' has been confined to the methods. It has been replaced by 'region' where applicable.

46. If I understand correctly, the units of WBDD is "°C days". I think it would be good to add this unit to the colorbar in subplot c.

RESPONSE:

Indeed. The unit of WBDD is (cumulative) °C. We have added this unit to the legend.

Figure 2:

47. There should be enough space to write the full names of the analysed sectors in subplot b instead of the abbreviations.

RESPONSE:

Done. Thanks for the suggestion.

48. It would be nice to highlight the selected figures in Figure S2.

RESPONSE:

Do you mean to highlight these regions in the text or assign a different marker to them in the figure? We already label these regions with their NUTS code. In the revised version, we also colour these regions differently. We will be very happy to accommodate the reviewer's suggestion in case our interpretation was wrong. Please see Figure R8 below.

Figure R8: New Fig.S2 in the revised manuscript. Highlighted regions are now assigned a different colour. These changes are reflected in the caption of this figure.

Figure 3:

49. I think it would be feasible to show all countries here.

RESPONSE:

All the analysed countries are now shown in the figure (see Figure R9 below).

Figure R9: New Fig.3b in the revised manuscript.

50. I think it would be better to not use country codes but the full country names, so that readers can identify the countries more easily.

RESPONSE:

Done. Please refer to comment #49 and Figure R9.

References

- Anderson JE, Wincoop E, (2003) Gravity with Gravitas: A Solution to the Border Puzzle. *American Economic Review* 93(1):170–192.
- Ballester J, Rodó X, Giorgi F (2010) Future changes in Central Europe heat waves expected to mostly follow summer mean warming. *Clim Dyn* 35, 1191–1205.
- Bosello F, Standardi G (2018) The New Generation of Computable General Equilibrium Models, eds. Perali F, Scandizzo LP. (Springer), pp. 249–277.
- Casanueva A, Kotlarski S, Herrera S et al. (2016) Daily precipitation statistics in a EURO-CORDEX RCM ensemble: added value of raw and bias-corrected high-resolution simulations. *Clim Dyn* 47:719–737.
- Casanueva A, Kotlarski S, Herrera S, Fischer AM, Kjellstrom T, Schwierz C (2019) Climate projections of a multivariate heat stress index: the role of downscaling and bias correction, *Geoscientific Model Development* 12:3419–3438.
- Casanueva A, Herrera S, Iturbide M et al. (2020a) Testing bias adjustment methods for regional climate change applications under observational uncertainty and resolution mismatch. *Atmos Sci Lett*. 21:e978.
- Casanueva A, Kotlarski S, Fischer A, Flouris A, Kjellstrom T, Lemke B, Nybo L, Schwierz C, Liniger M (2020b) Escalating environmental summer heat exposure - a future threat for the European workforce. *Regional Environmental Change* 20(40):1–14.
- Collins M, Knutti R, Arblaster J, Dufresne JL, Fichet T, Friedlingstein P, Gao X, Gutowski WJ, Johns T, Krinner G, Shongwe M, Tebaldi C, Weaver AJ, Wehner M (2013) Long-term Climate Change: Projections, Commitments and Irreversibility. In: *Climate Change 2013: The Physical Science Basis. Contribution of Working Group I to the Fifth Assessment Report of the Intergovernmental Panel on Climate Change* [Stocker, T.F., D. Qin, G.-K. Plattner, M. Tignor, S.K. Allen, J. Boschung, A. Nauels, Y. Xia, V. Bex and P.M. Midgley (eds.)]. Cambridge University Press, Cambridge, United Kingdom and New York, NY, USA.

Day E, Fankhauser S, Kingsmill N et al (2019) Upholding labour productivity under climate change: an assessment of adaptation options. *Clim Pol* 19:367–385.

Déqué M (2007) Frequency of precipitation and temperature extremes over France in an anthropogenic scenario: model results and statistical correction according to observed values. *Glob Planet Chang* 5(57):16–26.

Di Napoli C, Barnard C, Prudhomme C, Cloke HL, Pappenberger F (2020) ERA5-HEAT: A global gridded historical dataset of human thermal comfort indices from climate reanalysis. *Geosci Data J* 00:1–9.

Dosio A, Mentaschi L, Fischer EM, Wyser K (2018) Extreme heat waves under 1.5°C and 2°C global warming. *Environmental Research Letters* 13:054006.

European Commission (2018) The 2018 Ageing Report Economic & Budgetary Projections for the 28 EU Member States (2016-2070) Institutional Paper 079. Luxembourg: Publications Office of the European Union.

Gobiet A, Suklitsch M, Heinrich G (2015) The effect of empirical-statistical correction of intensity-dependent model errors on the temperature climate change signal. *Hydrology and Earth System Sciences* 19:4055–4066.

Gutiérrez JM, Maraun D, Widmann M, Huth R, Hertig E, Benestad R, Roessler O, Wibig J, Wilcke R, Kotlarski S, San Martín D, Herrera S, Bedia J, Casanueva A, Manzanás R, Iturbide M, Vrac M, Dubrovsky M, Ribalaygua J, Pórtoles J, Rätty O, Räisänen J, Hingray B, Raynaud D, Casado MJ, Ramos P, Zerenner T, Turco M, Bosshard T, Štěpánek P, Bartholy J, Pongracz R, Keller DE, Fischer AM, Cardoso RM, Soares PMM, Czernecki B, Pagé C (2019) An Intercomparison of a large Ensemble of Statistical Downscaling Methods over Europe: results from the VALUE perfect predictor cross-validation experiment. *International Journal of Climatology*, 39(9):3750–3785.

Grant N, Hawkes A, Napp T et al (2020) The appropriate use of reference scenarios in mitigation analysis. *Nat Clim Chang* 10:605–610.

Grasso L (2000) The differentiation between grid spacing and resolution and their application to numerical modeling. *Bull Am Meteorol Soc* 81:579–580.

Ivanov, M.A., Luterbacher, J. and Kotlarski, S. (2018) Climate model biases and modification of the climate change signal by intensity-dependent bias correction. *Journal of Climate*, 31, 6591–6610. <https://doi.org/10.1175/JCLI-D-17-0765.1>.

IPCC (2014) *Climate Change 2014: Synthesis Report*. Contribution of Working Groups I, II and III to the Fifth Assessment Report of the Intergovernmental Panel on Climate Change [Core Writing Team, R.K. Pachauri and L.A. Meyer (eds.)]. IPCC, Geneva, Switzerland, 151 pp.

Maraun D, Wetterhall F, Ireson AM, Chandler RE, Kendon EJ, Widmann M, Brienen S, Rust HW, Sauter T, Themessl M, Venema VKC, Chun KP, Goodess CM, Jones RG, Onof C, Vrac M, Thiele-Eich I (2010) Precipitation downscaling under climate change: Recent developments to bridge the gap between dynamical models and the end user. *Rev. Geophys.*, 48, RG3003.

Maraun D (2013) Bias Correction, Quantile Mapping, and Downscaling: Revisiting the Inflation Issue. *J. Climate* 26:2137–2143.

Kjellstrom T (2016) Impact of climate conditions on occupational health and related economic losses: a new feature of global and urban health in the context of climate change. *Asia Pac J Public Health* 28:28S–37S

Knittel N, Jury MW, Bednar-Friedl B, Bachner G, Steiner AK (2020) A global analysis of heat-related labour productivity losses under climate change—implications for Germany's foreign trade. *Climatic Change* 160: 251–269.

ISO 7243:2017 (2017) *Ergonomics of the thermal environment - Assessment of heat stress using the WBGT (wet bulb globe temperature) index*. International Organisation for Standardisation, Geneva.

Morabito M, Messeri A, Crisci A, Bao J, Ma R, Orlandini S, Huang C, Kjellstrom R (2020) Heat-related productivity loss: benefits derived by working in the shade or work-time shifting. *International Journal of Productivity and Performance Management* (in press).

Morgan C, de Dear R (2003) Weather, clothing and thermal adaptation to indoor climate. *Clim Res* 24:267–284.

Narayanan B, Aguiar A, McDougall R (2012) Global Trade, Assistance, and Production: The GTAP 8 Data Base. Center for Global Trade Analysis, Purdue University.

Orlov A, Sillmann J, Aaheim A, Aunan K, de Bruin K (2019) Economic losses of heat-induced reductions in outdoor worker productivity: a case study of Europe. *Economics of Disasters and Climate Change* 3: 191–211.

Orlov A, Sillman J, Aunan K, Kjellstrom T, Aaheim A (2020) Economic costs of heat-induced reductions in worker productivity due to global warming. *Global Environmental Change* 63:102087.

Piil JF, Christiansen L, Morris NB, Mikkelsen CJ, Ioannou LG, Flouris AD, Lundbye-Jensen J, Nybo L (2020) Direct exposure of the head to solar heat radiation impairs motor-cognitive performance. *Scientific Reports* 10:7812.

Takakura J, Fujimori S, Takahashi K, Hijioka Y, Hasegawa T, Honda Y, Masui T (2017) Cost of preventing workplace heat-related illness through worker breaks and the benefit of climate-change mitigation. *Environmental Research Letters* 12:064010.

Thiemeßl MJ, Gobiet A, Heinrich G (2012) Empirical-statistical downscaling and error correction of regional climate models and its impact on the climate change signal. *Climatic Change* 112:449–468.

Watts N, et al. (2019) The 2019 report of The Lancet Countdown on health and climate change: ensuring that the health of a child born today is not defined by a changing climate. *The Lancet* 394(10211):1836–1878.

REVIEWER COMMENTS

Reviewer #1 (Remarks to the Author):

Thank you for revising the manuscript. In the rebuttal, the authors emphasized that they used higher-resolution data and widened the temporal range to analyze. I understand the authors completed a non-trivial task, and I highly appreciate authors' efforts. However, I still feel that the authors don't make use of their results and the strong points of this study, and feel that the originality is limited to be published as an original article. In other words, lack of non-obviousness or inventiveness. What are the results or conclusions that cannot be obtained without the progresses made in this study? It sounds like - they just replaced the data with new one, and reached almost the same conclusions which have already shown in the previous studies.

For example, the authors pointed out the spatial heterogeneity of the impacts, but almost only mentioned between-country differences, not within-country differences. These discussions could be done even if they used coarser input data as previous studies (e.g., Orlov et al. 2019). For the historical analysis, the authors used hourly-resolution data, but discussions that require hourly data is not included in this study (For example, Takakura et al. (2018) conducted a study making use of hourly data).

I think the procedure of this study is methodologically sound in general. I believe this study can add more value to the existing literature if the manuscript is more drastically restructured focusing much more on spatial, temporal, and sectoral heterogeneity.

[References]

Orlov et al. Economic Losses of Heat-Induced Reductions in Outdoor Worker Productivity: a Case Study of Europe. *Economics of Disasters and Climate Change* 3, 191–211 (2019).
<https://doi.org/10.1007/s41885-019-00044-0>

Takakura et al. Limited Role of Working Time Shift in Offsetting the Increasing Occupational-Health Cost of Heat Exposure. *Earth's Future*, 6. (2018) <https://doi.org/10.1029/2018EF000883>

Reviewer #2 (Remarks to the Author):

The revised manuscript was adapted based on the comments I had. However, I still think that some points are a bit unclear and, most of all, there are a few statements for which clear evidence is missing.

Comments that might require a bit more work:

- The authors say that the intermediate goods mechanism distributes the heat impacts in the economy while trade mechanisms mitigate this, without indicating where this statement comes from. Evidence for this statement should be clearly indicated.
- The effects on indoor and outdoor workers are apparently very different (lines 272-276). But where does this statement come from?
- Line 534-546: I find this part hard to understand. In the comments to the reviewer, there is more information and also a figure about this. I would recommend to extend this paragraph and also include the figure (e.g. in the supplementary) to clearly explain the method that was used. It should also be clearly stated that in principle ERA5-Land provides hourly data for the full period 1981-2010. Thus, the chosen approach is not due to data not being available (as suggested by the text) but because of the computationally expensive calculations. However, I believe that it would be important to exactly quantify the background GDP loss due to heat and thus I would recommend to perform the analysis based on the actual WBGT values calculated from hourly ERA5-Land data for the full period (if possible given that it is computationally expensive).

Specific comments:

- Line 30: Does "this area" refer to Europe or to the economical assessment (i.e. research area)?
- Line 32: I repeat my comment from the first round that "relatively hot" is a very vague indication. Can it be replaced e.g. by "heatwaves" or something more specific (see also comment below)?
- Lines 47-52: This list contains mostly impacts on the cognitive capacity of humans, which is of course important. However, for people working outdoors in agriculture or constructions also physiological impacts of heat can be important. I think this should also be mentioned here.
- Line 52: Although I acknowledge the aim to reach the "finest possible precision", to me this phrase seems rather empty. I would rather suggest to mention what is needed to achieve better economical assessments, e.g. including more regional and sector-specific information.
- Lines 56-57: The term "exceptionally warm" days should be clearly defined. If a trend of 10 days per decade was identified, there must be a clear definition on how these "exceptionally warm" periods were defined.

Is it possible to be a bit more concrete on the human contribution to this? Would be nice to have for example a percentage value/range (if possible).

- Lines 93-95: I am not sure if it is clear to all readers why 2035-2064 is less affected by uncertainties in mitigation pathways. This should be shortly explained. Moreover, it remains unclear to what this is compared (I guess to a period at the end of the 21st century). I would also add a reference here (e.g. Samset et al., 2020, doi:10.1038/s41467-020-17001-1).
- Line 101: I doubt that a characterisation "with precision" is possible when it comes to assessments of the future. The future evolutions of climate and the economy are associated with substantial uncertainty and this statement pretends a certainty about the future that we do not have.
- Lines 105-108: The time period of this analysis should be indicated, and it should be mentioned that the standard deviation refers to interannual variability (if I understand correctly).
- Line 116: I think "JJA" should be explained at some point.
- Line 120: I am not sure if the first part of this sentence is grammatically correct.
- Lines 122-124: Does this refer to area or percentage of regions (since Figure 1b is about regions)?
- Lines 138-142: Is WBDD based on daily data?
- Lines 146-149: These lines are not entirely clear to me. What is this statement meant to say?
- Line 172: Does this refer to your study or to current research activities in general? This should be mentioned more precisely.
- Line 174: Why are transport services affected? I would assume that cars and trucks are among the best air-conditioned workplaces.
- Lines 180-182: Where is this shown or indicated?
- Lines 183-189: This paragraph is not entirely clear to me. Is the sun version of WBGT used for outdoor activities or not? That does not become clear in this paragraph.
- Lines 190-191: This specific comparison with the Orlov et al. study comes a bit surprising. I would rather recommend to generally mention that the results are consistent with other studies and then highlight differences to specific studies. Now it looks more like you defend your approach against this specific study.
- Lines 210-211: Did you check if these mechanisms have a significant effect on your estimates? The statement suggests that this is the case. If yes, this could be written more clearly, but if not this statement is misleading.
- Lines 218-219: I would not agree with this statement. Given the uncertainty in climate projections, it should not be the aim to "minimise" the uncertainty because that gives a wrong impression of a certainty that is not achievable. Isn't it rather that the uncertainty is considered in this study by using two different climate model runs with different trend strength? I would thus recommend to rephrase this sentence.
- Lines 239-240: From the methods section I had the impression that the Hothaps approach was also tested, so it would be possible to check this statement (anyhow, no matter if it was already tested or not, it should be possible and would be good to test it).
- Line 251: Was a statistical test for significance performed? If yes, which one and what are the p-values? If not, the word "significant" should be avoided here. In any case, I would suggest to perform a significance test.

As an additional comment to this question: Would it be possible to show the (interannual) variability of the future estimates in Figure 3b? I don't know if the figure would get too crowded in

this case, but it would be an interesting addition to see how variable the GDP estimates are.

- Lines 272-276: Where is the evidence for this statement?
- Lines 283: I guess this should be something like "further adaptation" instead of "further climate action" as in the current formulation it contradicts the statement in lines 257-260, at least for the upcoming decades (of course not for the second half of the 21st century).
- Lines 292-297: Why would these indicators be more appropriate for quantifying urban heat island effects? Is that not possible with WBGT? And I guess this also strongly depends on the resolution of climate models (e.g. 0.44° is too coarse to reliably estimate heat effects in cities) and on how urban areas are represented in climate models.
- Lines 302-306: I think this is very crucial. In your study you perform a sophisticated analysis, considering hourly WBGT values and considering different effects in different regions and economic sectors. But for translating the heat impacts into economic impacts, the transfer functions are still rather simple. I think that a better quantification of the economic impacts due to heat would help to obtain a more sophisticated transfer function.
- Lines 366-367: "a fair proxy" – how can this be said? Did you test that WBGT is indeed a fair proxy given the measures not considered in this study?
- Line 382: I guess this should mean that it is a compromise of being close enough to the present to be relevant, but far enough so that the signal emerges from the noise, right? The statement about "strong influence of random internal variability" might not be understandable for all readers, and I would thus suggest to argue with the emergence of signals.
- Line 456: I would recommend to use a more sophisticated method for interpolation, as nearest neighbour interpolation is very simple and a lot of information might be lost if grid resolutions are very different. Moreover, it is not clear to me whether the climate data or the population data were interpolated.
- Line 465: What are calendar effects?
- Lines 512-516: The authors explain very nicely in their answer to the reviewer comments why such a high value is used as a barrier for trade in Europe. I would suggest to extend this paragraph a bit more to explain that it is difficult to obtain these values and to defend the usage of 20%
- Figure 2b: I think that only few readers are familiar with the region abbreviations used here. I would thus strongly suggest to either write the country names (plus maybe the region code in parentheses) or add a map that shows where the regions are located. Although the information is also contained in the supplementary information, several readers might not take the effort to check that.

Reviewer #3 (Remarks to the Author):

I find the paper a very well written piece research with three original and important contributions:

1. Assessment of heat effects on workability, and hence economic performance, at regional scale (NUTS2) for Europe; while climate hazards and health impacts have been analyzed at high spatial resolution, an economic assessment at regional scale (below nation scale) so far was missing and is an extremely valuable and notable contribution
2. Analysis based on different impacts functions, i.e. heat to workability transfer functions (using ISO vs Hothap vs NIOSH); impacts are first calculated on seasonal scale (particularly relevant for sectors like agriculture and construction), only afterwards aggregated to yearly scale
3. Comparison of decadal projections with past heat wave years (2003, 2010, 2018) which is quite unique (usually you find either econometric assessments of past events or simulation based projections for the future)

Overall, the paper is very well written and the analysis is rigorously conducted. However, I would like to echo two previous reviewer requests: you should make more out of the regional economic analysis and you should carefully investigate scenario and model uncertainties. In addition, I find the assumption of "fixed economic activity" in the simulations for the future as problematic. Here are my suggestions in more detail:

1. Climate change impacts/risks are the combination of hazard, exposure and sensitivity. In Fig. 1, hazard is illustrated for past heatwaves. Fig. 2 illustrates GDP effects split up by latitude and main economic sectors. I would find it more informative to present economic impacts also as map as there is not only a north-south gradient in exposure but countries/regions differ also in economic exposure (which economic sectors dominate?) and sensitivity (effect of climatization, adaptive capacity etc.)

2. Uncertainties matter not only in terms of RCP scenarios, GCMs selected, and impact function used (ISO vs Hothap vs NIOSH) but also in terms of SSP scenarios. This has been demonstrated in the context of health e.g. in Rohat et al. (2019) for Europe. It is therefore common practice to investigate different RCP-SSP combinations by drawing on IIASA's SSP marker database. For heat driven impacts on occupational health, the most important variables to consider would be not only population but also economic growth (available both at NUTS0 level from this database, but regionalizations for Europe exist, see e.g. Rohat et al. 2018; Kok et al. 2019 and various deliverables from the IMPRESSIONS project). For instance, analysis in the context of water scarcity have shown that e.g. a move of people from Eastern Europe towards Western Europe has quite substantial consequences for risks (Harrison et al. 2019).

3. In addition, while the assumption of "holding fixed the current economic development" is still employed in some CGE studies (e.g. Ciscar et al. 2014 and subsequent PESETA projects), today the majority of CGE modeling exercises integrate RCP-SSP scenarios based on information on GDP growth, fossil fuel use, population growth etc. This "everything remains equal" assumption is particularly problematic as both exposure and sensitivity depend on economic structures and where the (working) population lives. Takakura et al. (2019) demonstrate that for Europe, half of the variance by 2050 is contributed by socioeconomic development. The argument put forward by the authors of the current paper that the same economic structure allows for comparability can be easily countered and addressed by the usual approach of comparing e.g. RCP4.5-SSP2 to SSP2 with no (additional) climate change.

In addition, I have a couple of smaller points:

4. In the model base year (2007), the damages are already included in the economic data, i.e. input output tables report economic activity considering these damages. So all WBGT/WBDD values in the heat wave years 2003, 2010, 2015, 2018 need to be expressed relative to this base year, not to the base period 1981-2010. The same holds also for impacts in the base period. My presumption is that changing this will not alter results significantly but it matters in terms of consistency of assumptions.

5. In principle, the setup would allow for a re-analysis of observed impacts by comparing them to the simulated damages with the CGE model. While such an analysis would constitute a paper in itself, it would be good to compare the scale of simulated damages to reported damages in media and elsewhere (e.g. in the introduction).

6. Overall, I suggest to strive for consistency in terms of either reporting regional effects or country effects both for past heat waves (figs. 1c, 2b) and future simulations (Fig. 3b). Personally, I would also present regional effects (NUTS1 level) always in the form of maps (not as bar diagrams as in Fig. 2b). I also find the reporting of sectoral effects more relevant at the national scale than the regional scale. But this is a matter of taste.

7. In terms of balance between past and future effects, I find the current presentation tilted too strongly towards the past, and also towards hazards. E.g. the information that both the duration and the severity of heat waves matter is important, but in the subsequent analysis only WBDD are used. So I would move Figs. 1a+b into the supplementary material. For understanding future risks, some information is however needed on how exposure and sensitivity differs across European regions and how these socioeconomic conditions change under different SSPs. This could be added to the main text as figure and used to explain differences in results across regions/countries.

8. Fig. 1c: I would find WBGT (or WBDD) reported as anomalies (delta approach) easier to

understand than the absolute WBDD values for the years. It would be good to have similar maps for future periods (e.g. as Supplementary material). In the caption, there is a typo ("sun version" instead of "sum version").

9. Fig. 2a: as argued above, I think there are better ways to explore economic impacts across regions (e.g. a panel of maps instead of bar chart).

10. Fig. 3a: why do you have different box plots for the two GCMs? Usually different GCMs and impact models are collated within one box plot, but different plots are used for different RCP-SSP scenario combinations (which you currently do not have). Is the difference in mean values between different decades statistically significant?

11. Regarding the contribution to the most related literature, in addition I would also add Takakura et al.(2017) who conduct a global assessment and also look into sectoral effects.

Literature cited:

Harrison PA, Dunford RW, Holman IP, et al (2019) Differences between low-end and high-end climate change impacts in Europe across multiple sectors. *Regional Environmental Change* 19:695–709. <https://doi.org/10.1007/s10113-018-1352-4>

Kok K, Pedde S, Gramberger M, et al (2019) New European socio-economic scenarios for climate change research: operationalising concepts to extend the shared socio-economic pathways. *Regional Environmental Change* 19:643–654. <https://doi.org/10.1007/s10113-018-1400-0>

Rohat G, Flacke J, Dao H, van Maarseveen M (2018) Co-use of existing scenario sets to extend and quantify the shared socioeconomic pathways. *Climatic Change* 151:619–636. <https://doi.org/10.1007/s10584-018-2318-8>

Rohat G, Flacke J, Dosio A, et al (2019) Influence of changes in socioeconomic and climatic conditions on future heat-related health challenges in Europe. *Global and Planetary Change* 172:45–59. <https://doi.org/10.1016/j.gloplacha.2018.09.013>

Takakura J, Fujimori S, Takahashi K, et al (2017) Cost of preventing workplace heat-related illness through worker breaks and the benefit of climate-change mitigation. *Environmental Research Letters* 12:064010. <https://doi.org/10.1088/1748-9326/aa72cc>

Response to reviewers

Many thanks to the editor and reviewers for these additional input and constructive comments to the manuscript, that we have incorporated in our revised manuscript following completion of all the additional experimental simulations requested as well as specific suggestions by the reviewers.

Please find the associated point-to-point responses to each reviewer below. To facilitate the work of the reviewers, each comment has been repeated and our responses inserted after that. We include how we have revised things, or if we have slightly disagreed with something, we stated why.

We hope that the reviewers will find our responses to their comments satisfactory. Any further suggestion aimed at improving the final quality of the manuscript will be warmly welcomed.

A revised version of the manuscript including all reviewers' suggestions and re-designed graphical and tabular output are attached to this submission.

Looking forward to hearing from you soon.

Sincerely,

The authors

To Reviewer #1's comments

1. Thank you for revising the manuscript. In the rebuttal, the authors emphasized that they used higher-resolution data and widened the temporal range to analyze. I understand the authors completed a non-trivial task, and I highly appreciate authors' efforts. However, I still feel that the authors don't make use of their results and the strong points of this study, and feel that the originality is limited to be published as an original article. In other words, lack of non-obviousness or inventiveness. What are the results or conclusions that cannot be obtained without the progresses made in this study? It sounds like - they just replaced the data with new one, and reached almost the same conclusions which have already shown in the previous studies.

Thanks to this reviewer for his comments, which have helped us to improve the manuscript.

The reviewer rightly asks: "What are the results or conclusions that cannot be obtained without the progresses made in this study?" The main contributions of this paper can be synthesised as follows:

- This paper provides accurate estimates of the economic burden of heatwaves in Europe. Compared to previous studies, **our methodology shows a greater level of spatial** (regional level), **temporal** (climate hourly data) **and sectoral detail** and is based on an economic model specifically regionalised and calibrated to reproduce the behaviour of the European economy. For a detailed list of all the methodological contributions and novelties of our paper, we refer the reviewer to the previous review, in which all of them were extensively described.
- It unveils evidence about the **regional disparities** of the economic effects of this climate risk while it illustrates the **driving factors** of these differences.
- It gives a **complete overview on the past, present and projected future** evolution of the impacts caused by this hazard, covering a timespan of 85 years (1981-2065) based on yearly estimates for the full time series.

We highlighted these aspects in the revised version of the manuscript by including more detailed (both regional and sectoral) results explicitly.

2. For example, the authors pointed out the spatial heterogeneity of the impacts, but almost only mentioned between-country differences, not within-country differences. These discussions could be done even if they used coarser input data as previous studies (e.g., Orlov et al. 2019).

Thanks for raising the issue of the spatial heterogeneity of impacts, which is inherent and central to our analysis. Within-country differences are present in the text in several instances, especially (but not only) in the analysis of past events (please refer to Fig. 2 and Fig. 3 and references to these figures in the main text).

We have included new graphical material to reinforce this aspect, improving the regional dimension of the analysis of past heatwaves (new Fig. 2a) and projected regional costs (new Supplementary Fig. 5).

3. For the historical analysis, the authors used hourly-resolution data, but discussions that require hourly data is not included in this study (For example, Takakura et al. (2018) conducted a study making use of hourly data).

Thanks for this comment and for pointing us to the interesting work by Takakura and colleagues.

We would like to highlight that we use hourly WBGT data for the past, but also for the assessment of future events (see the Supplementary Section “Climate Data”). The use of future hourly data is another feature of our study since, as of today, future projections of such meteorological variables are seldom publicly available at hourly intervals.

Unlike Takakura’s approach, based on establishing a statistical machine learning model between daily modelled and hourly observed data (trained only for some selected stations), we applied the 4+4+4 method (Kjellstrom et al., 2018; Casanueva et al., 2020) which constitutes a good approximation of the WBGT diurnal cycle (compared to more complex and computationally intense temperature models, Bilbao et al. 2002) for each gridbox independently. This way, we account for the spatial variability in the diurnal cycle and allow WBGT values derived from changing conditions in the model beyond the observed ones (i.e., WBGT stemming from unobserved combinations of the input variables). In what concerns the use of hourly data, we both consider 9-17h as the baseline working time.

The gains of using hourly data are implicit, since the heat stress level measured circumscribes to the actual time devoted to work, avoiding the presence of potential biases resulting from the use of 24h, day- or night-time temperatures. This has been highlighted in the revised manuscript (Supplementary Section “Heat stress index”).

Changes in the manuscript (Lines 398-402): “The use of hourly WBGT is essential, since the heat stress level encompasses the actual time devoted to work, avoiding the presence of potential biases resulting from the use of 24h, day- or night-time temperature (e.g., Casanueva et al. 2020 illustrate the clear underestimation of heat stress based on daily mean WBGT).”

4. I think the procedure of this study is methodologically sound in general. I believe this study can add more value to the existing literature if the manuscript is more drastically restructured focusing much more on spatial, temporal, and sectoral heterogeneity.

We have extensively revised our paper in all the dimensions highlighted by this reviewer (also following recommendations from the remaining reviewers).

Spatial: Greater regional detail of historical impacts is shown in the revised Fig. 2. Spatial differences in the analysed years are illustrated in the two panels of Fig. 2, together with a comparison of current versus historical damages. Furthermore, sub-national impacts under future conditions are shown in Supplementary Fig. 5 for selected countries.

Temporal: The historical period 1981-2010 is now covered with extensive detail after having simulated all the years over that period, thus obtaining the full time series of impacts (shown in Fig. 2b, grey-shaded areas). We have also filled the temporal gap 2021-2034, offering a continuity between the past recent years, the present time, the immediate future and a likely medium-term future, spanning a complete time window analysis of 85 years. The importance of the use of hourly input data is highlighted in the revised manuscript.

Sectoral: More insights about the sectoral composition of damages are provided in an expanded version of Fig. 3 (previously, Fig. 2b). This analysis is however restricted to the analysis of current impacts, as future sectoral economic structure is subject to high uncertainty.

To Reviewer #2's comments

The revised manuscript was adapted based on the comments I had. However, I still think that some points are a bit unclear and, most of all, there are a few statements for which clear evidence is missing.

Thanks for recognising the effort we did in reshaping the former version of the manuscript. We hope to have addressed all the comments in the present round of revision.

Comments that might require a bit more work:

1. The authors say that the intermediate goods mechanism distributes the heat impacts in the economy while trade mechanisms mitigate this, without indicating where this statement comes from. Evidence for this statement should be clearly indicated.

The evidence for this aspect is embedded in the theoretical structure of the economic model. Heatwaves affect mostly outdoor sectors, and these sectors are in turn linked to indoor sectors through the consumption of intermediate goods (i.e., inputs in their production processes). Given the complementarity assumption (Leontief) between gross value added and intermediate goods, when outdoor production declines, indoor production is also negatively affected through intermediate goods. The observed reduction in the production of indoor businesses is in general smaller than that observed in outdoor sectors and is determined by the weight of the outdoor-produced products in that specific indoor activity. However, this negative spillover is mitigated by the trade mechanism. In fact, when the unitary cost of the intermediate goods produced in the outdoor activity increases because of the heatwave, the firm has the option to substitute the domestic intermediate good with the same intermediate good, possibly imported from another region potentially less affected by the heatwave. This contributes to alleviate the overall economic loss.

Said that (the following replicates our argument from the previous round), it is very difficult to disentangle the two effects in the economic model because all markets are linked to each other and must be in equilibrium simultaneously. After a shock, relative prices change endogenously, and markets adjust accordingly to restore equilibrium. An explorative way to isolate the trade effect could be setting to zero all the Armington elasticities, removing all the possibilities to substitute both the domestic and the imported goods. However, this poses two problems. First, setting all the Armington elasticities to zero could imply computational problems for solving for the new equilibrium. Second, even if we isolate the trade effect via switching off Armington elasticities, we will probably not be capturing the intermediate effect taking place due to the simultaneous action of different mechanisms that also play a role in determining the final equilibrium outcome. For example, how investments are redistributed across regions, or how labour and capital re-allocate across sectors.

The following clarification is proposed:

Changes in the manuscript (Lines 195-197): “These two mechanisms are embedded into our economic model. The fact that we identify economic losses in most indoor sectors, suggest that the intermediate goods mechanism outweighs the mitigating effect of trade.”

2. The effects on indoor and outdoor workers are apparently very different (lines 272-276). But where does this statement come from?

This statement refers to the direct impacts of heat on production, i.e., labour productivity damages caused by excessive heat. In our framework, heat stress is captured by WBGT. Productivity damages are obtained by converting the WBGT signal into productivity damages using different heat exposure response functions (NIOSH, Hothaps and, mainly ISO).

The bottom panel of Supplementary Fig. 3 shows that the current distribution of the shade version of WBGT, that is, WBGT experienced in indoor sectors, hardly interacts with the various exposure-response curves. Meanwhile, the distribution of the sun version of WBGT overlaps with all the exposure-response functions for a wide range of heat values. We analyse this evidence in two sections of the manuscript (lines 198-203 and lines 242-247).

To make explicit the point that we refer to labour productivity damages caused by heat, we propose the following amendment:

Changes in the manuscript (Lines 301-306): “Under current climate conditions, outdoor workers seem disproportionately more affected by extreme heat, while most indoor work remains insulated. The analysis of the distribution of our heat stress measures ($WBGT_{sun}$ and $WBGT_{shade}$) suggests the presence of generalised and widespread outdoor productivity impacts but very mild indoor productivity damages only in southernmost regions.”

3. Line 534-546: I find this part hard to understand. In the comments to the reviewer, there is more information and also a figure about this. I would recommend to extend this paragraph and also include the figure (e.g. in the supplementary) to clearly explain the method that was used. It should also be clearly stated that in principle ERA5-Land provides hourly data for the full period 1981-2010. Thus, the chosen approach is not due to data not being available (as suggested by the text) but because of the computationally expensive calculations. However, I believe that it would be important to exactly quantify the background GDP loss due to heat and thus I would recommend to perform the analysis based on the actual WBGT values calculated from hourly ERA5-Land data for the full period (if possible given that it is computationally expensive).

After long discussion, we cannot but agree with the suggestion of this reviewer about performing a full analysis of the historical period 1981-2010 using ERA5-Land hourly data. We also believe that exactly quantifying the long-term distribution of costs incurred by regional economies under heatwaves is crucial to have a clear picture of the economic burden posed by this extreme event.

The new experiment has entailed a considerable effort in terms of downloading, processing and manipulating the necessary data. In return for this, we acknowledge several ways in which our analysis has improved:

- We now have not only the average regional-level historical impacts of heatwaves, but their complete distribution.
- We have now a robust estimate of background GDP loss for a fairer comparison with future GDP losses.

These improvements can be seen, for example, in Fig. 2b and Fig. 4(a,b), respectively.

With this addition, the full timespan covered by our analysis amounts to 85 years (1981-2064). This has been possible by also filling the gap for the period 2021-2034. Please refer to the caption of Fig. 4a, where the method for dealing with this period is described.

Changes in the manuscript (Lines 590-602): The former section '*Benchmark GDP loss*' (now '*Historical GDP losses*') has been accordingly redesigned.

Specific comments:

4. Line 30: Does "this area" refer to Europe or to the economical assessment (i.e. research area)?

Our area of study is a broad definition of Europe (consisting of EU countries, UK and EFTA countries). In our opinion, the two areas the reviewer is mentioning are basically interchangeable in this context.

5. Line 32: I repeat my comment from the first round that "relatively hot" is a very vague indication. Can it be replaced e.g. by "heatwaves" or something more specific (see also comment below)?

Thanks for the comment and our apologies for not amending this already in the previous version.

Changes in the manuscript (Lines 31-32): "*In the analysed years, the total estimated damages attributed to heatwaves amounted to 0.3%–0.5% of European gross domestic product (GDP).*"

Analogously, in the conclusion:

Changes in the manuscript (Lines 296-298): "*At the European level, total annual losses attributable to heatwaves amounted to 0.3%–0.5% of European GDP in the analysed years while the average GDP loss over the period 1981-2010 was estimated to be close to 0.2%.*"

6. Lines 47-52: This list contains mostly impacts on the cognitive capacity of humans, which is of course important. However, for people working outdoors in agriculture or constructions also physiological impacts of heat can be important. I think this should also be mentioned here.

Thanks for the suggestion. Indeed, the main representative effect of heat on outdoor workers has to do with physiological impacts, so this has to be mentioned.

Changes in the manuscript (Lines 47-52): “Excessively hot environments are precursors of biophysical and cognitive impacts, causing physiological strain to workers (Ioannou et al., 2021), lowering the number of hours of work supplied (Takakura et al., 2017), affecting the capacity of assimilating information (Park et al., 2020) and interfering with decision-making (Heyes and Saberian, 2019), ultimately undermining human capital accumulation and, therefore, economic growth.”

7. Line 52: Although I acknowledge the aim to reach the “finest possible precision”, to me this phrase seems rather empty. I would rather suggest to mention what is needed to achieve better economical assessments, e.g. including more regional and sector-specific information.

We agree with the reviewer in that it is important to explicitly mention what should be desired for better quality assessments.

Changes in the manuscript (Lines 52-54): “In a context of rising temperatures, quantifying the economic impact of these externalities with spatially resolved socioeconomic data and models is key to combat their effect,…”

8. Lines 56-57: The term “exceptionally warm” days should be clearly defined. If a trend of 10 days per decade was identified, there must be a clear definition on how these “exceptionally warm” periods were defined. Is it possible to be a bit more concrete on the human contribution to this? Would be nice to have for example a percentage value/range (if possible).

We have slightly amended this phrase and provided an exact definition of ‘hot/warm’ plus a reference supporting our statement.

Regarding the specific contribution of climate change to the increasing trend of heatwaves, it is not possible to come with an exact figure since attribution studies are usually developed for specific events. For instance, Stott et al. (2004) claimed that human influence has at least doubled the risk of heatwaves as extreme as the 2003 heatwave and Vogel et al. (2019) found that the 2018 heat event would not have occurred without human-induced greenhouse gas emissions. In the same vein, IPCC (2013) concluded that it is likely that human influence has more than doubled the probability of occurrence of heatwaves in some locations.

Changes in the manuscript (Lines 57-63): “The number of days exceeding the 90th percentile threshold (baseline period, 1970-2000) have doubled between 1960 and 2017 across the European land area (EEA, 2019), partly attributed to human-induced climate change (King et al., 2016; Diffenbaugh, 2020; Perkins-

Kirkpatrick and Lewis, 2020). According to Stott et al. (2004) and IPCC (2013), it is likely that the human influence has more than doubled the risk of heatwaves for some locations in particular events, such as the 2003 European heatwave.”

9. Lines 93-95: I am not sure if it is clear to all readers why 2035-2064 is less affected by uncertainties in mitigation pathways. This should be shortly explained. Moreover, it remains unclear to what this is compared (I guess to a period at the end of the 21st century). I would also add a reference here (e.g. Samset et al. 2020, doi:10.1038/s41467-020-17001-1).

Thank you for these valuable points. We added a short explanation to make it clearer why 2035-2064 is less affected by uncertainties in mitigation pathways. We also included some standard references for more background in the sources of uncertainty in climate model projections, which change as one moves from short- to mid- and long-term projections. Hawkins and Sutton (2009) and Giorgi (2010) show that the uncertainty due to the emission scenario becomes important at multidecadal scales and increases constantly over time, whereas internal variability is relevant in the early 21st century (especially for small regions) and becomes negligible very rapidly for later decades. IPCC (2013) also claims that, for global temperatures after mid-century, scenario and model ranges dominate the amount of variation due to internally generated variability, with scenarios accounting for the largest source of uncertainty in projections by the end of the century. Differences among scenarios also emerge after mid-century for 21st-century projections of WBGT in Europe (Casanueva et al. 2020).

Changes in the manuscript (Lines 93-101): “We then applied this model to a high emission scenario represented by two climate model simulations forced by the Representative Concentration Pathway 8.5 (RCP8.5, thereafter) to estimate the projected heatwave-induced costs over the years 2035–2064. This mid-21st century period offers a good balance between foresight and uncertainty. It is less affected by uncertainties associated with the climate model inherent variability than early future periods. And, at the same time, it is subject to less uncertainty associated to mitigation pathways than late-21st century periods, as the latter increases constantly over time (Hawkins and Sutton, 2009; Giorgi, 2010).”

10. Line 101: I doubt that a characterisation “with precision” is possible when it comes to assessments of the future. The future evolutions of climate and the economy are associated with substantial uncertainty and this statement pretends a certainty about the future that we do not have.

The reviewer is right. The term ‘with precision’ is not coherent when referring to (uncertain) projections, so it has been removed from the main text.

11. Lines 105-108: The time period of this analysis should be indicated, and it should be mentioned that the standard deviation refers to interannual variability (if I understand correctly).

Although already mentioned in line 84, the analysed time period is repeated here for convenience, following the recommendation from the reviewer.

The standard deviation refers to the number of events identified per year, so it refers indeed to an annual standard deviation.

Changes in the manuscript (Lines 111-115): “Extreme hot spells in Europe varied greatly in frequency, duration, extension and severity in the years analysed (2003, 2010, 2015 and 2018). Considering the 274 regions contained in our area of study (see the Methods for further details) and adopting the TX90p criterion, an average of N=1180 (sd: ±230.2) regional heatwave events were identified per year.”

12. Line 116: I think “JJA” should be explained at some point.

Thanks for the suggestion. The following clarification has been included in the main text:

Changes in the manuscript (Lines 123-125): “Most heatwaves were concentrated during the summer months (June, July and August; JJA henceforth), but extended before and after this time frame, particularly in 2003 and 2018 (Fig. 1a,b.)”

13. Line 120: I am not sure if the first part of this sentence is grammatically correct.

We have rephrased this sentence based on suggestions by this reviewer.

Changes in the manuscript (Lines 128-129): “The total European area affected by heatwaves varies according to the time of the year analysed.”

14. Lines 122-124: Does this refer to area or percentage of regions (since Figure 1b is about regions)?

Thanks for the comment. Indeed, it would be more accurate to speak about ‘regions’ rather than ‘area’. In the legend of Fig. 1b, the term ‘NUTS’ has also been replaced by ‘Regions’

Changes in the manuscript (Lines 131-132): “...with an average spatial extension of 49% of the total number of regions studied and a maximum coverage of more than 95% during large-scale episodes.”

15. Lines 138-142: Is WBDD based on daily data?

Our measure of WBDD results from the sum of daily values but is based on hourly data. Daily WBDD values are obtained by averaging observations over the working day (9-17h), considering when temperatures are above the temperature threshold of 26°C.

A clarification has been added to the main text.

Changes in the manuscript (Lines 149-151): “A Wet Bulb Degree-Day (WBDD) is here defined as any additional Wet-Bulb degree over 26°C experienced by a worker under heatwave days, considering only working hours.”

16. Lines 146-149: These lines are not entirely clear to me. What is this statement meant to say?

Thanks for flagging this confusing statement. We have rephrased the complete sentence.

Changes in the manuscript (Lines 155-159): “Our analysis of regional heatwaves shows that these events are largely heterogeneous in terms of spatial and temporal characteristics. This underpins the importance of using local and timely data and high-resolution economic tools when it comes to analyse the impacts of heatwaves and other related climate extreme events.”

17. Line 172: Does this refer to your study or to current research activities in general? This should be mentioned more precisely.

This statement is based on our results. We have stated it more clearly in the main text.

Changes in the manuscript (Lines 186-188): “Our results suggest that, in present times, direct impacts of heat on labour productivity take place mostly in outdoor sectors. However, these losses propagate to the entire economy.”

18. Line 174: Why are transport services affected? I would assume that cars and trucks are among the best air-conditioned workplaces.

In this study we assume that transportation services are carried out mainly outdoors, incurring in a medium workload (Supplementary Table 3). This assumption is based on examples of the literature suggesting that this activity is highly environmentally exposed, not only to temperature and radiation, but also to air pollutants (Schifano et al., 2019). This argument would apply broadly to all kinds of transportation activities, from freight to emergency transportation.

19. Lines 180-182: Where is this shown or indicated?

We thank the reviewer for pointing this out. As explained in a previous comment (please see comment #1), this is not explicitly indicated in the results, but is the consequence of the theoretical structure of the CGE model, which is commonly accepted in this kind of macro-economic models. We have amended the text accordingly.

20. Lines 183-189: This paragraph is not entirely clear to me. Is the sun version of WBGT used for outdoor activities or not? That does not become clear in this paragraph.

Indeed, the sun and shade versions of WBGT are used for outdoor and indoor sectors, respectively. From Supplementary Fig. 3, it can be seen that, no matter the exposure-response function used, indoor workers are hardly affected by heat-induced productivity impacts under current climate conditions. This is what we meant to say with the referred sentence. We have rephrased this bit to clarify our message.

Changes in the manuscript (Lines 197-205): “In our analysis we predominantly identified direct impacts of heat on labour in outdoor sectors. This becomes clear by looking at the differences between the distributions of the sun and shade versions of WBGT (Supplementary Fig. 3), used for outdoor and indoor sectors, respectively. Under current climate conditions and, irrespectively from the exposure-response function used, heatwaves tend to show a strong impact on ambient exposed work, while indoor work remains hardly affected. The lack of solar radiation and the typically lower metabolic intensity of indoor jobs helps to protect further this group of workers.”

21. Lines 190-191: This specific comparison with the Orlov et al. study comes a bit surprising. I would rather recommend to generally mention that the results are consistent with other studies and then highlight differences to specific studies. Now it looks more like you defend your approach against this specific study.

Thanks for the suggestion. The following modification is proposed:

Changes in the manuscript (Lines 206-210): “Our results are qualitatively consistent with other studies dealing with labour impacts of excessive heat in Europe (Orlov et al., 2019). However, this work differs from the previous literature in several aspects. Among them, we introduce a systematic heatwave characterisation throughout the year, consider all the productive economic sectors, or adopt a higher spatial resolution level.”

22. Lines 210-211: Did you check if these mechanisms have a significant effect on your estimates? The statement suggests that this is the case. If yes, this could be written more clearly, but if not this statement is misleading.

This statement was meant to refer to the implementation of air conditioning, which seems only relevant for indoor sectors. We agree with the reviewer that it could be understood that we are claiming that heat-insulation measures (of all kinds) are only relevant to indoor workers, which is not the case. We have rephrased the sentence to clarify this point.

Changes in the manuscript (Lines 226-229): “However, since the implementation of heat-insulation measures is still quite low in outdoor sectors and air conditioning availability only affects indoor sectors, these adaptation effects do not seem to have a significant effect in our current estimates.”

23. Lines 218-219: I would not agree with this statement. Given the uncertainty in climate projections, it should not be the aim to “minimise” the uncertainty because that gives a wrong impression of a certainty that is not achievable. Isn't it rather that the uncertainty is considered in this study by using two different climate model runs with different trend strength? I would thus recommend to rephrase this sentence.

The reviewer raises a relevant point here. The affected sentence has been rephrased.

Changes in the manuscript (Lines 236-237): “We sought to account for these uncertainties by analysing two different climate models,...”

24. Lines 239-240: From the methods section I had the impression that the Hothaps approach was also tested, so it would be possible to check this statement (anyhow, no matter if it was already tested or not, it should be possible and would be good to test it).

The reviewer is right. We used three different exposure-response functions for a sensitivity analysis of our results (see Supplementary Fig. 3). But we used ISO as our benchmark function. By construction, the ISO function tends to penalise lower temperature levels with higher productivity damages compared to Hothaps. This is also evidenced by Orlov et al. (2020):

“We find that when using the Hothaps function, reductions in global GDP are considerably less pronounced than under the ISO 7243:1989 standards.”

This difference amounts in our case to more than 0.3 percentage points of GDP by 2050, according to our calculations.

The second part of the sentence has been amended. Under RCP8.5, the ‘average’ effect of temperatures on labour productivity should, in principle, be higher in annual terms than effect of heatwaves, since heatwave days (even if we define them according to temperature percentiles obtained from the historical period 1981-2010) should represent a smaller fraction of the year than hot days.

We have identified other relevant methodological references that can explain the differences between our results and Orlov’s. Specifically, the adaptive capacity implied by the high economic growth implied by the RCP8.5-SSP5 scenario, which let regional economies to endogenously adapt to warmer environments by shifting to a more capital-intensive economy. This affects Orlov and not our results because, in contrast to Orlov, we adopt a comparative-static approach, which, in plain words, amounts to a counterfactual exercise using future climate conditions while holding fixed today’s economic structure. We have checked with our dynamic experiments that this adaptation mechanism also take place in our setup (see Response to Reviewer 3, comments 2,3 for a detailed description of the experiment and the adaptation mechanism).

Changes in the manuscript (Lines 256-269): “As a matter of comparison, Orlov et al. (2020) found that by 2050 Europe will experience economic losses in the range of 0.5%. We attribute this difference with respect to our findings mainly to the heat-exposure function used (Hothaps vs ISO), as ISO is more sensitive to lower temperatures (Supplementary Fig. 3; Supplementary Fig. 4) as well as to differences in the parametrisation of the economic model and their experimental design, based on a dynamic framework. Specifically, including socioeconomic dynamics into our model would imply lower economic damages associated with a higher adaptive capacity of regional economies (see Supplementary Discussion). This result, however, should be interpreted with caution, as is based on a future scenario (SSP5) featuring strong economic growth and technological

progress as well as rapid and costless adaptation. Carefully exploring different RCP-SSP combinations (now constrained by the lack of WBGT hourly data) would be helpful to assess the uncertainty posed by socioeconomic projections to the present results.”

25. Line 251: Was a statistical test for significance performed? If yes, which one and what are the p-values? If not, the word “significant” should be avoided here. In any case, I would suggest to perform a significance test.

In the previous version, we did not perform any statistical test to back up this argument. Below, the reviewer can find, for each country, the t-test and corresponding p-values of the significance tests for the damages in the decade 2055-2064. Specifically, we test $H_0: \mu = 0$ versus $H_1: \mu \neq 0$. Then, we build the t-statistic

$$T = \frac{\bar{X} - 0}{\hat{\sigma}/\sqrt{N}} \sim t_{N-1}$$

and calculate its p-value as

$$\hat{\alpha} = \Pr(T > \text{abs}(t))$$

Country	AVG 2055-2064	SD 2055-2064	t-statistic
AT	-0.706	0.181	12.325
BE	-0.143	0.053	8.587
CY	-3.376	0.338	31.581
CZ	-0.473	0.147	10.171
DK	-0.109	0.080	4.320
EE	-0.348	0.193	5.692
FI	-0.286	0.179	5.048
FR	-1.458	0.306	15.072
DE	-0.462	0.159	9.192
EL	-2.174	0.244	28.207
HU	-0.801	0.198	12.783
IE	-0.067	0.073	2.930
IT	-2.171	0.269	25.529
LV	-0.570	0.190	9.508
LT	-0.375	0.123	9.649
LU	-0.143	0.054	8.346
MT	-2.820	0.367	24.270
NL	-0.130	0.059	6.957
PL	-0.420	0.124	10.674
PT	-3.027	0.541	17.682
SK	-0.673	0.214	9.966
SI	-0.865	0.211	12.961
ES	-2.796	0.339	26.092
SE	-0.163	0.147	3.502

UK	-0.053	0.038	4.443
BG	-1.856	0.292	20.121
HR	-3.122	0.622	15.875
RO	-2.635	0.435	19.150
EFTA	-1.292	0.438	9.328

We compared the respective t-statistics with the percentile $t_{9,0.95}=1.8331$ and $t_{9,0.995}=3.2498$. As it can be seen, all the damages are statistically different from 0 for all significance levels (except for Ireland, p-value=0.01).

25'. As an additional comment to this question: Would it be possible to show the (interannual) variability of the future estimates in Figure 3b? I don't know if the figure would get too crowded in this case, but it would be an interesting addition to see how variable the GDP estimates are.

Thanks for the suggestion. It would be indeed interesting to explore the temporal variability of damages. We do so at the spatially aggregated level in Fig. 4a*, where interannual variability is shown per decade and climate model in each box. If we included the interannual variability of damages (along the decade 2055-2064) in Fig. 4b*, we would be interfering with the main objective of this figure, i.e., showing the cross-period/decadal variability in damages at the country-level (as is, for example, done here: <https://ec.europa.eu/eurostat/cache/RCI/#?vis=nuts2.economy&lang=en>). Therefore, we prefer this figure to remain as it looks now.

*Fig. 4 was Fig. 3 in the previous version of the manuscript.

26. Lines 272-276: Where is the evidence for this statement?

Please refer to comment #2.

27. Lines 283: I guess this should be something like “further adaptation” instead of “further climate action” as in the current formulation it contradicts the statement in lines 257-260, at least for the upcoming decades (of course not for the second half of the 21st century).

Thanks for this accurate remark. As we argue in the text (see also comment #9), no matter the emission scenario, the climate change signal will be very similar during the first part of the century. Hence, as the reviewer correctly points out, ‘climate action’ cannot mitigate the projected damages of heatwaves in the economy. We have substituted ‘climate action’ by ‘further adaptation to climate change’, as the reviewer suggests.

Changes in the manuscript (Lines 312-314): “Looking ahead, *the projected costs of heatwaves are expected to increase steadily in the next decades if no further measures to adapt to warmer temperatures are implemented in work environments.*”

28. Lines 292-297: Why would these indicators be more appropriate for quantifying urban heat island effects? Is that not possible with WBGT? And

I guess this also strongly depends on the resolution of climate models (e.g. 0.44° is too coarse to reliably estimate heat effects in cities) and on how urban areas are represented in climate models.

Different heat stress indices have different advantages and limitations. Here MTR and UTCI are mentioned because they may give some added value in the sense that they have a physiological component but, on the other hand, their connection to labour workability is limited and their systematic calculation with climate models is not straightforward. We have rephrased the affected part.

Changes in the manuscript (Lines 325-331): “Second, the incremental heat effect in urban areas (see Burgstall, 2019 and references therein), i.e., the urban heat island effect, and its potential large effects on, for example, construction workers, should be explored with the WBGT or alternative heat stress indicators including a human heat balance model, such as the Mean Radiant Temperature (MRT) and the Universal Thermal Climate Index (UTCI) (Di Napoli et al., 2020), always adopting enough spatial resolution and/or parameterised processes to capture heat signals at the city level.”

29. Lines 302-306: I think this is very crucial. In your study you perform a sophisticated analysis, considering hourly WBGT values and considering different effects in different regions and economic sectors. But for translating the heat impacts into economic impacts, the transfer functions are still rather simple. I think that a better quantification of the economic impacts due to heat would help to obtain a more sophisticated transfer function.

Exposure-response functions are considered a major bottleneck in this literature: they are poorly calibrated and hardly match a handful of economic activities. We point in the text to the possible way forward, based on more sophisticated, sector-specific functions based on field-level studies at the workplace. A brief comment has been added to reinforce this argument.

Changes in the manuscript (Lines 336-342): “Fourth, we used the existing state-of-the-art functions to translate heat stress into labour productivity impacts. Emerging field studies are beginning to provide evidence on the existence of heat impacts on productivity at temperatures below 26°C WBGT (Ioannou et al., 2017; Quiller et al., 2017; Flouris et al., 2018), featuring also notable disparities within economic sub-sectors. Nevertheless, the results from these field studies have yet to be translated into more sophisticated, sector-specific transfer functions.”

30. Lines 366-367: “a fair proxy” – how can this be said? Did you test that WBGT is indeed a fair proxy given the measures not considered in this study?

We are sorry for the confusion. We meant that WBGT, solely based on meteorological data, represents a good approximation of the real situation of heat stress suffered by the worker, either for indoor or outdoor conditions. We rephrased the sentence as follows:

Changes in the manuscript (Lines 413-415): “Thus, the inherent flexibility of the WBGT, which allows to account for heat stress either for indoors or outdoors with a single index, entail an important advantage.”

31. Line 382: I guess this should mean that it is a compromise of being close enough to the present to be relevant, but far enough so that the signal emerges from the noise, right? The statement about “strong influence of random internal variability” might not be understandable for all readers, and I would thus suggest to argue with the emergence of signals.

Thanks for the comment. We agree with the reviewer in that this statement might not be understandable for all readers. We changed the manuscript according to the reviewer’s suggestion and included the argument of emerging signals. See also comment #9 for further details.

Changes in the manuscript (Lines 428-431): “In the climate change exercise, we studied a period by mid-21st century (2035– 2064), which is a compromise of being not too distant for immediate action and far enough so that the climate change signal emerges from the internal climate variability in the models (Samset et al., 2020).”

32. Line 456: I would recommend to use a more sophisticated method for interpolation, as nearest neighbour interpolation is very simple and a lot of information might be lost if grid resolutions are very different. Moreover, it is not clear to me whether the climate data or the population data were interpolated.

We temporally (linearly) interpolated the population datasets, as they were provided on a 5-year interval.

Since workers are the cornerstone of our paper, we took the population data as our reference. Hence, WBGT data were assigned to each population pixel using the method described in the text (nearest neighbour). We agree with the reviewer in that more sophisticated interpolation methods, such as bilinear interpolation or conservative remapping, could yield more accurate results. However, we ask the reviewer to consider that the resulting interpolated data was further spatially averaged at the regional level, which certainly smoothed the biases and information losses incurred during the interpolation phase.

33. Line 465: What are calendar effects?

GDP is a synthetic measure of the economic activity carried out in a territory during a concrete period, typically a year. The flux of goods and services produced is not homogenous over the period. Some activities are concentrated in specific parts of the year: construction usually avoids very cold months, tourism services during summer, seasonal agricultural products,... only to name a few.

We make use of the Quarterly Accounts provided by Eurostat (available only from year 1995) to calculate the corresponding annual share of production of each of the five macro-sectors considered in our study. For example, if one area produces more agricultural goods in Q3 and that quarter has suffered more extreme

temperatures, then our representative annual shock to agricultural workers will take this into account, which results in more precision in the actual annual effect of heat on sectoral labour productivity. Bear in mind that the time scale of the economic model is annual.

34. Lines 512-516: The authors explain very nicely in their answer to the reviewer comments why such a high value is used as a barrier for trade in Europe. I would suggest to extend this paragraph a bit more to explain that it is difficult to obtain these values and to defend the usage of 20%

We thank the reviewer for this suggestion. We have integrated in the main text a part of the answer provided in the first revision to further clarify this point.

Changes in the manuscript (Lines 567-573): “We adopt a conservative choice on the value of Armington elasticities for sub-national units because we focus on the short-term economic consequences of heatwaves, when trade frictions can be high. This value is meant to provide only a reference to make trade more fluid within a country than between countries. Further research and a sensitivity analysis on this value would be certainly valuable but is out of the scope of the paper.”

35. Figure 2b: I think that only few readers are familiar with the region abbreviations used here. I would thus strongly suggest to either write the country names (plus maybe the region code in parentheses) or add a map that shows where the regions are located. Although the information is also contained in the supplementary information, several readers might not take the effort to check that.

Thanks for the suggestion. Fig. 2b (now, Fig. 3) has been expanded with more regions. Full names are provided as well as country code between parentheses. See Fig. R1 below.

Figure R1: New Figure 3 in the revised manuscript.

To Reviewer #3's comments

I find the paper a very well written piece research with three original and important contributions:

i. Assessment of heat effects on workability, and hence economic performance, at regional scale (NUTS2) for Europe; while climate hazards and health impacts have been analyzed at high spatial resolution, an economic assessment at regional scale (below nation scale) so far was missing and is an extremely valuable and notable contribution

ii. Analysis based on different impacts functions, i.e. heat to workability transfer functions (using ISO vs Hothap vs NIOSH); impacts are first calculated on seasonal scale (particularly relevant for sectors like agriculture and construction), only afterwards aggregated to yearly scale

iii. Comparison of decadal projections with past heat wave years (2003, 2010, 2018) which is quite unique (usually you find either econometric assessments of past events or simulation based projections for the future)

Overall, the paper is very well written and the analysis is rigorously conducted. However, I would like to echo two previous reviewer requests: you should make more out of the regional economic analysis and you should carefully investigate scenario and model uncertainties. In addition, I find the assumption of "fixed economic activity" in the simulations for the future as problematic. Here are my suggestions in more detail:

Many thanks to this reviewer for his comments and the positive feedback and for pointing out the contributions of our work.

1. Climate change impacts/risks are the combination of hazard, exposure and sensitivity. In Fig. 1, hazard is illustrated for past heatwaves. Fig. 2 illustrates GDP effects split up by latitude and main economic sectors. I would find it more informative to present economic impacts also as map as there is not only a north-south gradient in exposure but countries/regions differ also in economic exposure (which economic sectors dominate?) and sensitivity (effect of climatization, adaptive capacity etc.)

Thanks for the comment. As the reviewer mentions, risks/impacts are the combination of three elements. In our paper, we dissect the contributions of the first two. First, we look at the hazard and its spatio-temporal characteristics (Fig. 1). (Economic) exposure is also accounted for by considering outdoor/indoor activities and different working intensities. We attribute the occurrence of high impacts to the joint effect of these two (see, for example, Supplementary Fig. 2), as we indicate in different parts of the text. We explicitly acknowledge the effect of the third ingredient (the reviewer refers to it as 'sensitivity') in the text (lines 291-294), which was not analysed due to the difficulty to obtain reliable measures.

As the reviewer suggests, we have modified the structure of Fig. 2. Now, Fig. 2a features a panel of maps showing regional impacts in the in-depth analysed years. Simultaneously, we have enlarged and moved the second part (previously, Fig. 2b), which is now a self-standing figure (Fig. 3b). In this way, we reinforce our focus on the regional and sectoral dimensions of impacts.

We would like to highlight the contribution of the, in our opinion, quite visually innovative Fig. 2b (previously, Fig. 2a). This figure shows not only a north-south gradient in impacts, but also how impacts tend to be always higher (as a share of regional output) in southern latitudes, even when heat anomalies are greater in the north. Besides that, Fig. 2b also captures the cross-year variation in impacts for all latitudes and, lastly, it lets the reader situate how hard regional economies were hit, relative to the distribution of historical damages affecting each latitude. This last feature has been possible after estimating heatwave impacts all over the time series 1981-2010 (upon request from Reviewer #2).

2. Uncertainties matter not only in terms of RCP scenarios, GCMs selected, and impact function used (ISO vs Hothaps vs NIOSH) but also in terms of SSP scenarios. This has been demonstrated in the context of health e.g. in Rohat et al. (2019) for Europe. It is therefore common practice to investigate different RCP-SSP combinations by drawing on IIASA's SSP marker database. For heat driven impacts on occupational health, the most important variables to consider would be not only population but also economic growth (available both at NUTS0 level from this database, but regionalizations for Europe exist, see e.g. Rohat et al. 2018; Kok et al. 2019 and various deliverables from the IMPRESSIONS project). For instance, analysis in the context of water scarcity have shown that e.g. a move of people from Eastern Europe towards Western Europe has quite substantial consequences for risks (Harrison et al. 2019).

We thank the reviewer for raising a very important point, which we have carefully considered and explored running additional simulations. The additional simulations have been time consuming, but also useful to better understand the importance of the socio-economic dynamics that the reviewer was accurately mentioning.

First of all, we must note that the physical impacts of climate change considered in our study (based on the RCP8.5 scenario) are only compatible with the socioeconomic scenario SSP5 (as illustrated by Fig. R2). This limits the possibilities of considering different RCP-SSP scenarios to a single combination, namely, RCP8.5-SSP5. We replicated in our model the GDP and population patterns implied by SSP5, as obtained from the IIASA SSP database (<https://tntcat.iiasa.ac.at/SspDb/dsd?Action=htmlpage&page=10>), assuming that the stock of labour follows the same growth pattern shown by population and that subnational regions follow the projections of the country they belong to. We imposed exogenously the population and labour dynamics in the CGE and calibrated the GDP using the Total Factor Productivity (TFP) to meet the SSP5 targets. Meanwhile, capital accumulation takes place endogenously in the CGE via the recursive addition of investment coming from the previous period.

Figure R2: (from Rogelj et al. 2018, Supplementary Information) Overview of available scenario runs in the SSP-RCP matrix framework. Values in each box represent the number of available scenario runs over the number of participating modelling frameworks. Given that used climate data forced by RCP8.5, the only SSP scenario compatible with this data is SSP5.

The results of this additional experiment offer interesting insights. We observe that, in general, economic losses tend to be lower in Europe (Fig. R3) compared to the previous set of results (Fig. 4a) based on a comparative static framework.

Figure R3: Evolution of economic damages in Europe considering a dynamic framework RCP8.5-SSP5 characterised by a push for economic and social development coupled with the exploitation of abundant fossil fuel resources and absence of climate policies.

The main driver for this difference is the fact that capital becomes more important in the structure of the future economy. Since the productivity impacts of heatwaves only affect labour, this has a direct consequence on the final allocation of factors (Fig. R4). In this setup, the exogenous growth in population (labour) is not sufficient to guarantee a high macroeconomic growth path, as imposed by the SSP5. Hence, capital grows more relative to labour and accumulates in the economy. Thus, the decrease in total damages in response to labour shocks. This higher adaptive capacity shown under the combination RCP8.5-SSP5 can also be observed, for example, in Orlov et al. (2020).

Figure R4: Trends (% variation) of European GDP and their components over the period 2036-2064 under SSP5 dynamic scenario. L denotes Labour; K, capital; TFP, Total Factor Productivity.

Changes in the demographic composition across regions play a limited role because, in general, population is expected to grow more in Northern Europe, it shows a moderate growth rate in Mediterranean Europe and remains stable or even declining in Eastern Europe. Since the hardest impacts of heatwaves are concentrated in Southern Europe, this pattern does not influence dramatically the final macroeconomic outcome. Instead, one key driver appears to be the sectoral structure of the economy, especially the weight of outdoor and indoor economic activities in the future. While in the comparative-statics experiment the sectoral composition remains fixed, in the SSP5 baseline it evolves endogenously. In the dynamic setup, the economy develops according to the GDP and population targets, which are exogenous in the SSP5 but the sectoral composition responds to different market mechanisms, which are all endogenous and confounded in the CGE. These market mechanisms refer to trade specialization (based on regional comparative advantages), primary factor reallocation across sectors, investment dynamics and how all these forces interact with the GDP and population targets. This makes the economic structure of some sub-national regions to change substantially over time leading to, for example, a retreat of impacts in some southern European economies in the last decade of our analysis (Fig. R5).

Figure R5: Evolution of heatwave-derived economic damages at the country level under the RCP8.5-SSP5 dynamic scenario.

One way to restrict severe sector reallocations would be to calibrate the sectoral economic composition of the NUTS-2 regions, but this could be computationally infeasible and would also imply a certain degree of arbitrariness, as SSPs do not provide information about the sectoral evolution.

These results point us to existing endogenous mechanisms through which the European economy would partly absorb the projected increasing heat load of work. However, results should be interpreted cautiously. Some aspects should be controlled with more detail as, for example, the evolution of the sectoral economic composition. In addition, because assumptions of demographic and economic developments over long-time spans are highly uncertain (Dellink et al., 2017; Christensen et al., 2018), it would be desirable to perform a comprehensive assessment covering the whole spectrum of RCP-SSP scenarios to account for these uncertainties.

In light of the above and considering that it is unfeasible for us to explore several representative RCP-SSP combinations (mainly because of data restrictions), we prefer to stick to the original results (those described by a static economy). By embracing the static approach, we believe our results are more intelligible and can be better compared across different years.

We have included a clarification about this topic in the main text (see below). In addition, a summary of the findings of this experiment have been made available in the Supplementary Information (see Supplementary Discussion).

Changes in the manuscript (Lines 262-269): “Specifically, including socioeconomic dynamics into our model would imply lower economic damages associated with a higher adaptive capacity of regional economies (see Supplementary Discussion). This result, however, should be interpreted with caution, as is based on a specific future scenario (SSP5) featuring strong economic growth and technological progress as well as rapid and costless adaptation. Carefully exploring different RCP-SSP combinations (now constrained by the lack of WBGT hourly data) would be helpful to assess the uncertainty posed by socioeconomic projections to the present results.”

3. In addition, while the assumption of "holding fixed the current economic development" is still employed in some CGE studies (e.g. Ciscar et al. 2014 and subsequent PESETA projects), today the majority of CGE modeling exercises integrate RCP-SSP scenarios based on information on GDP growth, fossil fuel use, population growth etc. This "everything remains equal" assumption is particularly problematic as both exposure and sensitivity depend on economic structures and where the (working) population lives. Takakura et al. (2019) demonstrate that for Europe, half of the variance by 2050 is contributed by socioeconomic development. The argument put forward by the authors of the current paper that the same economic structure allows for comparability can be easily countered and addressed by the usual approach of comparing e.g. RCP4.5-SSP2 or SSP2 with no (additional) climate change.

We thank the reviewer again for the valuable comment, tightly connected to the previous one. As noted in the previous reply, we also think that it is important to include socio-economic future trends in this experiment. We have carried out the additional simulations for SSP5 to understand better the role played by these future trends in our work. In this sense, one important element is the higher weight of capital in the future economy as a result of the SSP5's GDP and population targets. This element can be easily understood and communicated and explains relatively well the reduced macroeconomic impacts of heatwaves in the future. However, in our study we are forced to refer to one single SSP-RCP combination, namely SSP5-RCP8.5, and this limits substantially the significance of the analysis on the role played by socio-economic dynamics. We think that it can be misleading to generalize the results we observe with this specific combination as the general socio-economic development in the coming decades. Moreover, the temporal evolution of indoor and outdoor activities is endogenous in the CGE model and we cannot properly control for it. This also creates a potential bias, especially in some EU sub-national regions, which can generate some confusion for the readers.

Since the reviewer cites the work by Takakura et al (2019), we would like to highlight that in that paper it can be found that

“Most of the uncertainty (variance) in the projected impacts is attributable to anthropogenic factors (SED(=SSP) and CCM(=RCP) pathways), particularly in the latter half of the twenty-first century, and the contribution of the RCP pathway is the greatest.”

which suggests a more prominent role of socio-economic uncertainties beyond our studied time period. And in Takakura et al (2017)

“Until the middle of the 21st century, when the climate is not yet stabilized, the difference in GCM is the primary contributor to the variance of the result of the global total GDP loss rates. On the other hand, at the end of the 21st century, when the climate is almost stabilized except RCP8.5, RCP is the primary contributor to this variance. The proportion of the variance due to GCM and RCP is more than 80%, whereas that due to SSP is less than 5% throughout the simulation periods.”

We added the lack of a comprehensive dynamic assessment as one of the limitations of our study and as an avenue for future research.

Changes in the manuscript (Lines 342-349): “Fifth, a comprehensive assessment based on the combination of future climate and socioeconomic pathways must be carried out in order to obtain a complete overview of the distribution of future heatwave impacts and their associated uncertainties, especially during the second part of this century. However, for this kind of assessment, hourly-level and spatially downscaled projections of heat stress measures should first be available under different climate forcings.”

In addition, I have a couple of smaller points:

4. In the model base year (2007), the damages are already included in the economic data, i.e. input output tables report economic activity considering these damages. So all WBGT/WBDD values in the heat wave years 2003, 2010, 2015, 2018 need to be expressed relative to this base year, not to the base period 1981-2010. The same holds also for impacts in the base period. My presumption is that changing this will not alter results significantly but it matters in terms of consistency of assumptions.

The base year for calibrating the model is 2007. The reviewer correctly points out that the base year is already inherently endowed with a specific heatwave shock/component. Our point is that this effect is impossible to be disentangled from the myriad of shocks that also simultaneously affected the European economy during the calibration year.

To have an idea of the potential bias caused by the calibration year, we simulated/shocked the model in 2007 according to the heat stress experienced that year. The outcome we obtained was that Europe experienced aggregate losses of -0.1836% GDP, a figure slightly below the historical average over 1981-2010 (-0.2039%), denoting that 2007 was a moderate-low year in terms of heat load. This leads us to think that the inherent bias in our results can be relatively low.

Also, since heatwaves are defined as extreme events by nature, it makes more sense to compare single years to climatological periods, as we do in Fig. 2b. There we compare the impacts in analysed years with respect to the reference historical period 1981-2010.

A brief comment has been added in the methods to clarify this point.

Changes in the manuscript (Lines 524-528): “The choice of the calibration year might bias to some extent the outcome of our simulations, as the regional database already incorporates the effect of heatwaves in the economy during the calibration year. We expect, however, this bias to be low in our case, as 2007 was in general a lower-than-average year in terms of heat load”.

5. In principle, the setup would allow for a re-analysis of observed impacts by comparing them to the simulated damages with the CGE model. While such an analysis would constitute a paper in itself, it would be good to compare the scale of simulated damages to reported damages in media and elsewhere (e.g. in the introduction).

Thanks for the suggestion. Indeed, it would be very interesting to compare the simulated results with observed/documented damages. However, we are afraid this would not be a straightforward task and lies beyond the scope of our paper. Heatwaves are climatic risks that trigger impacts in many dimensions beyond the one studied in this paper. For example, they can affect the productivity of land or the amount of water resources available. This means that, even if we collected damages reported during heatwaves, it would be hard to compare them to our labour-only impacts.

In any case, we compare our findings with the results obtained by other scientific papers that have addressed this issue in Europe (Orlov et al., 2019) or countries of Europe (Knittel et al., 2020).

6. Overall, I suggest to strive for consistency in terms of either reporting regional effects or country effects both for past heat waves (figs. 1c, 2b) and future simulations (Fig. 3b). Personally, I would also present regional effects (NUTS1 level) always in the form of maps (not as bar diagrams as in Fig. 2b). I also find the reporting of sectoral effects more relevant at the national scale than the regional scale. But this is a matter of taste.

Differences in how damages are reported are intentional. In general, we choose to communicate our results using a regional focus on past assessments (based on realised weather), while we shift the focus to coarser spatial units on climate change assessments (based on projections and thus more uncertain). Not only that, by adopting the country-level perspective in the climate change analysis, we are able to show what will happen in countries projected to be less affected by this risk, let this be because of their expected lower hazard risk or lower exposure. This is also an important message of our paper.

To accommodate the suggestion of the reviewer, we have included a new Figure in the Supplementary Material (Supplementary Fig. 5), where we analysed the regional evolution of damages in a handful of southern countries, evidencing the cross-regional variation of damages within largely affected countries (see Fig. R6).

Figure R6: Regional-level projected impacts of heatwaves in France. In the Supplementary material, we report projected regional damages in France, Italy, Spain and Greece (Supplementary Fig. 5).

7. In terms of balance between past and future effects, I find the current presentation tilted too strongly towards the past, and also towards hazards. E.g. the information that both the duration and the severity of heat waves matter is important, but in the subsequent analysis only WBDD are used. So I would move Figs. 1a+b into the supplementary material. For understanding future risks, some information is however needed on how exposure and sensitivity differs across European regions and how these socioeconomic conditions change under different SSPs. This could be added to the main text as figure and used to explain differences in results across regions/countries.

The reviewer has correctly spotted that the balance of this paper is more oriented towards the analysis of past events rather than future events. This lack of balance is intentional, as we found necessary to primarily pursue an accurate quantification of the current costs of heatwaves, based on spatio-temporal precision, sectoral decomposition, and by properly delimiting the time spells of heatwaves. This is also why we devote one section of the paper to an in-depth analysis of the hazard. There we show, for example, that heatwaves extend before and after the summer season or that different years can show very different regional patterns in terms of the duration and intensity of events. In sum, the first section deploys a comprehensive regional characterisation of heatwave events in Europe, which has an intrinsic interest in our opinion.

Regarding the determinants of damages, we identify heat and economic exposure as their key drivers. We classify regions according to the combination of these two variables, as shown in Supplementary Fig. 2. In that figure, we also speak about the possible transition of regions between groups in the near future due to increasing warming.

8. Fig. 1c: I would find WBGT (or WBDD) reported as anomalies (delta approach) easier to understand than the absolute WBDD values for the years. It would be good to have similar maps for future periods (e.g. as Supplementary material). In the caption, there is a typo (“sun version” instead of “sum version”).

WBDD are, by construction, cumulative anomalies: cumulative differences of the observed WBGT values with respect to a certain temperature threshold (26°C) during working hours.

The term “sun WBGT” shown in the caption is not a typo. It is used to show that we used the ‘sun’ version of WBGT in the calculation of WBDD.

9. Fig. 2a: as argued above, I think there are better ways to explore economic impacts across regions (e.g. a panel of maps instead of bar chart).

We have tried to accommodate this suggestion by including more maps in our analysis. See response to comment #1.

10. Fig. 3a: why do you have different box plots for the two GCMs? Usually different GCMs and impact models are collated within one box plot, but different plots are used for different RCP-SSP scenario combinations (which you currently do not have). Is the difference in mean values between different decades statistically significant?

By using boxplots for each candidate model, we are able to simultaneously describe climate model interannual variability and climate variability between different models. Then, we identify a common trend in damages over time.

Here we provide the test for statistical significance between mean decadal differences.

$$H_0: \mu_A - \mu_B = 0$$

$$H_1: \mu_A - \mu_B \neq 0$$

T-statistic is built

$$T = \frac{(\bar{X}_A - \bar{X}_B) - 0}{S_p \sqrt{\frac{1}{N_A} + \frac{1}{N_B}}} \sim t_{N_A + N_B - 2}$$

with

$$S_p^2 = \frac{(N_A - 1)S_A^2 + (N_B - 1)S_B^2}{N_A + N_B - 2}$$

And calculate its p-value as

$$\hat{\alpha} = \text{Pr}(T > \text{abs}(t))$$

$\mu_A - \mu_B$	$ t $	$N_A + N_B - 2$	p-value
(2001-2020) – (1981-2000)	3.2396	30	0.000
(2021-2034) – (2001-2020)	5.0578	24	0.000
(2035-2044) – (2021-2034)	6.8030	32	0.000
(2055-2044) – (2035-2044)	3.4151	38	0.000
(2065-2054) – (2055-2044)	1.3344	38	0.100
(2065-2054) – (2055-2044) [KNMI-RACMO]	2.7411	18	0.000

We can conclude that all the decadal mean differences are significantly different from 0. For the last decade, the pooled difference between periods shows a p-value equal to 0.1, due to the large interannual variability of the MPICSC-REMO2 model. However, the signal provided by the KNMI-RACMO is univocally increasing in economic damages.

11. Regarding the contribution to the most related literature, in addition I would also add Takakura et al. (2017) who conduct a global assessment and also look into sectoral effects.

We have revised the series of papers on the topic by Takakura and co-authors (2017, 2018, 2019). We must note that the approach from Takakura and colleagues differs substantially from ours. For example, in their 2017 paper, they write

“The estimated economic cost should not be interpreted as an estimation of GDP loss simply caused by heat-related stress in workers. Heat stress can reduce the efficiency of work [40, 41], but this type of reduction differs from the recommended reduction in worktime considered in the present study.”

What these authors are quantifying is

“the relationship between the economic cost of heat-related illness prevention through worker breaks and climatic/socioeconomic conditions.”

They obtain their measure of cost as the additional wages required to compensate the work time loss associated with the additional labour requirements.

We have cited Takakura et al (2017) in the main text as an example of the possible effects of heat stress in the labour market.

Changes in the manuscript (Lines 47-52): *“Excessively hot environments are precursors of biophysical and cognitive impacts, causing physiological strain to workers (Ioannou et al., 2021), lowering the number of hours of work supplied (Takakura et al., 2017), affecting the capacity of assimilating information (Park et al., 2020) and interfering with decision-making (Heyes and Saberian, 2019), ultimately undermining human capital accumulation and, therefore, economic growth”.*

References

- Bilbao J, Miguel AH, Kambezidis HD (2002) Air temperature model evaluation in the North Mediterranean Belt Area. *J Appl Meteorol* 41:872–884.
- Casanueva A, Kotlarski S, Fischer A, Flouris A, Kjellstrom T, Lemke B, Nybo L, Schwierz C, Liniger M (2020) Escalating environmental summer heat exposure - a future threat for the European workforce. *Regional Environmental Change* 20(40):1–14.
- Christensen P, Gillingham K, Nordhaus W (2018) Uncertainty in forecasts of long-run economic growth. *Proc. Natl Acad. Sci. USA* 115, 5409–5414.
- Dellink R, Chateau J, Lanzi E, Magné, B (2017) Long-term economic growth projections in the shared socioeconomic pathways. *Glob. Environ. Change* 42, 200–214.
- EEA, European Environment Agency (2019) Indicator Assessment. Global and European temperatures. <https://www.eea.europa.eu/data-and-maps/indicators/global-and-european-temperature-8/assessment>.
- Foster J, Smallcombe JW, Hodder S, Jay O, Flouris AD, Nybo L, Havenith G (2021) An advanced empirical model for quantifying the impact of heat and climate change on human physical work capacity. *Int J Biometeorol* <https://doi.org/10.1007/s00484-021-02105-0>.
- Giorgi F (2010) Uncertainties in climate change projections, from the global to the regional scale. *EPJ Web of Conferences* 9 115-129. <https://doi.org/10.1051/epjconf/201009009>.
- Harrison PA, Dunford RW, Holman IP, et al (2019) Differences between low-end and high-end climate change impacts in Europe across multiple sectors. *Regional Environmental Change* 19:695–709. <https://doi.org/10.1007/s10113-018-1352-4>.
- Hawkins E, Sutton R (2009) The potential to narrow uncertainty in regional climate predictions. *Bull Am Meteorol Soc.* 90(8): 1095–1108 <https://doi.org/10.1175/2009BAMS2607.1>.
- Ioannou LG, Mantzios K, Tsoutsoubi L, Panagiotaki Z, Kapnia AK, Ciuha U, Nybo L, Flouris AD, Mekjavic IB (2021) Effect of a Simulated Heat Wave on Physiological Strain and Labour Productivity. *Int J Environ Res Public Health*. 2021 Mar 15;18(6):3011. doi: 10.3390/ijerph18063011.
- IPCC (2013) *Climate Change 2013: The Physical Science Basis. Contribution of Working Group I to the Fifth Assessment Report of the Intergovernmental Panel on Climate Change* [Stocker, T.F., D. Qin, G.-K. Plattner, M. Tignor, S.K. Allen, J. Boschung, A. Nauels, Y. Xia, V. Bex and P.M. Midgley (eds.)]. Cambridge University Press, Cambridge, United Kingdom and New York, NY, USA, p 1535.
- Kjellstrom T, Freyberg C, Lemke B, Otto M, Briggs D (2018) Estimating population heat exposure and impacts on working people in conjunction with climate change. *International Journal of Biometeorology* 01 3(62):291–306.

Knittel N, Jury MW, Bednar-Friedl B, Bachner G, Steiner AK (2020) A global analysis of heat-related labour productivity losses under climate change—implications for Germany's foreign trade. *Climatic Change* 160:251–269.

Kok K, Pedde S, Gramberger M, et al (2019) New European socio-economic scenarios for climate change research: operationalising concepts to extend the shared socio-economic pathways. *Regional Environmental Change* 19:643–654. <https://doi.org/10.1007/s10113-018-1400-0>.

O'Neill BC, Kriegler E, Ebi KL, et al (2017) The roads ahead: Narratives for shared socioeconomic pathways describing world futures in the 21st century. *Global Env Change* 42, 169–180. <http://dx.doi.org/10.1016/j.gloenvcha.2015.01.004>.

Orlov et al (2019) Economic Losses of Heat-Induced Reductions in Outdoor Worker Productivity: a Case Study of Europe. *Economics of Disasters and Climate Change* 3, 191–211. <https://doi.org/10.1007/s41885-019-00044-0>.

Orlov A, Sillman J, Aunan K, Kjellstrom T, Aaheim A (2020) Economic costs of heat-induced reductions in worker productivity due to global warming. *Global Environmental Change* 63:102087.

Rogelj J, Popp A, Calvin KV, et al (2018) Scenarios towards limiting global mean temperature increase below 1.5 °C. *Nature Clim Change* 8, 325–332.

Rohat G, Flacke J, Dao H, van Maarseveen M (2018) Co-use of existing scenario sets to extend and quantify the shared socioeconomic pathways. *Climatic Change* 151:619–636. <https://doi.org/10.1007/s10584-018-2318-8>.

Rohat G, Flacke J, Dosio A, et al (2019) Influence of changes in socioeconomic and climatic conditions on future heat-related health challenges in Europe. *Global and Planetary Change* 172:45–59.

Samset BH, Fuglestad JS, Lund MT (2020) Delayed emergence of a global temperature response after emission mitigation. *Nature Communications* 11, 3261. <https://doi.org/10.1038/s41467-020-17001-1>.

Schifano P, Asta F, Marinaccio A, et al (2019) Do exposure to outdoor temperatures, NO₂ and PM₁₀ affect the work-related injuries risk? A case-crossover study in three Italian cities, 2001–2010 *BMJ Open* 2019;9:e023119.

Stott P, Stone D, Allen M (2004) Human contribution to the European heatwave of 2003. *Nature* 432, 610–614.

Takakura J, Fujimori S, Takahashi K, et al (2017) Cost of preventing workplace heat-related illness through worker breaks and the benefit of climate-change mitigation. *Environmental Research Letters* 12:064010.

Takakura J, et al (2018) Limited Role of Working Time Shift in Offsetting the Increasing Occupational-Health Cost of Heat Exposure. *Earth's Future*, 6.

Takakura J, Fujimori S, Hanasaki N, et al (2019) Dependence of economic impacts of climate change on anthropogenically directed pathways. *Nature Climate Change* 9, 737–741. <https://doi.org/10.1038/s41558-019-0578-6>.

Vogel MM, Zscheischler J, Wartenburger R, Dee D, Seneviratne, SI (2019) Concurrent 2018 hot extremes across Northern Hemisphere due to human-induced climate change. *Earth's Future*, 7, 692– 703.

REVIEWER COMMENTS

Reviewer #1 (Remarks to the Author):

Thank you for considering the comments to the previous version of the manuscript. I appreciate the effort made by the authors. However, I am not yet convinced by the rebuttal and feel that the manuscript can be improved by adding more descriptions. Please see the comments below.

In the rebuttal, the authors stated "This paper provides accurate estimates of the economic burden of heatwaves in Europe. Compared to previous studies, our methodology shows a greater level of spatial (regional level), temporal (climate hourly data) and sectoral detail and is based on an economic model specifically regionalised and calibrated to reproduce the behaviour of the European economy". I understand that the authors used higher spatial and temporal resolution data, but this does not guarantee the estimates are more accurate. Higher resolution data requires more parameters in the models and some of the parameters were not directly available in the statistics and need to be estimated. This possibly introduced errors and uncertainty in the estimate rather than reducing them. If the authors claim that their estimates are more accurate compared to those of the previous studies and it is the contribution of this study, the accuracy should be demonstrated empirically.

"For a detailed list of all the methodological contributions and novelties of our paper, we refer the reviewer to the previous review, in which all of them were extensively described." Yes. In the rebuttal, the authors described these points, but not in the manuscript. It is important to review the current state of the research filed and explain how the authors addressed the existing issues to the readers of the paper, not only to the referees.

Reviewer #2 (Remarks to the Author):

I thank the authors for the detailed answers to my comments and for updating the manuscript. I only have a few specific comments left.

- Line 31: The connection to the sentence before should be more emphasized. Maybe using "selected years" instead of "analysed years" or "analysed years with anomalous heatwaves" or something similar.
- Line 61: I guess it should be "strongly" or "largely" instead of "partly". I assume that climate change is the main reason for the doubling.
- Line 62-64: This sentence is not entirely clear to me due to its structure. Particularly, what does "more than doubled the risk of heatwaves for some locations in particular events" mean? Please rephrase this sentence to make it better understandable.
- Line 98: I would add "forced by GHG emissions following the Representative Concentration Pathway 8.5"
- Line 101: I guess this sentence refers to the signal-to-noise ratio, but the statement is a bit misleading since internal variability will also play an important role in the future. A similar statement in lines 439-440 is formulated more adequately. I would suggest to write this sentence here in a similar way as the one in lines 439-440.
- Line 103: I would add "the latter uncertainty increases"
- Line 130: Longer than what? Please indicate.
- Line 145-146: "sustained positive temperature anomalies"
- Line 169: I would add that the losses mentioned here (i.e., the ones mentioned in parentheses) are also due to heat
- Figures 1c and Figures 2a: I find it interesting that the patterns show strong differences for some regions (e.g. for Northern Italy in 2003 or for Croatia in 2003 and 2015). It might be worth to write a few remarks about these differences in the text.
- Line 214: Rather "and" than "or"

- Line 220: Maybe I have missed it, but what is the reason that the ISO standard is selected for this study and not, e.g. Hothaps? This should be shortly explained.
- Line 233: I would find the formulation "do likely not have a significant impact" more intuitive
- Line 264: potentially highlight that ISO is the one used in this study.
- Line 267: Who uses the dynamic framework? Orlov et al or this study? Please specify this more clearly in the sentence.
- Line 276-277: "southern European countries"
- Line 287: Why is the p-values 0? It should be larger than 0. Is this a mistake? If it is very low, it could be indicated as $p < 0.001$.
- Line 304: I would mention here the years that were analysed.
- Line 343: "Do likely not" instead of "does not seem", since from your response to my previous comments I understand that the effect of adaptation measures were not tested.
- Line 514: It is still unclear to me whether the population or the climate data were spatially interpolated. Please be more specific here! And if it is the population data, nearest neighbour interpolation could cause that the total population in Europe is not conserved. In that case I would strongly recommend to use conservative remapping.
- Line 520-521: Remove "to these population projections" since also the interpolation of climate data is mentioned in this sentence.
- Figure 2: What does "(50th percentile)" at the end of the caption refer to? If it is just the definition of the median, I think it can be omitted here as people should know what the median is. If it is something else, it should be explained more precisely.
- Figure 4a: What do the boxes and whiskers indicate? The red line should also be explained.

Reviewer #3 (Remarks to the Author):

Most of my points have been properly assessed, reanalyzed and changed in the manuscript, except for my comment of considering different RCP-SSP combinations:

The population projections and GDP growth rates both affect exposure and vulnerability across Europe - and one important aspect here is the regional distribution of population (e.g. depopulation of Eastern Europe, movements towards Western/Central). Ideally, also assumptions on structural change would be incorporated, e.g. more fossil based energy sector in SSP5. While there is little difference in RCPs until mid century, looking only into RCP8.5, and therefore ruling out other SSPs, is a too simple procedure.

I do agree that RCP8.5 is most consistent, according to IAM evidence, in terms of emissions only with SSP5, but several papers also look into the combination with SSP3. Alternatively, plausible combinations of RCPs and SSPs are assessed, e.g. RCP2.6-SSP1, RCP4.5-SSP2, RCP8.5-SSP5.

To conclude: while I understand that a new analysis based on other RCPs is beyond the scope of the paper, the conclusion that "including socioeconomic dynamics [...] would imply lower economic damages" cannot be drawn by only looking into SSP5. Also the conclusion that increases in adaptive capacity more than offset increases in exposure seems a strong conclusion. My suggestion is therefore to delete lines 267-272 and to expand on the caveat in lines 272-275 by describing potential mechanisms that could reduce or amplify damages. Similarly, in the conclusion the half-sentence "especially during the second half of the century" in line 355 should be deleted, because SSP differences are already pertinent earlier. I also do not agree that "hourly-level and spatially downscaled projections of heat stress measures" are the most pertinent gap for doing different RCP-SSP combinations. It is rather that existing regional SSP narratives for Europe (e.g. Kok et al. 2019) need to be expanded towards structural economic change.

Finally, one minor point is that in Knittel et al. (2020) GDP effects reported for Germany are that of labor productivity changes on a global scale, so also changes embodied in imports. Please adjust lines 289-290 accordingly.

References cited:

Kok, K., Pedde, S., Gramberger, M. et al. New European socio-economic scenarios for climate change research: operationalising concepts to extend the shared socio-economic pathways. *Reg Environ Change* 19, 643–654 (2019). <https://doi.org/10.1007/s10113-018-1400-0>

Response to Reviewers

We thank the reviewers for their additional and constructive feedback, which has further strengthened the article.

Please find our point-to-point responses to each comment below. To facilitate the work of the reviewers, each comment has been repeated and our responses inserted after that (in blue).

Sincerely,

The authors

To Reviewer #1's comments

Thank you for considering the comments to the previous version of the manuscript. I appreciate the effort made by the authors. However, I am not yet convinced by the rebuttal and feel that the manuscript can be improved by adding more descriptions. Please see the comments below.

1. In the rebuttal, the authors stated “This paper provides accurate estimates of the economic burden of heatwaves in Europe. Compared to previous studies, our methodology shows a greater level of spatial (regional level), temporal (climate hourly data) and sectoral detail and is based on an economic model specifically regionalised and calibrated to reproduce the behaviour of the European economy”. I understand that the authors used higher spatial and temporal resolution data, but this does not guarantee the estimates are more accurate. Higher resolution data requires more parameters in the models and some of the parameters were not directly available in the statistics and need to be estimated. This possibly introduced errors and uncertainty in the estimate rather than reducing them. If the authors claim that their estimates are more accurate compared to those of the previous studies and it is the contribution of this study, the accuracy should be demonstrated empirically.

Thanks for the comment. The ‘accuracy versus resolution’ discussion is indeed relevant and we agree that the use of the term “accurate” could perhaps be misleading because, as this reviewer indicates, the proliferation of parameters in regional models and other factors, such as climate downscaling techniques, could bring uncertainty to some of our estimates. We are confident that the benefits of our detailed spatial, temporal, and sectoral approach do largely outweigh the costs that regional data/models can pose but, as this reviewer says, accuracy gains should be demonstrated.

Please note, however, that we used the term “accurate” only in our reply to reviewers (previous response). The term is not used anywhere in the manuscript, nor we state anywhere in the text that the present estimates are more accurate than others; we just highlight the improvements provided by our approach in relation to its higher spatial detail. For example, in both the Introduction and the Conclusion, we argue that the present work has the advantage of providing higher spatial resolution, which can be useful for the identification of vulnerable areas and can help policymakers in the formulation of evidence-based, local-level occupational/adaptation policies.

2. “For a detailed list of all the methodological contributions and novelties of our paper, we refer the reviewer to the previous review, in which all of them were extensively described.” Yes. In the rebuttal, the authors described these points, but not in the manuscript. It is important to review the current state of the research field and explain how the authors addressed the existing issues to the readers of the paper, not only to the referees.

We thank the reviewer for pointing this out. In the present version of the manuscript, all the novelties of our approach are mentioned (either in the main text or in the Methods).

Please find below a table with the identified methodological/non-methodological contributions and where to find them in the paper (contributions are underlined in the right-hand side column).

Contribution of the paper	Where is it mentioned?
In our methodology, a complete characterisation of heatwaves is present for all the analysed years. Heat stress is accounted for only at instances of time where temperatures were above the 90th percentile of maximum temperatures for at least three consecutive days. As such, we study the impacts of extreme heat, understood as periods when a region’s temperatures are abnormally high compared with the average, caring for the additional effect of low probability deviations from the average of the climate distribution rather than measuring the average effect of temperatures on the productivity of workers in some months.	(In Results: ‘More exposed regions spearhead economic losses’, L.204-209) “However, this work differs from the previous literature in several aspects. Among them, we study the impacts of extreme heat, understood as periods when a region’s temperatures are abnormally high (rather than measuring the effect of summer average temperatures), consider all the productive economic sectors, and adopt a higher spatio-temporal resolution level.”
Due to using a relative threshold for the identification of heatwaves, we can study heatwave events taking place at any time of the year. As evidenced by Fig.1a and Fig.1b this is quite relevant, since not only summer months experience heatwaves. This aspect is also relevant for the climate change analysis, due to the projected changes in the standard deviation of the temperature distribution (Ballester et al. 2010).	(In Introduction, L.70-74) “In this study, we selected the TX90p criterion, i.e., a heatwave occurs when the 90th percentile of maximum temperatures is exceeded for at least 3 consecutive days. This criterion is based on the anomaly of maximum temperature and includes information about the entire annual cycle, which eases the identification of productivity impacts above a certain threshold of temperature.”
We provide in a single paper a comprehensive analysis of the present (current heatwaves) and future (climate change projections) impacts of this hazard on the European economy by covering a timespan of 85 years (1981-2065) based on yearly estimates for the full time period.	This is implicit in how the paper was conceived and it is evident from the title and beginning of the abstract. Thanks to successive recommendations of the reviewers, we are now able to characterise impacts over the full time window 1981-2065. Please refer, for example, to Figure 4 for an overview of the past, present and future projected impacts of heatwaves in the area.
The spatial resolution of the climate data and economic model used are much finer in our case, enabling us to capture climatic heterogeneities as well as regional economic characteristics. It can also help policymakers in the formulation of evidence-based, local-level occupational and adaptation policies.	(In Introduction, L.100-104) “Thanks to its high level of spatial disaggregation, we were able to (1) better understand the distribution of costs between sectors and regions as well as the mechanisms of impact propagation and (2) characterise the areas more vulnerable to extreme heat stress as we quantify their present and expected future damages.”

Contribution of the paper	Where is it mentioned?
	(In Results: 'More exposed regions spearhead economic losses', L.204-209) "However, this work differs from the previous literature in several aspects. Among them, we study the impacts of extreme heat, understood as periods when a region's temperatures are abnormally high (rather than measuring the effect of summer average temperatures), consider all the productive economic sectors, and adopt a higher spatio-temporal resolution level." (In Conclusion, L.347-352) "(...) the proposed methodology can also be used as a tool for the assessment of future occupational health and the formulation of local-level adaptation policies. Finally, this study reinforces the need for spatially resolved, bottom-up approaches as a requisite to capture local socio-economic and climatic idiosyncrasies, crucial to analyse the potential economic consequences of climate change."
We use ERA5-Land (climate) hourly data, which lets us account for intra-daily temperature variation. as opposed to daily averages or 4-hour time snapshots	(In Methods: 'Heat stress index', L.395-399) "The use of hourly WBGT is essential to capture intra-daily heat variability, since the heat stress level encompasses the actual time devoted to work, avoiding the presence of potential biases resulting from the use of 24h, day- or night-time temperature (e.g., Casanueva et al. 2020 illustrate the clear underestimation of heat stress based on daily mean WBGT)."
We consider all the productive sectors in the economy, not a subset of them.	(In Results: 'More exposed regions spearhead economic losses', L.204-209) "However, this work differs from the previous literature in several aspects. Among them, we study the impacts of extreme heat, understood as periods when a region's temperatures are abnormally high (rather than measuring the effect of summer average temperatures), consider all the productive economic sectors, and adopt a higher spatio-temporal resolution level." Please see also Figure 3.
Using quarterly accounts, we attribute economic activity to the time of the year (quarter) this activity takes place, resulting in a more precise productivity shock characterisation.	Please see Methods: 'Accounting for seasonal patterns in economic activity' (L.513-521).
Three different heat-exposure functions were used in the historical analysis in order to test for the sensitivity of our results to the choice of these functions.	(in Results: 'More exposed regions spearhead economic losses', L.209-214) "We also tested the sensitivity of our findings to the choice of different heat-exposure functions (see Methods, Heat exposure functions). Resulting differences responded to how different heat exposure functions were constructed and were proportional across the three considered approaches, producing on average 11% less damages in the case of NIOSH and 30% in the case of Hothaps compared to ISO standards (Supplementary Fig. 4)."

Contribution of the paper	Where is it mentioned?
We provide a regional vulnerability assessment to heatwaves by considering environmental and economic exposure to extreme heat.	Refer to Supplementary Figure 2 and to the second paragraph of 'More exposed regions spearhead economic losses' (L.175-182).
Our approach is based on an economic model specifically regionalised and calibrated to reproduce the behaviour of the European economy. It unveils evidence about the regional disparities of the economic effects of this climate risk while it illustrates the driving factors of these differences	(In Results: 'More exposed regions spearhead economic losses', L.183-192) "Our results suggest that, in present times, direct impacts of heat on labour productivity take place mostly in outdoor sectors. However, these losses propagate to the entire economy. This propagation takes place mainly through the mechanism of intermediate goods used in the production processes, for example, in services relying on agricultural and industrial products or transport services as inputs. Given the complementarity between primary and intermediate inputs, indirect effects spread substantially through the service sector. In contrast, trade mechanisms, i.e., trade between regions, act as a buffer to mitigate this negative effect by substituting intermediate goods from less affected regions. These two mechanisms are embedded into our economic model." Refer also to Methods: 'Regional Computable General Equilibrium (CGE) model' (L.522-590)

To Reviewer #2's comments

I thank the authors for the detailed answers to my comments and for updating the manuscript. I only have a few specific comments left.

1. Line 31: The connection to the sentence before should be more emphasized. Maybe using “selected years” instead of “analysed years” or “analysed years with anomalous heatwaves” or something similar.

First, many thanks for the additional comments and concrete suggestions.

As suggested, we have modified the abstract to reflect that our conclusions about the current impacts of heatwaves are drawn from the in-depth analysis of four specific years. Along with this addition, we now underline that a historical benchmark effect is calculated and used for comparison.

The remaining abstract (reproduced below for convenience) has undergone a general redesign process. In particular, it has been shortened to make it more adequate to the length requirements of this journal.

Changes in the manuscript (Lines 23-35): “Extreme heat undermines the working capacity of individuals, resulting in lower productivity, and thus economic output. Here we analyse the present and future economic impacts of extreme heat in Europe. For the analysis of current impacts, we focused on heatwaves occurring in four recent anomalously hot years (2003, 2010, 2015, and 2018) and compared our findings to the historical period 1981-2010. In the selected years, the total estimated damages attributed to heatwaves amounted to 0.3%–0.5% of European gross domestic product (GDP). However, the identified losses were largely heterogeneous across space, consistently showing GDP impacts beyond 1% in more vulnerable regions. Future projections indicate that by 2060 impacts might increase in Europe by a factor of almost five compared to the historical period 1981-2010 if no further mitigation or adaptation actions are taken, suggesting the presence of more pronounced effects in the regions where these damages are already acute.”

2. Line 61: I guess it should be “strongly” or “largely” instead of “partly”. I assume that climate change is the main reason for the doubling.

Changed to “largely”, as suggested. Although, as mentioned in our previous response, “it is not possible to come with an exact figure since attribution studies are usually developed for specific events”, it is reasonable to think that the likely portion attributable to climate change is not minor. It is thus justified to emphasise the climate change effect. This amplifying effect of climate change is also illustrated in the sentence after the one referred here.

Changes in the manuscript (Lines 51-54): “The number of days exceeding the 90th percentile threshold (baseline period, 1970-2000) have doubled between 1960 and 2017 across the European land area (EEA, 2019), largely attributed to human-induced climate change (King et al., 2016; Diffenbaugh, 2020; Perkins-Kirkpatrick and Lewis, 2020).”

3. Line 62-64: This sentence is not entirely clear to me due to its structure. Particularly, what does “more than doubled the risk of heatwaves for some locations in particular events” mean? Please rephrase this sentence to make it better understandable.

The sentence has been rephrased as suggested and the subsequent sentence has also been modified to accommodate the new additions.

Changes in the manuscript (Lines 54-60): “According to Stott et al. (2004) and IPCC (2013), it is likely that the human influence has more than doubled the risk of some past heatwaves, such as the 2003 European heatwave. Along with the proliferation of these extreme weather events, climate change projections show that they might become more frequent and to last longer across all Europe during the 21st century (Fischer and Schär, 2010; IPCC, 2013; Russo et al., 2014).”

4. Line 98: I would add “forced by GHG emissions following the Representative Concentration Pathway 8.5”.

The suggested phrase has been added.

Changes in the manuscript (Lines 87-90): “We then applied this model to a high emission scenario represented by two climate model simulations forced by greenhouse gases emissions following the Representative Concentration Pathway 8.5 (RCP8.5, thereafter) over the years 2035–2064”.

5. Line 101: I guess this sentence refers to the signal-to-noise ratio, but the statement is a bit misleading since internal variability will also play an important role in the future. A similar statement in lines 439-440 is formulated more adequately. I would suggest to write this sentence here in a similar way as the one in lines 439-440.

Internal natural climate variability (natural fluctuations that arise in the absence of any radiative forcing of the planet) is present in all timescales. However, its relative importance compared to model and scenario uncertainties is reduced when lead time increases, i.e. towards the end of the century (Hawkins and Sutton, 2009). This sentence has been rewritten following the reviewer’s suggestion.

Changes in the manuscript (Lines 92-94): “It is less affected by uncertainties associated with the internal natural climate variability, which dominate for near-term projections, thus allowing the emergence of signals”.

6. Line 103: I would add “the latter uncertainty increases”.

We agree. Term “uncertainty” included as suggested (Lines 95-96).

7. Line 130: Longer than what? Please indicate.

That statement was meant to compare heatwaves initiated during summer with non-summer heatwaves, as is now indicated in the text. Also, the term ‘initiated’ has been included to emphasise that heatwaves can be initiated during a month/season, but can expand beyond that time window.

Changes in the manuscript (Lines 121-123): “However, heatwaves initiated during summer were on average two times longer (8.5 versus 4.3 days) and more severe than non-summer heatwaves.”

8. Line 145-146: “sustained positive temperature anomalies”.

The reviewer is right. Writing only “anomalies” can lead to confusion. The term “positive” has been added (Line 138).

9. Line 169: I would add that the losses mentioned here (i.e., the ones mentioned in parentheses) are also due to heat.

Thanks for the suggestion. The phrase between parentheses now reads:

Changes in the manuscript (Lines 162-163): “(0.2% GDP losses experienced on average over the period 1981-2010 due to extreme heat).”

10. Figures 1c and Figures 2a: I find it interesting that the patterns show strong differences for some regions (e.g. for Northern Italy in 2003 or for Croatia in 2003 and 2015). It might be worth to write a few remarks about these differences in the text.

We interpret this comment as how different heat/economic impacts are revealed, even for regions very close in space. We agree with this appreciation, which is actually behind the motivation for this paper. Differences in heat impacts (Fig.1c) influence but not directly determine differences in economic impacts (Fig.2a), given the unequal socioeconomic characteristics of each region. We have made an effort throughout the paper to account for and explain these differences and to underpin the need to assess climate economic impacts on a regional/local scale. We have added the suggested examples to reinforce the presence of heterogeneities, even for very close regions.

Changes in the manuscript (Lines 151-153): “Our analysis of regional heatwaves shows that these events are largely heterogeneous in terms of spatial and temporal characteristics (see, for example, Northern Italy in 2003 or Croatia in 2003 and 2015 in Fig. 1c).”

11. Line 214: Rather “and” than “or”.

Thanks for noting this. The term “and” has been used and “spatio-temporal” rather than “spatial” introduced. As per the suggestions of Reviewer #1, some other features have been included. Please refer to lines 204-209 in the revised manuscript.

12. Line 220: Maybe I have missed it, but what is the reason that the ISO standard is selected for this study and not, e.g. Hothaps? This should be shortly explained.

ISO standards are based on heat safety regulation and have been used in several impact studies. In contrast, the Hothaps exposure-response functions are derived from only two industry-field studies, so their functional parameters are subject to greater uncertainty. This explanation has been added to the Methods.

Changes in the manuscript (Lines 487-491): “However, the Hothaps functions are subject to great parameter uncertainty due to being based on a few empirical studies. Therefore, we adopted the ISO standards as our benchmark functions and used the NIOSH and Hothaps functions to test for the sensitivity of our estimates.”

13. Line 233: I would find the formulation “do likely not have a significant impact” more intuitive.

This expression has been reformulated following the reviewer’s suggestion.

Changes in the manuscript (Lines 226-227): “(…), these adaptation effects do likely not have a significant impact on our current estimates.”

14. Line 264: potentially highlight that ISO is the one used in this study.

The different approaches used by each study have now been clarified.

Changes in the manuscript (Lines 255-257): *“We attribute this difference with respect to our findings mainly to the heat-exposure function used (Hothaps in Orlov et al., 2020; ISO in this study), as ISO is more sensitive to lower temperatures (...).”*

15. Line 267: Who uses the dynamic framework? Orlov et al or this study? Please specify this more clearly in the sentence.

This aspect has been clarified.

Changes in the manuscript (Lines 258-260): *“(...) as well as to differences in the parametrisation of the economic model and the experimental design, as the cited work is based on a dynamic framework.”*

16. Line 276-277: “southern European countries”.

Changed as suggested (Lines 265-266).

17. Line 287: Why is the p-values 0? It should be larger than 0. Is this a mistake? If it is very low, it could be indicated as $p < 0.001$.

The p-values expressed here are very low, being their first three decimal positions equal to zero. We have followed the reviewer’s suggestion, i.e. “ $p < 0.001$ ”, to indicate this throughout the revised manuscript.

18. Line 304: I would mention here the years that were analysed.

Included as suggested (Lines 293-294). See also comment #1 of this reviewer.

19. Line 343: “Do likely not” instead of “does not seem”, since from your response to my previous comments I understand that the effect of adaptation measures were not tested.

Expression amended following this reviewer’s suggestion (Line 331).

20. Line 514: It is still unclear to me whether the population or the climate data were spatially interpolated. Please be more specific here! And if it is the population data, nearest neighbour interpolation could cause that the total population in Europe is not conserved. In that case, I would strongly recommend to use conservative remapping.

As indicated in our previous response (see comment #32), population data were used as the reference dataset (hence, population count is preserved). WBGT data were assigned to each population gridbox using the nearest neighbour. Now, we make this point explicit in the text. Considering that the resulting interpolated data were further spatially averaged at the regional level, we think that the effect of the interpolation method at this stage is small.

Changes in the manuscript (Lines 502-505): *“The spatial mismatch between the resolution of population (0.25°) and climate data (0.1° for the reanalysis and 0.5° for the bias-corrected climate model data) were handled by interpolating the latter towards the population grid using the nearest neighbour.”*

21. Line 520-521: Remove “to these population projections” since also the interpolation of climate data is mentioned in this sentence.

Corrected as suggested (Lines 511-512).

22. Figure 2: What does “(50th percentile)” at the end of the caption refer to? If it is just the definition of the median, I think it can be omitted here as people should know what the median is. If it is something else, it should be explained more precisely.

It refers to the median. As the reviewer points out, this clarification is perhaps redundant. The phrase has been removed.

23. Figure 4a: What do the boxes and whiskers indicate? The red line should also be explained.

The following clarification has been added to the caption of Fig. 4.

Changes in the manuscript: “Each boxplot shows the interannual distribution of total European, annually estimated impacts over different time periods. In-depth analysed years (2003, 2010, 2015, and 2018) are highlighted. Boxes cover the interquartile range (IQR, 25th-75th percentiles) of the damage distribution and whiskers show the values contained within $\pm 1.5 \cdot \text{IQR}$. Thick solid lines denote the estimated median (multi-model median in the climate change analysis) GDP impact over each time period.”

To Reviewer #3's comments

Most of my points have been properly assessed, reanalyzed and changed in the manuscript, except for my comment of considering different RCP-SSP combinations:

1. The population projections and GDP growth rates both affect exposure and vulnerability across Europe - and one important aspect here is the regional distribution of population (e.g. depopulation of Eastern Europe, movements towards Western/Central). Ideally, also assumptions on structural change would be incorporated, e.g. more fossil based energy sector in SSP5. While there is little difference in RCPs until mid-century, looking only into RCP8.5, and therefore ruling out other SSPs, is a too simple procedure.

I do agree that RCP8.5 is most consistent, according to IAM evidence, in terms of emissions only with SSP5, but several papers also look into the combination with SSP3. Alternatively, plausible combinations of RCPs and SSPs are assessed, e.g. RCP2.6-SSP1, RCP4.5-SSP2, RCP8.5-SSP5.

To conclude: while I understand that a new analysis based on other RCPs is beyond the scope of the paper, the conclusion that "including socioeconomic dynamics [...] would imply lower economic damages" cannot be drawn by only looking into SSP5. Also the conclusion that increases in adaptive capacity more than offset increases in exposure seems a strong conclusion. My suggestion is therefore to delete lines 267-272 and to expand on the caveat in lines 272-275 by describing potential mechanisms that could reduce or amplify damages. Similarly, in the conclusion the half-sentence "especially during the second half of the century" in line 355 should be deleted, because SSP differences are already pertinent earlier. I also do not agree that "hourly-level and spatially downscaled projections of heat stress measures" are the most pertinent gap for doing different RCP-SSP combinations. It is rather that existing regional SSP narratives for Europe (e.g. Kok et al. 2019) need to be expanded towards structural economic change.

We thank this reviewer for his/her additional comments and specific input to the dynamic experiment proposed in the previous revision. As emphasised in our previous response, the scope of our climate change assessment was limited, mainly because of the lack of climate projections of hourly-level WBGT data, a rare product. We benefitted from an effort of the HEAT-SHIELD project (<https://www.heat-shield.eu/>), some of whose partners are coauthors of this paper, which provided the climate projections. These projections were focused on high emission scenarios (RCP8.5), thus our inability to systematically explore different RCP-SSP combinations. We, however, analysed the (only feasible) RCP8.5-SSP5 combination to accommodate your suggestion.

We agree with the reviewer in that extracting any conclusion from the exploration of a single (RCP-SSP) possible state of the world is dangerous, as we highlight in the main text (lines 260-264, 338-345) and in the Supplementary Discussion.

In agreement with the reviewer's comment, in the revised version of the manuscript, we have removed the take-out messages of the dynamic experiment. This information has now been confined to the Supplementary Discussion. We kept the part discussing the relevance of a comprehensive RCP-SSP impact assessment to study future climate and socioeconomic uncertainties.

Changes in the manuscript (Lines 260-264): “Carefully exploring different *climate and socioeconomic (RCP-SSP) combinations* is vital to assess the uncertainty posed by future projected scenarios to the present results (the reader is referred to the *Supplementary Discussion* for an overview of the projected outcomes under the scenario *RCP8.5-SSP5*).”

Regarding the gaps for implementing a comprehensive regional assessment of heatwaves' impacts under different RCP-SSP combinations, we have expanded the obstacles to be overcome by including the suggestion of this reviewer related to adapt SSP narratives towards changes in the regional economic structure, currently endogeneised in the CGE.

Changes in the manuscript (Lines 342-346): “However, for this kind of assessment, hourly-level and spatially downscaled projections of heat stress measures should first be available under different climate forcings *and existing regional SSP narratives* need to be able to accommodate changes in the regional economic structure (Kok et al., 2019).”

This is an aspect that we also highlight in the Supplementary Discussion (“Some other aspects should also be controlled with more detail as, for example, the evolution of the sectoral economic composition”).

Finally, as per this reviewer's suggestion, the phrase “especially during the second half of the century” has been removed when referring to the need for exploring different RCP-SSP combinations in future assessments.

2. Finally, one minor point is that in Knittel et al. (2020) GDP effects reported for Germany are that of labor productivity changes on a global scale, so also changes embodied in imports. Please adjust lines 289-290 accordingly.

The reviewer is right in pointing out that Knittel and coauthors assess labour impacts on a global scale, so estimated GDP impacts are also measuring the effects of imports, specifically of non-EU imports. Our approach takes into account intra-EU trade and trade flows with the rest of the world, but the latter do not include the effect of heat-induced productivity damages. We now reflect this in the main text.

Changes in the manuscript (Lines 277-280): “(...) with Germany being projected to experience a negative impact *on GDP* of 0.5% by 2050, similar to what is reported in Knittel et al. (2020), who also include the productivity effects of non-EU imports in their estimates.”

References

Ballester J, Rodó X, Giorgi F (2010) Future changes in Central Europe heatwaves expected to mostly follow summer mean warming. *Clim Dyn* 35, 1191–1205.

Hawkins E, Sutton R (2009) The potential to narrow uncertainty in regional climate predictions. *Bull Am Meteorol Soc* 90(8): 1095–1108.

Knittel N, Jury MW, Bednar-Friedl B, Bachner G, Steiner AK (2020) A global analysis of heat-related labour productivity losses under climate change—implications for Germany's foreign trade. *Climatic Change* 160:251–269.

Kok K, Pedde S, Gramberger M, et al. (2019) New European socio-economic scenarios for climate change research: operationalising concepts to extend the shared socio-economic pathways. *Reg Environ Change* 19, 643–654. <https://doi.org/10.1007/s10113-018-1400-0>.

REVIEWERS' COMMENTS

Reviewer #1 (Remarks to the Author):

Thank you for the response. There are descriptions of the contributions of this study as the authors indicate in the table in the rebuttal. However, I still think readers will not understand whether it is new in this study or is a standard, already-existing practice in this field. A suggestion from my side is to insert a paragraph that is devoted to reviewing the relevant literature (e.g., Orlov et al. (2019 and 2020), Knittel et al. (2020)) in the introduction section, and explain how researchers have estimated the impacts before this study.

Reviewer #2 (Remarks to the Author):

I thank the authors for their detailed answers to my comments.
I have only one final remark: The second sentence in the abstract should include that the study specifically deals with economic damages due to reduced labour productivity, not general impacts of heat on the economy.

Reviewer #3 (Remarks to the Author):

All my comments have now been properly addressed.

Response to Reviewers

Reviewer #1 (Remarks to the Author):

Thank you for the response. There are descriptions of the contributions of this study as the authors indicate in the table in the rebuttal. However, I still think readers will not understand whether it is new in this study or is a standard, already-existing practice in this field. A suggestion from my side is to insert a paragraph that is devoted to reviewing the relevant literature (e.g., Orlov et al. (2019 and 2020), Knittel et al. (2020)) in the introduction section, and explain how researchers have estimated the impacts before this study.

Thank you. We understand the concern. As the reviewer suggests, we now briefly discuss the previous relevant literature (Orlov et al. 2019, 2020; Knittel et al. 2020) in the Introduction.

Reviewer #2 (Remarks to the Author):

I thank the authors for their detailed answers to my comments.

I have only one final remark: The second sentence in the abstract should include that the study specifically deals with economic damages due to reduced labour productivity, not general impacts of heat on the economy.

We are grateful for the reviewer's careful and thoughtful engagement with our work. We have rewritten the affected sentence.

Changes in the manuscript (Lines 24-26): *"Here we analyse the present and future economic damages due to reduced labour productivity caused by extreme heat in Europe".*

Reviewer #3 (Remarks to the Author):

All my comments have now been properly addressed.

We are thankful for the reviewer's positive assessment of our work.

References

Knittel N, Jury MW, Bednar-Friedl B, Bachner G, Steiner AK (2020) A global analysis of heat-related labour productivity losses under climate change—implications for Germany's foreign trade. *Climatic Change* 160: 251–269.

Orlov A, Sillmann J, Aaheim A, Aunan K, de Bruin K (2019) Economic losses of heat-induced reductions in outdoor worker productivity: a case study of Europe. *Economics of Disasters and Climate Change* 3: 191–211.

Orlov A, Sillman J, Aunan K, Kjellstrom T, Aaheim A (2020) Economic costs of heat-induced reductions in worker productivity due to global warming. *Global Environmental Change* 63:102087.